



# Development of a methodological framework for the assessment of seismic induced tsunami hazard through uncertainty quantification: application to the Azores-Gibraltar Fracture Zone

Vito Bacchi[1], Ekaterina Antoshchenkova[1], Hervé Jomard[1], Lise Bardet[1], Claire-Marie Duluc[1], Oona Scotti[1], Hélène Hebert[2]

[1]Institute for Radiological Protection and Nuclear Safety (IRSN), Fonteany-aux-Roses, 92262, France
[2]CEA, DAM, DIF, 91297 Arpajon Cedex, France

*Correspondence to*: Vito Bacchi (vito.bacchi@irsn.fr)

**Abstract.** The aim of this study is to show how a numerical database constructed using "uncertainty quantification techniques" can be a useful tool for the analysis of the tsunamigenic potential of a seismic zone. This methodology mainly relies on the construction and validation of some "emulators", or "meta-models", used instead of the original models in order to evaluate and quantify the uncertainty and the sensitivity of the tsunamis height at a given location to the seismic source parameters. The proposed approach was tested by building a numerical database of nearly 50,000 tsunamis scenarios generated by the Azores-Gibraltar Fracture Zone (AGFZ) and potentially impacting the French Atlantic Coast. This seismic area was chosen as test-case because of its complexity and the large uncertainty related to its characterization of tsunamigenic earthquake sources. Finally, the tsunami hazard resulting from uncertainty quantification was presented and discussed with respect to the results which can be obtained with a more classical deterministic (or scenario-based) approach. It must be underlined that the results from this study are the illustration of a general methodology through a case study with simplified hypothesis, which is not an operational assessment of tsunami hazard along the French Atlantic Coast.

## 1 Introduction

Tsunami hazard science starts to develop intensively about 50 years ago (Zoback et al., 2013) following four main axes of development: the (i) identification of past tsunamis in the geologic record on land, the (ii) characterization of tsunamis of different source types, the (iii) hydrodynamic modelling to predict run-up and inundation areas and the (iv) ocean buoy systems for tsunamis detection and recording. Even if it is generally accepted that large-scale tsunamis can be generated by three main types of geologic events, namely the earthquakes, the submarine and subaerial landslides, and a variety of mechanisms associated with volcanism, in this study we only focused on tsunamis generated by earthquakes.

Earthquakes are most common source of large-scale tsunamis, with dip-slip earthquakes (i.e. with vertical movement) being more tsunamigenic than strike-slip earthquakes (i.e. with horizontal movement). It is generally observed that only large magnitude earthquakes (M>6.5) lead to widely observable tsunamis, because earthquakes needs to be large



enough to be able to produce significant displacements at the sea bottom surface (Wells and Coppersmith, 1994). The typical tectonic environment able to generate tsunamigenic earthquakes is the subduction zone where major tsunami earthquakes occur on shallow inter-plate thrusts. However, they may also be generated by outer rise earthquakes, either within the subducting slab or the overlying crust (Satake and Tanioka, 1999). Other oceanic converging boundaries may also be

responsible for tsunamis even if not classified as subduction zones, as the North-African coasts for instance (Álvarez-Gómez et al., 2011). On the opposite, diverging plate boundaries are also able to generate tsunamis although earthquakes are in general lower in magnitudes in comparison to subduction zones (Ambraseys and Synolakis, 2010). Finally, purely strike-slip earthquakes may also affect oceanic floors, but they are in general not able to generate major tsunamis, unless a significant earthquake vertical component exists or if steep slopes are affected (Tanioka and Satake, 1996).

Considering the variety and the complexity of the geophysical mechanisms involved in tsunami generation, the tsunami hazard assessment (THA) for seismic sources is generally associated with strong uncertainties (aleatory and epistemic). Generally, two approaches are widely used by the scientific community to quantify the THA: the scenario-based (or deterministic) and the probabilistic approaches (Omira et al., 2016). The probabilistic approach reposes on four basic steps (Geist and Lynett, 2014): (i) the specification of source parameters, including source probabilities for all relevant

tsunami sources, (ii) the choice of a probability model that describes source occurrence in time (most often Poisson), (iii) the hydrodynamic modelling for each source location and set of source parameters to compute tsunami hazard characteristics at a site (or a point) of interest and (iv) the aggregation of the modelling results and incorporation of uncertainty. Probabilistic Tsunami Hazard Analysis (PTHA) calculates the likelihood of tsunami impact using a large number of possible tsunami sources and including the contribution of small and large events from both near- and far-field (Omira et al., 2016). The

objectives of PTHA are to condense the complexity and the variability of tsunamis into a manageable set of parameters and to provide a synopsis of the tsunami hazard along entire coastlines in order to help identifying vulnerable locations along the coast and specific tsunamis source regions to which these vulnerable locations on the coastline are sensitive (Geist and Parsons, 2006). Early studies are based on different assumptions (e.g. (Rikitake and Aida, 1988) and (Downes and Stirling, 2001)) and a surge of PTHA studies started in the early 2000s up to present (e.g. (Geist and Parsons, 2006;Annaka et al.,

2007;Geist et al., 2009;González et al., 2009;Blaser et al., 2012)), in particular after the major tsunamis in Indonesia in 2004 and in Japan in 2011. However, though the probabilistic approach appears to be highly enticing in theory (Geist and Lynett, 2014), its practical implementation is often clouded by lack of complete understanding of the underlying parameters and phenomena and its temporal variations (Roshan et al., 2016). Moreover, in seismic area with no (or poor) knowledge of the faults characteristics it is not trivial to associate a probability distribution to the seismic inputs parameters. In this case,

deterministic approaches seem more adapted than the probabilistic method.

Scenario-based (or deterministic hazard assessment) approach offers simple, directly relatable events or scenarios describing different possible characteristics of the event. The tsunamis hazard is assessed by means of considering particular source scenarios (usually maximum credible scenario with respect to the actual knowledge of the tsunamigenic seismic sources) and then estimating the coastal tsunamis impact through numerical modelling (JSCE, 2002;Lynett et al., 2016),



without addressing the likelihood that such a big event occurs (Omira et al., 2016). The outputs of the deterministic analysis are, in general, tsunami travel time, wave height, flow depth, run-up, and current velocity maps corresponding to the chosen scenario (Omira et al., 2016). Nevertheless, the classical deterministic approach for THA could be hampered by the use of specific values of input parameters which may be subjective depending on the person or group carrying out the analysis

(Roshan et al., 2016). A good example is the 1755 Lisbon tsunami, generated by an earthquake in the Azores Gibraltar Fracture Zone (AGFZ). The Great 1755 Lisbon earthquake generated the most historically destructive tsunami near Portugal coasts (Santos and Koshimura, 2015). However, source location and contemporary effects of such tsunami are not precisely identified. Several earthquake scenarios have already been published in the literature since last decades (Baptista et al., 1998;Grandin et al., 2007;Gutscher et al., 2006;Horsburgh et al., 2008;Johnston, 1996;Zitellini et al., 1999), mainly based on

a comparison of several fault hypothesis based on geological properties of the Azores-Gibraltar fault zone (AGFZ). All of these studies show how variable parameters of the seismic source can be, depending on the studied fault and focal mechanisms.

For THA, the uncertainties related to model parameters are generally not taken into account in a rigorous and robust way (Lynett et al., 2016;JSCE, 2002), considering the mathematical framework available in the literature (Saltelli et al.,

2004;Saltelli et al., 2008;Faivre et al., 2013;Iooss and Lemaître, 2015) and the classical approach to deal with uncertainties consists to perform a limited number of deterministic simulations. For instance, Allgeyer et al. (Allgeyer et al., 2013) modelled the impact of three different scenarios for the 1755 earthquake on the French Atlantic Coast, with a focus on La Rochelle harbour. The authors show that, depending on the source hypothesis and tide conditions, several areas (western part of the island of Ré and northern coast of the island of Oléron) may have experienced a moderate impact from 0.5 to 1 m for

this tsunami event. Recently, Roshan et al. (Roshan et al., 2016) tried to improve the THA procedure detailed in (Yanagisawa et al., 2007) in order to better evaluate the effects of the seismic source uncertainties. The authors focused on a possible range of parameters which could produce more robust estimates of hazard instead of using single value of each seismic source parameter (Roshan et al., 2016). Basically, their methodology is based on the classical steps of an uncertainty propagation study: (i) the identification and characterization of tsunamigenic earthquake sources, (ii) the estimation of

rupture parameters and associated parametric variations, (iii) the intensive exploration of a validated numerical model and, finally, (iv) the parametric study capturing the uncertainties through the numerical simulations of the tsunami waves associated to different model inputs parameters varying within a given range. The deterministic numerical results are then processed and hazard maps of mean simulated water height are produced to assess the tsunami hazard (Roshan et al., 2016). Finally, the mean simulated water height of the different tsunami scenarios is employed to design flood level due to tsunami

along Indian coast (Roshan et al., 2016).

The objective of this exploratory and innovating methodological work was not directly the analysis of the benefits of uncertainty quantification on the THA (Roshan et al., 2016). In fact, the authors focused on a limited number of seismic source parameters (the dip angle, the strike and the source location) leading to a limited number of tsunami scenarios (around 320) in order to introduce an uncertainty associated to the main known fault characteristics. In this sense, they were not





interested, for instance, to the robustness of the mean simulated tsunami height considering the low number of simulations. Moreover, the sensitivity of model results to the input parameters was not studied (Roshan et al., 2016). Finally, the other seismic source parameters (e.g. the mean rupture length and width, the slip and the magnitude) were considered as constant for each tsunami scenario and their uncertainty was not evaluated, thus strongly limiting the impact of their study with

respect to a more classical uncertainty quantification approach.

In this context, the objective of this work is to analyse the benefits of a methodology based on uncertainty quantification techniques (i.e. (Saltelli et al., 2000;Saltelli et al., 2008)) on the estimation and the analysis of the tsunami hazard. The proposed methodology, presented in section 2, is mainly based on the statistical criteria classically employed in uncertainty quantification studies (Saltelli et al., 2000;Saltelli, 2002;Saltelli et al., 2008;Iooss and Lemaître, 2015) and the

use of an emulator of the tsunami numerical model, called meta-model, allowing to perform a large number of tsunami scenarios and, consequently, intensively explore the tsunamigenic potential of a seismic zone. The construction (and validation) of meta-models representing the maximum tsunami height at four selected location along the French Atlantic Coast is presented in section 3. These emulators permitted to build (section 3) a numerical database of a wide bunch of the maximum possible tsunamis generated by the AGFZ. This tsunamis database is then analysed in section 4 and a hazard level

(with the associated uncertainty) from uncertainty quantification is finally proposed.

It must be noted that the aim of this study is to show how a numerical database constructed using "uncertainty quantification techniques" can be a useful tool for the analysis of the tsunamigenic potential of a seismic zone. As a consequence, this paper illustrates a methodology through a case study, which is not an operational assessment of tsunami hazard along the French Atlantic Coast.

## 20  2 Presentation of the methodology: theoretical background

### 2.1 Steps of uncertainty quantification and global sensitivity analysis

IRSN conducted cooperative research activities since 2011 (Abily et al., 2016;Nguyen et al., 2015) in order to investigate feasibility and benefits of Uncertainty Quantification "UQ" and Global Sensitivity Analysis "GSA" when applied to hydraulic modelling. The proposed methodology is based on the classical uncertainty study steps (Saltelli et al.,

2008;Saltelli et al., 2004;Faivre et al., 2013;Iooss and Lemaître, 2015), as represented in Figure 1. The first step (Step A), "the definition of the problem", consists in the definition of the input variables of the model, the choice of the numerical model and of the output variables of interest. Once step A has been carried out, the next step (Step B) consists in the identification of the inputs affected by uncertainty, and consequently the definition of the variation intervals and the probability density functions (PDF) for each parameter uncertainty. We remind that, for a random variable X defined on an

$[a,b]$ interval, a PDF $f_X(X)$ is defined as follows:





$$P(X \in [a,b]) = \int_a^b f_X(X)dx \qquad (1)$$

Most of the time, as in the present study, the probability distributions of random inputs is unknown. Hence, expert's knowledge or the experimental evidence available is used as the basis for the PDF definition. The Step C requires propagating in the model the uncertainty defined in the previous step. The objective is to compute the PDF $f_y(Y)$ of the quantities of interest y. This step is most complex and computationally demanding. There are various methods to deal with

this step, and we choose, in this study, to build a kriging surrogate (or kriging meta-model), presented in the next paragraph and which equations are detailed in APPENDIX 1. Finally, a better understanding of uncertainties can be achieved by analysing the contribution of the different sources of uncertainty to the uncertainty of the variables of interest (Step D). For each couple "criteria of the study / propagation method used in Step C", post-treatment procedures are used in order to rank the sources of uncertainty. It is important to note that an uncertainty study rarely stops after a first processing of steps A, B,

C and D, and the last step then plays a crucial role. Indeed, the ranking results highlight the variables that truly determine the relevancy of the final results of the study. Step A and Step B are strictly related to the objectives of the study, in this case the Tsunami Hazard Assessment, and will be defined in section 3. The methods employed for Step C and D are described in the next paragraphs.

**2.2 The Monte-Carlo method for Uncertainty Propagation (Step C)**

The Monte-Carlo method used in this study requires random generation of input variables from their probability distributions. The resulted sampling of a given size $N$ is a $N \times V$ matrix, $V$ being the number of uncertain parameters. Each row of the matrix $x^i = (x_1,...,x_V)^i$ represents a possible configuration of the input parameters of the model. Corresponding realizations of the output are generated by successive deterministic simulations with each configuration of the inputs. Statistical estimators of the response $Y = (Y_1,...,Y_N) = (G(x^i))_{i \in \{1,...,N\}}$ can therefore be computed from the output as

follows:

$$E[Y] = \mu_Y = \frac{1}{N}\sum_{i=1}^{N} G(x^i) \qquad (2)$$

$$Var(Y) = \frac{1}{N-1}\sum_{i=1}^{N} [G(x^i) - \mu_Y]^2 \qquad (3)$$

$$\sigma_Y = \sqrt{Var(Y)} \qquad (4)$$

Where $E[Y] = \mu_Y$, $Var(Y)$ and $\sigma_Y$ are, respectively, the mean, the variance and the standard deviation of the response $Y$ given by the model $G$. These statistical moments are useful for both the uncertainty propagation and the uncertainty sensitivity analysis (see next paragraph).





The convergence order of the Monte-Carlo sampling method is given by the Central Limit Theorem (Faivre et al., 2013) as $O\left(\dfrac{1}{\sqrt{N}}\right)$, implying that a large amount of simulations are necessary to obtain convergent statistics. Moreover, the number of simulations increase with the increasing of the uncertainty parameters, which is computationally demanding. As a consequence, other techniques have been developed to reduce the number of simulations by preserving the quality of the

statistical results (i.e. quasi Monte-Carlo screening methods, meta-models methods, and so on). For instance, in this study we used kriging meta-models instead of the original model, drastically reducing the computational time.

## 2.3 Sensitivity Analysis (Step D)

The role of the sensitivity analysis is to determine the strength of the relation between a given uncertain input parameters and the model outputs (Saltelli et al., 2004). In this study, we focused our attention on Global Sensitivity Analysis

approaches which rely on sampling based methods for uncertainty propagation, willing to fully map the space of possible model predictions from the various model uncertain input parameters and then, allowing to rank the significance of the input parameter uncertainty contribution to the model output variability (Baroni and Tarantola, 2014). GSA approaches are well suited to be applied with models having nonlinear behaviour and when interaction among parameters occurs (Saint-Geours, 2012). The GSA approach we focused on in this study is the Sobol index computation, that considers the output hyperspace

*(x)* as a function *(Y(x))* and performs a functional decomposition (Iooss and Lemaître, 2015;Iooss, 2011) or a Fourier decomposition (FAST method) of the variance (Saltelli, 2002;Saltelli et al., 1999).

With respect to deterministic methods, as the Morris method (Morris, 1991), probabilistic GSA approaches relying on Sobol index computation go one step further, allowing to quantify the contribution to the output variance of the main effect of each input parameters (Sobol, 2001;Saltelli et al., 1999;Saint-Geours, 2012;Sobol, 1993). The definition of Sobol

Indices is a result of the ANOVA (ANalysis Of VAriance) variance decomposition. To be able to write the variance decomposition, some mathematical hypothesis are required (see i.e. (Saltelli et al., 2008) or Faivre et al. (Faivre et al., 2013)), briefly recalled here.

Let us consider an integrable model $G$ on the domain $A = A_1 \times ... \times A_V$. It exists an unique decomposition of the model:

$$G(Y) = f_0 + f_1(Y_1) + ... + f_m(Y_m) + f_{1,2}(Y_1, X_2) + ... + f_{v-1}(Y_{v-1}, Y_v) + ... + f_{1,...,v}(Y_1,...,Y_v) \qquad (5)$$

Where all the functions fi are mutually orthogonal, implying that $\langle f_i | f_j \rangle = 0$ if $i \neq j$, with the scalar product defined as follow:

$$\langle f | g \rangle = \int_A f(x)g(x)\pi(x)dX \qquad (6)$$



Where $\pi$ is a probability distribution defined on $A$. Under these assumptions, given a set of $V$ independent uncertain parameters $X$, the variance of a response $Y = G(X)$ can be calculated, using the total variance theorem, as follows:

$$Var(Y) = \sum_{i=1}^{V} D_i(Y) + \sum_{i<j}^{V} D_{ij}(Y) + ... + D_{12..V}(Y) \tag{7}$$

Where $D_i(Y) = Var[E(Y|X_i)]$ and $D_{ij}(Y) = Var[E(Y|X_i X_j)] - D_i(Y) - D_j(Y)$ and so on for higher order interactions. $E(Y|X_i)$ is the $Y$ conditional expectation with the condition that $X_i$ remains constant. The so-called "Soblol' indices" are obtained as follows:

$$S_i = \frac{D_i(Y)}{Var(Y)}, \quad S_{ij} = \frac{D_{ij}(Y)}{Var(Y)}, \quad ... \tag{8}$$

These indices express the share of variance of $Y$ that is due to a given input of input combination. For instance, $S_1$ is the first order sobol index. The number of indices grows in an exponential way with the number $V$ of dimensions. So, for computational time and interpretation reasons, Homma and Saltelli (Homma and Saltelli, 1996) introduced the so-called "total indices" or "total effects" that write as follows:

$$S_{Ti} = S_i + \sum_{i<j} S_{ij} + \sum_{j\neq i, k\neq i, j<k} S_{ijk} + ... = \sum_{l\in\neq i} S_l \tag{9}$$

Where $\neq i$ are all the subsets of $\{1,...,V\}$ including i.

To estimate Sobol' indices, a large variety of methodology are available, as the so-called "extended-FAST" method (Saltelli et al., 1999), already used in previous studies by IRSN (Nguyen et al., 2015). In this study, we used the methodology proposed by Janssen (Jansen et al., 1994) already implemented in the open source sensitivity-package R (Pujol et al., 2016). This method allows the estimation of first order and total Sobol' indices for all the factors "$v$" at higher total cost of "$v \times (p + 2)$" simulations (Faivre et al., 2013).

## 2.4 Meta-model design and validation

For the evaluation of numerical codes in industrial applications, where the computational code is expensive in computation time, so that it cannot be evaluated intensively (e.g. only several hundred calculations are possible), it is usually not easy to estimate some numerical parameters (i.e. the sensitivity Sobol indices), requiring for each input several hundreds or thousands of evaluations of the model (Iooss and Lemaître, 2015). As a consequence, a meta-model instead of the model is generally used in the estimation procedure. A meta-model is a mathematical model of approximation of the numerical model, built on a learning basis (Fang et al., 2006). The meta-model solution is a current engineering practice for estimating sensitivity indices (Gratiet et al., 2016). In this study, we use the Gaussian process meta-model (also called kriging) (Sacks et



al., 1989), which good predictive capacities have been demonstrated in many practical situations (see (Marrel et al., 2008) for example). As a consequence, kriging meta-model will be useful both for sensitivity analysis and for a numerical database construction through uncertainty quantification (as reported in the next section). The classical steps for meta-models construction and validation are reported in (Faivre et al., 2013) or (Saltelli, 2002;Saltelli et al., 2008), and can be shortly

summarized as follow (see Table 1). The step related to the estimation of the meta-model parameters consists in the mathematical construction of the meta-model and it obviously depends on the chosen meta-model. The theoretical background for kriging-metamodel design used in this study is fully detailed in (Roustant et al., 2012), which is described in the R-packages DiceKring and DiceOptim that we used for meta-models construction. The basic hypothesis and the mathematical equations of the meta-models used in this study are reported in the APPENDIX 1 for the interested reader.

### 2.4.1 The initial design for meta-model fitting

For computer experiments, selecting a *design database* (or space filling design) is a key issue in building an efficient and informative meta-model. Several authors have shown that the space filling design are well suited to meta-model fitting (Iooss et al., 2010). The design of the experience is the choice of the input parameters (and, consequently, of the initial simulations) to use in order to build the most accurate meta-model at a minimum computational coast (minimum number of

model simulations).The design of the experience can depend on the chosen meta-model (Kleijnen, 2005). For Gaussian emulators as kriging, the most popular design type is the Latin Hypercube Sampling (LHS) (McKay et al., 1979). LHS is a simple technique which offers flexible design size for any number of simulation inputs (Kleijnen, 2005). Moreover, LHS has a good space (of the input parameters) filling, implying that the model input parameters are well represented by simulations. Despite these advantages, in this study we do not use the LHS design but the numerical results issued by a preliminary

research activity (Antoshchenkova et al., 2016), as explained in section 3.3.

### 2.4.2 The partitioning of the original data base

It is generally assumed that the simulations performed to build the meta-model (the *design database*) can be partitioned in a *training set* (data used for the estimation of the meta-model) and a *validation set* (simulations performed to test the ability of the meta-model to reproduce the model results with other data). According to (Amari et al., 1997), to obtain

an optimum unbiased meta-model, nearly 80% of the simulations of the *design database* must be used for the *training set* and 20% for the *validation set*. This ratio is used in our study.

### 2.4.3 The validation of the chosen type of meta-model: training set

In order to assess if a given meta-model (i.e. kriging, gam method, random forest, chaos polynomials, and so on…) is adapted to reproduce the model behaviour for the design data base, probably the simplest and most widely used method is

the *cross-validation* or *K-fold* cross-validation method (Hastie et al., 2002). This step is performed with the *training set*. The principle of cross-validation is to split the data into $K$ folds of approximately equal size $A_1A1,...,A_kAK$. For $k = 1$ to $K$, a





model $\hat{Y}^{(-k)}$ is fitted from the data $U_{j \neq k} A_k$ (all the data except the $A_k$ fold) and this model is validated on the fold $A_k$.

Given a criterion of quality $L$ as the Mean Square Error (Eq. 10):

$$L = MSE = \frac{1}{n}\sum_{i=1}^{n}(\hat{y}_i - y_i)^2 \tag{10}$$

The quantity used for the "evaluation" of the model is computed as follow:

$$L_k = \frac{1}{n/K}\sum_{i \in A_k} L\left(y_i Y^{(-k)}(x_i)\right) \tag{11}$$

Where $\hat{y}_i$ and $y_i$ are, respectively, the meta-model and the model response and $n$ is the number of simulations in the *kth*

5   sample. Finally, the cross-validation used in this study is evaluated through the mean of the quantity $L_k$ computed for each fold:

$$L\_CV = \frac{1}{K}\sum_{k=1}^{K} L_k \tag{12}$$

It must be noted that when K is equal to the number of simulations of the training set, the cross-validation method corresponds to the leave-one-out technique (Oliveira, 2008). The methodology employed is described in the DiceEval R-package (Dupuy et al., 2015) reference-manual. We considered $K = 10$ and compute the MSE between meta-model approximation and the k-fold observations for each fold (*k* times). The mean MSE is an index of the mean differences between meta-model predictions and model results. As a consequence, it is an index of the ability of the chosen meta-model to reproduce the original model. The more this indicator approaches zero, the more the meta-model can be considered accurate.

### 2.4.4 Performance of the meta-model and residual analysis: validation set

15    This last operation is performed on the *validation-set* (composed of the 20% of the *design database*). The objectives are to evaluate the meta-models built on the entire *testing set* (after cross-validation) outside their construction domain (*training set*) and to estimate the uncertainty related to the use of the meta-model instead of the original model. The first statistical parameter we chose for evaluating the meta-models performance is the correlation coefficient $R^2$ (eq. 13), classically used in statistic:

$$R^2 = 1 - \frac{\sum_{i=1}^{n}(\hat{y}_i - y_i)^2}{\sum_{i=1}^{n}(\hat{y}_i - \bar{y}_i)^2} \tag{13}$$





Where $\bar{y}_i$ is the mean model response.

Finally, a particular attention was devoted to the analysis of the distribution of the residuals (the difference between model results and meta-model predictions). Especially, the residual plots are some useful tools in evaluating and comparing meta-models predictions as they can help avoiding inadequate meta-models and help adjusting the meta-model for better results. In general, it is assumed that the simplest meta-model that produces random residuals is a good candidate for being a relatively precise and unbiased model. In this study, residuals $\varepsilon_i$ between model $y_i$ and meta-model $\hat{y}_i$ predictions are evaluated as follow:

$$\varepsilon_i = y_i - \hat{y}_i \qquad (14)$$

Two kind of analysis are possible, both performed in this study: (i) the analysis of the distribution of the residuals (according to (Jarque and Bera, 1980), residuals should be normally distributed with mean zero) and the analysis of the cumulative frequency distribution of the modulus of residuals (reported in the next section). This analysis provides an order of magnitude of the total uncertainty related to meta-model prediction, as discussed in the next sections.

## 3 Construction of the numerical data base of tsunamis generated by the Azores Gibraltar Fracture Zone (AGFZ)

### 3.1 The Azores Gibraltar Fracture Zone characteristics

In order to assess the tsunami hazard induced by the *AGFZ* through numerical simulations, the seismic characteristics of the *AGFZ* must be previously defined. With this aim, a preliminary research was conducted in the framework of the ANR-TANDEM research project (Antoshchenkova et al., 2016). The variation ranges of the seismic source input parameters issued from this study are reported in Table 5 (in the column "*Design database*"). The idea was to explore as largely as possible the potential tsunami earthquakes from *AGZF* integrating all supposed possible sources of the 1755 Lisbon event as well as earthquakes data coming from the following earthquakes catalogues: http://www.isc.ac.uk; http://earthquake.usgs.gov; http://www.globalcmt.org; http://www.bo.ingv.it/RCMT/searchRCMT.html. The earthquake magnitude distribution associated with these tsunami scenarios varied in the range of [6.8; 9.3]. It must be noted that, considering the random and independent choice of the seismic source input parameters reported in Table 5, this preliminary database is composed of numerous tsunamis scenarios that are not likely from the geological or seismological point of view. Especially, the authors explored extreme faults depths, reaching 180 km. Finally, the database contains nearly 20,000 tsunami scenarios. The considered AGFZ zone extends from 34° to 40°N and from 18 to 7°W, encompassing to the East the southern part of Portugal down to Morocco, and reaching the oceanic sea-floor west of the Madeira tore rise (as reported in Figure 3).





### 3.2 Numerical tool

Tsunami numerical simulations were performed by coupling the CEA tsunami code with the IRSN Promethee bench in a preliminary research activity (Antoshchenkova et al., 2016). Promethee is an environment for parametric computation that allows carrying out uncertaintiy quantification studies, when coupled (or warped) to a code. This software

is freely distributed by IRSN (http://promethee.irsn.org/doku.php) and allows the parameterization with any numerical code and is optimized for intensive computing. Moreover, statistical post-treatment, such as UA and GSA can be performed using Promethee as it integrates R statistical computing environment (R Core Team, 2016).

CEA's tsunami code exploits two models, one for tsunami initialization and the other one for tsunami propagation. The initial seabed deformation caused by an earthquake is generated with the Okada model (Okada, 1985) and is transmitted

instantaneously to the surface of the water. This analytical model satisfies the expression of the seismic moment $M_o$:

$$M_O = \mu \cdot D \cdot L \cdot W \qquad (15)$$

where μ denotes the shear modulus, D [m] the average slip along the rupture of length L [m] and width W [m]. Then, the seismic magnitude "$M_w$" is directly computed through equation 16, as follow:

$$M_W = \frac{2}{3} \cdot \log_{10}(M_O) - 6{,}07 \qquad (16)$$

The following parameters are also required for tsunami initiation: longitude, latitude, and depth [km] of the center of the source, strike [degrees], dip [degrees], and rake [degrees]. A conceptual scheme of the input parameters for the tsunami-code

is reported in Figure 2. Then, the shallow water equations are used to propagate the tsunami towards the coast. Shallow water equations are discretized using a finite-differences method in space and time (FDTD). Pressure and velocity fields are evaluated on uniform separate grids according to Arakawa's C-grid (Arakawa, 1972). Partial derivatives are approximated using upwind finite-differences (Mader, 2004). Time integration is performed using the iterated Crank-Nicholson scheme. No viscosity terms are taken in account in our simulations. The only parameters of this second model are the bathymetry

(space step and depth resolution) and the time step. The tsunami-code was largely validated through extensive benchmarks in the framework of the work package 1 of the TANDEM research project (Violeau and TANDEM, 2016). In this work (Antoshchenkova et al., 2016), all simulations were performed on the same bathymetric grid with a space resolution of 2' (~3.6 km). It must be noted that the space step parameter could influence the numerical model results. For this reason, a sensitivity study to the grid-resolution parameter based on more than 5000 tsunami scenarios was performed by CEA in a

previous study, using a grid-resolution varying from 1' to 14' (Imbert et al., 2015). This preliminary research showed that the model parameters influencing tsunami height at the selected gauges (located off the coast) are the seismic source parameters and that bathymetrical resolution is not relevant, at least off the French Atlantic Coast. As a consequence, we consider that this grid is adapted to the objectives of the study even if it does not permit to correctly estimate the tsunamis run-up and the inundation areas.





### 3.3 Design and validation of meta-models of the French Atlantic Gauges

#### 3.3.1 Meta-models design

Even if *LHS* design is the most employed for the construction of Gaussian meta-models such as kriging, in this work we used as *design database* the numerical results issued from the preliminary research (Antoshchenkova et al., 2016), for many reasons. First, the variation range of the seismic source input-parameters (Table 5) is very large and permits to cover the bibliography analysis associated to the Lisbon tsunami (Antoshchenkova et al., 2016;Imbert et al., 2015) and to the AGFZ earthquakes registered in earthquakes catalogues. Thus, if correctly estimated, meta-models will be able to reproduce the model behaviour for a large range of variations of the seismic inputs parameters, which is consistent with the objectives of the study (to simulate a wide range of possible tsunami scenarios). Second, except for some numerical tests reported in previous studies (Iooss et al., 2010), at the moment, no theoretical results give the type of initial design leading to the best fitted meta-model in terms of meta-model predictions. Finally, in this way, meta-models can be built directly, without any complementary (and costly) numerical runs. In this sense, another *design database* as *LHS* will be employed only if the meta-models accuracy is not judged satisfactory, according to the test-criteria presented in the previous paragraph and used for the *training* and *validation* sets.

In order to build the *design database*, fault parameters as defined in section *3.1* were sampled randomly and independently, using uniform distributions (Antoshchenkova et al., 2016). This *design database* is a matrix which associates to a given combination of fault parameters the maximum simulated water height at each point of the numerical grid and also at some selected given locations called *gauges*. The maximum tsunami water height is the relevant parameter when estimating tsunami hazard and it is more synthetic than the water height at a specific time which allows reducing the amount of output data. The original database was previously filtered in order to exclude the faults deeper than 60 km, which is not realistic for the study site. The resulting *design database* contains 5839 scenarios split in a *training set* of 4672 scenarios and a *testing set* of 1167 tsunamis scenarios, used for the meta-model validation and the residual analysis. The water height characteristics associated to these tsunamis scenarios are reported in Table 3.

Finally, the meta-models are built for four French Atlantic Gauges (see Figure 3 and Table 2), using the open-source R-packages DiceKriging (Roustant et al., 2012), according to the equations 21, 22 and 23 reported in the APPENDIX 1. These gauges are the most representative of the maximum tsunami height along the French Atlantic Coast (Antoshchenkova et al., 2016;Imbert et al., 2015). Each meta-model is a function able to compute the maximum tsunami water height at the gauge location for a given set of seismic source parameters (strike, length, dip, rake, width, slip, longitude, latitude and depth). Obviously, the input parameters should be included in the parameter range used for the meta-model construction and reported in Table 3.



### 3.3.2 Meta-models evaluation

As reported in the section 2.4.2, the *training set* is split into $K$ folds of approximately equal size and a model is fitted from the data and validated on the fold $A_k$. This procedure was automatically managed by R-package sensitivity for the four Atlantic Gauges and the results in terms of a criterion of quality $L$ are reported in Table 4. The mean *MSE* computed is

very low (varying from a minimum value of 0.004 for Saint Malo to a maximum value of 0.029 for La Rochelle), indicating that the kriging meta-model is a good emulator choice for reproducing the CEA-tsunami code behaviour. Finally, for each gauge, we constructed a kriging meta-model using all the scenarios of the *training-set* data base.

Further, the residual analysis was performed by comparing the model results with the meta-models predictions for the *validation set* parameters. The objectives of this analysis are (i) to test the ability of the selected meta-model at

reproducing model results for another set of parameters (outside of the training data-set space), by the computation of the correlation coefficient and by the analysis of the residuals distribution (eq. 14), and (ii) to estimate the uncertainty associated to the use of a meta-model instead of the original model.

The computation of the correlation coefficient shows a good agreement between the original model results and meta-models, for the four selected gauges (see Table 4 and Figure 5). $R^2$-values are generally high (the minor values for La

Rochelle, of 76%, is satisfying), even if we can observe that the strongest water height from the original model are slightly underestimated by the meta-models for the *validation-set* data base (Figure 5). Considering the strong differences (Table 3) between the *"maximum"* water height and the *"$\mu_m + 2\sigma_m$"* water height of the *design database* ($\mu_m$ and $\sigma_m$ being, respectively, the mean and the standard deviation of the tsunami water height), which indicate that the lower water height are largely represented in the *design database*, it is not surprising that the meta-models predictions are more accurate for lower

tsunami height than for extreme water height. These results are confirmed by the computation of the frequency distribution of the absolute residuals (the modulus of Eq. 14) and of the density frequency distribution of the relative residuals (Eq. 14), as reported in Table 4 and Figure 4. This analysis shows that the residuals are normally distributed with mean zero (*"$\mu_r$"*) and lower standard deviation (*"$\sigma_r$"*), which is satisfying according to (Jarque and Bera, 1980), and that the maximum residuals are high for the fourth gauges (Table 4), probably for the reasons reported above.

In conclusion, the residuals analysis confirms that, globally, the meta-models constructed for the four French Atlantic gauges are able to reproduce the model behaviour (the residuals are globally low with the *"$2\sigma_r$"* of few centimetres), and they can be used for the construction of the numerical tsunami database. Nevertheless, we consider that when meta-models are used for the predictions of the tsunamis height instead of the original model, an uncertainty must be considered and applied to the meta-modelled water heights. According to results reported in Table 4 and Figure 5, an

uncertainty equal to *"+/-$2\sigma_r$"* calculated with the *validation set* should be conservative, considering the global residual trend (Table 4, Figure 5).





### 3.4 Building the tsunamis database for the French Atlantic Gauges

Starting from the validated meta-models we finally build a database of tsunamis generated by the AGFZ at four French Atlantic Gauges. Because the modelled area encompass different seismotectonic domains, the overall AGFZ area was split into two main seismic source zones in order to feed the meta-model database. This was done to represent an

oceanic domain to the West where normal to transtensive earthquakes mainly occur within a thin oceanic crust and a continental domain to the East where reverse to transpressive earthquakes mainly occur on a thickened oceanic to continental crust (Buforn et al., 1988;Molinari and Morelli, 2011;Cunha et al., 2012). As a first order analysis, we considered the oceanic domain (hereafter called *oceanic shelf*) west of the 10°W meridian and the continental domain (hereafter called *continental shelf*) east of this meridian, coinciding roughly with the base of the continental slope facing the Portuguese

coastline.

Such a simplified zonation represents a first order approach of the AGFZ seismotectonics, consistent with the global objective of the study being the analysis of the tsunamigenic potential of a seismic zone through "uncertainty quantification techniques". Indeed, a more accurate consideration of both zonations and known faults of the *AGFZ* would be more in line with the work conducted by Roshan at al. (2016) or with more classical deterministic THA. The main fault characteristics

considered to perform earthquake scenarios in both continental and oceanic domains are reported in Table 5, representing our interpretation of data contained in the above mentioned papers. The maximum oceanic depth is considered to be of the order of 10 km whereas the continental crust can attain 30 km depth. The fault parameters are considered uniformly distributed and are randomly sampled in their range of variation. We finally filtered the resulting database according to an aspect-ratio criterion, allowing the ratio between length and the width of the faults not to exceed the value of 10, which

correspond to an upper bound of what is observed in nature (Mc Calpin, 2009). The final database contains nearly 50,000 tsunami scenarios, resulting of earthquake magnitudes varying from 6.7 to 9.3 (Table 5), which is consistent with the magnitude range of the *design database*. This resulting *global database* (Table 5) is composed by the union of the *oceanic* and the *continental* databases scenarios. The tsunamis height characteristics associated to the four gauges are reported in Table 6. It must be noted that the maximum tsunamis height of the *global database* are lower than tsunamis height of the

*design database* and that the convergence of statistics (mean water height) is largely achieved, as reported in Figure 6.

In conclusion, the built tsunamis database (the *global database*) should represent most of the possible tsunamis generated by the *AGFZ* (the *oceanic* and the *continental shelf*). It is analysed in the next section with a focus on THA.

In addition, for the critical analysis of this global numerical database obtained through uncertainty quantification, we decided to complement our study with two specific and nearly deterministic scenarios recognized as possible sources of

the Lisbonne 1755 earthquake, namely the Gorringe and Horseshoe structures (Buforn et al., 1988;Stich et al., 2007;Cunha et al., 2012;Duarte et al., 2013;Grevemeyer et al., 2017). Both structures were modelled taking into account available maps (Cunha et al., 2012;Duarte et al., 2013) and fault parameters (Stich et al., 2007;Grevemeyer et al., 2017) summarized in Table 5. These scenarios have been chosen because they are not included in the proposed zonation of the *global database*,





considering their location (Figure 3). Similarly, fault parameters are considered uniformly distributed and are randomly sampled in their range of variation (Table 5), for a total of nearly 10,000 tsunami scenarios. Results from these nearly deterministic scenarios (Table 6) will be utilized in order to complement the critical analysis of the results obtained from the global database.

## 4 Analysis of the numerical tsunamis database

### 4.1 Results from sensitivity analysis: the influence of the seismic-source parameters

Homma and Saltelli (Homma and Saltelli, 1996) introduced the total sensitivity index which measures the influence of a variable jointly with all its interactions. If the total sensitivity index of a variable is zero, this variable can be removed because neither the variable nor its interactions at any order have an influence on the results. This statistical index (called Sobol index "ST" in this paper, Eq. 9), is here of particular interest in order to highlight the earthquake source parameters that mostly control the tsunamis height at each tested gauge. In Figure 7, we reported the total Sobol index for the four meta-models of the French Atalntic Gauges computed with the methodology proposed by Janssen (Jansen et al., 1994) using the sensitivity-package R (Pujol et al., 2016). Results show that the slip parameter is globally the most-influencing parameter for all the French Atlantic Gauges meta-models, which is quite obvious considering that the fault-slip directly conditions the ocean floor deformation and hence the tsunami amplitude. However, Figure 7 suggests that the most influencing parameters for the four gauges are slightly different, depending on the gauge location. One can differentiate results obtained for the southern gauges (i.e. La Rochelle & Gastes) from those obtained at northern gauges (i.e. Saint Malo & Brest):

- For the southern gauges, *width* is the second most relevant parameter, especially at La Rochelle where it is almost as important as the *slip* parameter. Other important parameters are *length* and *rake,* suggesting that for these gauges, fault source parameters in terms of magnitudes (depending on *width*, *slip* and *length* according to equations 15 and 16) and kinematics are the most important in generating hazard;

- For the northern gauges, apart from *slip*, *strike*, *width* and *rake* are also important, but slightly less than the *longitude* parameter. This means that the location of the source is here of major importance. A possible physical reason could be associated to the lack of the natural barrier composed by the north of Spain and Portugal which protects southern gauges from AGFZ related tsunamis compared to northern gauges. As a consequence, the northern gauges should be more exposed to hazard in comparison to southern gauges.

### 4.2 Results from Uncertainty Quantification

As reported in the introduction, the objective of this work is to evaluate the benefits of uncertainty quantification for the improvement of the THA. The tsunami height frequency distribution of each considered tsunamigenic source is reported and compared in Figure 8 for each gauge. It can be observed that the tsunamis heights generated by the *oceanic shelf* are





globally lower than the tsunamis generated by the other sources. This effect seems more marked for the *southern gauges,* which is coherent with the analysis performed with the Sobol index. Moreover, the strongest tsunamis of the global database are globally generated by the *continental shelf* for both the *southern* and the *northern gauges*.

From an engineering point of view, the tsunami hazard can be directly deducted from Figure 8 (black line). (Roshan

et al., 2016) proposed to use the mean tsunami height from their distribution as hazard level. However, we propose in this study to set the tsunami hazard level at the value "$\mu_m+2\sigma_m$" of the modelled water height for each gauge and to consider an uncertainty associated to this level equal to the $+/-2\sigma_r$ residual (summarized in Table 7 for each gauge). Our arbitrary choice may be justified, firstly, by considering the shape of the density frequency distribution from UQ reported in Figure 8 (black line). In fact, the density frequency distributions for each gauge is centred around the lower tsunamis heights while the

maximum modelled tsunamis heights are very large with respect to the tsunamis statistical parameters of the distribution (reported in Table 7). As a consequence, this choice appears numerically reasonable in a context of THA as it considers both the low tsunamis height and the maximum modelled tsunamis height. Secondly, considering the roughly approximation of the earthquake input parameters (and the consequently wide range of seismic magnitude distribution) used for the construction of the global database, this hazard level appears more conservative in a context of THA.

The robustness of this choice is analysed in the next sections. Especially, it is of interest to analyse the strongest tsunamis of the numerical database (see section 4.3) and to evaluate if this hazard level is robust with respect to results which could be obtained with a more classical scenario-based approach (see section 4.4).

### 4.3 Focus on the strongest tsunamis of the numerical database

As reported in Table 7, there are strong differences between the maximum simulated tsunami height and the "$\mu_m +$

$2\sigma_m$" water height of the out-put frequency distribution chosen as hazard level in this study, for all the gauges of the numerical database. As a consequence, from a methodological point of view, and with the objective of assessing the tsunami hazard level from the uncertainty quantification study, it is of interest to analyse these extreme tsunami scenarios. The objective of this analysis is to answer to the central question *"which confidence level should be retained for assessing the tsunami hazard from uncertainty quantification and why?"*

With this aim, we first analysed the magnitude frequency distribution "$M_w$" of the tsunami scenarios of the global database (grey curve in Figure 9a) with respect to the magnitude of the earthquakes generating tsunamis higher than the proposed *hazard level* for each gauge (coloured lines in Figure 9a). The objective of this analysis is to understand to which magnitude are associated the strongest tsunami heights. This is a global analysis based on the global tsunami database. It must be noted that the sample size for each gauge is not the same and that the tsunamis from the oceanic shelf are under-

represented considering that the maximum magnitude associated to these seismic scenarios is around 8.5 (see Table 5).

Globally, the strongest tsunami heights correspond to the strongest seismic events of the database (Figure 9a). Especially, more than 80% of the tsunami heights higher than the chosen *hazard level* ("$\mu_m + 2\sigma_m$") are associated to a



seismic magnitude higher than *8.5*. These tsunamis events completely represent the tail of the earthquake magnitude distribution (the grey histogram in the Figure 9a). Nevertheless, some discrepancies are observed between the results of *the southern* gauges (*La Rochelle* and *Gastes*) and the results obtained for the *northern gauges* (*Brest* and *Saint-Malo*), as for the total Sobol index computation. In fact, even if it seems that only the very strong earthquakes (magnitude higher than *8.5*) can generate a tsunamis height higher than "$\mu_m + 2\sigma_m$" for the middle-south of Atlantic French coast (more than the *95%* of the tsunamis scenarios of the sample), this is more ambiguous for the *northern Gauges.* In this case, nearly *30%* of the tsunamis higher than the *hazard level* can be generated by an earthquake of magnitude in the range [7.8 – 8.5], which includes the oceanic shelf.

The distribution of the tsunamis height higher than the proposed *hazard level* ("$\mu_m + 2\sigma_m$") and in the magnitude range [7.8 – 8.5] is also analysed in Figure 9b. It results that the contribution of the oceanic shelf to the strongest tsunamis in this range of magnitude is not negligible, in frequency (nearly 50% of the simulated tsunamis with a height higher that "$\mu_m + 2\sigma_m$"), with respect to the amount of the strongest tsunamis. Especially, the differences between northern and southern gauges are very marked, in this range of magnitude. In fact, the *northern gauges* (*Saint-Malo* and *Brest*) are practically the only ones exposed to tsunamis generated by the oceanic shelf.

In conclusion, these results also suggest that, beyond earthquake magnitudes, the position and the orientation of the faults are influents parameters, in agreement with the analysis of the previous paragraph (see 4.2). Thus, for a given magnitude, a tsunami generated by a "*well located-oriented*" fault would be potentially more hazardous, at least for the northern of the French Atlantic Coast.

In terms of *hazard level*, which is the original objective of this analysis, one can consider, at first, that tsunamis higher than the hazard level are mainly generated by seismic magnitudes stronger than *8.5*, even if some rare combinations of tsunami input parameters can lead to higher tsunamis height.

**4.4 Place of a Lisbon-like 1755 tsunami scenario in the numerical database**

As reported in the introduction, the deterministic THA approach offers simple, directly relatable events or scenarios describing different possible characteristics of the event. The tsunami hazard is assessed by means of considering particular source scenarios (usually maximum credible scenario) and the maximum tsunami height is generally retained as hazard level. A more sophisticated method was recently proposed by (Roshan et al., 2016). The authors tested around 300 tsunamis scenarios in a magnitude range [8 – 9.5] associated to various faults potentially impacting the Indian coast. Finally, the authors suggested that an appropriate water level for hazard assessment (e.g. mean value or mean plus sigma value) should be retained. They proposed the mean value of the simulated water heights, as test, by considering that this value may need to be revisited in future. In the case of the tsunamis hazard associated to the *AGFZ*, the 1755 Lisbon tsunami is the classical reference scenario. As a consequence, in order to validate our methodology for the AGFZ, and considering also the results presented in section 4.3, it is of interest to quantify the relevance of the major tsunami event of this zone with respect to the



numerical database constructed through uncertainty quantification. The objective of this analysis is to understand if the tsunamis height generated by a Lisbon-like earthquake are included, or not, in the range of the "$\mu_m + 2\sigma_m$" water height of the *global database*.

We recall here that despite the numerous investigations dedicated to this event, there are still a lot of uncertainties about its source. The tsunami scenarios potentially associated to a Lisbon-like tsunami supposed to be generated by the *Gorringe bank* and the *Horseshoe bank*, which are the most-likely source generating the Lisbon tsunami of 1755 according to the hypothesis reported in (Duarte et al., 2013) or (Grevemeyer et al., 2017) and summarized in Table 5. These two scenarios were chosen in order to evaluate the global tsunami database, which was designed by roughly zoning continental and oceanic shelves. In fact, the tsunami hazard induced by the Horseshoe scenario, located at the limit between these two zones of the global database, may be partly contained in the continental contribution of the database. On the contrary, the Gorringe scenario represents a pure "Continental" scenario within the Oceanic zone, hence excluded by the global database. The magnitude range associated to these tsunamis scenarios varies from 7.7 to 8.9 and the hazard level associated to these faults corresponds to the "$\mu_m$" modelled water height. This hazard level for the scenario-based approach was arbitrarily chosen, considering that the philosophy of this kind of approach is to take into account only the worst possible scenarios. In this sense, according to the previous work from Roshan et al. (2016), we also proposed to take into account the "mean" modelled water height.

In Table 7 we summarized the tsunami hazard height (the modelled "$\mu_m + 2\sigma_m$" water height) of the global database for each gauge with the associated uncertainty (corresponding to the residuals +/- $2\sigma_r$). Similarly, the hazard level was computed for the scenarios corresponding to the Gorringe faults and the Horseshoe faults, considering the modelled "$\mu_m$" water height. The results of this analysis indicate that the strongest tsunamis are globally generated by the Gorringe bank. Nevertheless, it also must be noted that the proposed *hazard level* from the total database is representative of the scenarios based tsunamis once the uncertainty associated to meta-model predictions is considered (see Figure 10). In conclusion, from this analysis, the proposed hazard level resulting from uncertainty quantification seems reasonable in light of the results obtained through a more scenario-based approach, at least for the tsunamis generated off the French Atlantic coast.

## 5 Conclusions and perspectives

The research presented in this paper, mainly methodological, was performed in order to test the interest of using an uncertainty quantification approach for the assessment of the tsunami hazard generated by earthquakes. The methodology is based on the use of kriging meta-models built following numerical simulations performed in a precedent research activity (Antoshchenkova et al., 2016). The meta-models were tested and validated through a series of statistical indicators (correlation coefficient, residuals analysis, MSE). These performance-tests allowed us to conclude that, even if the maximum residuals (differences between model and meta-model results) are very high for the four gauges (see Table 4), the meta-models are able to reproduce the model behaviour and they can be used for the construction of the numerical tsunami





database. Nevertheless, when meta-models are used instead of the original model for the predictions of the tsunamis height, we propose to add an uncertainty corresponding to the *"+/-2σ_r"* residual value of each gauge.

All the results and the analysis presented in section 4 are directly related to the methodology proposed. At first, GSA performed with Jansen's method permits to underline that the seismic Magnitude generating the tsunamis in the AGFZ (depending on width, slip and length) is more relevant depending on the location of the gauge (the impacted zone). For instance, only the earthquakes inducing large displacements of the sea-floor (large *slip*) are potentially capable to affect the middle/south of France. However, concerning the gauges located to the north of the French Atlantic Coast (*Brest* and *Saint-Malo*), other influencing parameters are, apart from the slip, the location (*longitude*) and the orientation (*strike* and *rake* parameters) of the fault sources. These results are confirmed by the analysis of the strongest tsunamis (higher than the proposed *hazard level*) of the numerical data base which permits to conclude that, globally, the position and the orientation of the fault are influent parameters. In other words, for a given magnitude, a tsunami generated by a "*well located-oriented*" fault would be potentially more hazardous, at least for the sites located along the northern French Atlantic Coast. In this sense, the UQ study permitted to explore some rare combinations of seismic inputs parameters generating higher tsunamis level (see Figure 9b). Moreover, the UQ permits to underline that globally the tsunamis generated by the *oceanic shelf* are largely less impacting than the tsunamis generated by the *coastal shelf*, on average.

In terms of THA, this research is innovative with respect to the classical scenario-based approach reported in the literature (JSCE, 2002;Lynett et al., 2016), which are strongly limited by the number of tsunamis scenarios they can consider and do not allow to explore, in a rigorous and complete way, the tsunamigenic potential of a seismic zone. Our approach proposes a completely new philosophy for THA and can be considered as an extension of the previous innovative work from (Roshan et al., 2016), which first improves the classical deterministic approach by focusing its analysis on some selected earthquake source parameters. The numerical database (of nearly 50,000 tsunamis scenarios) allowed exploring the numerous uncertainties related to the characteristics of the *AGFZ* and most of the possible tsunamis scenarios should be represented. Moreover, this methodology of exploration of the tsunamigenic potential also permits to show some "unexpected" result: for instance, the strong oceanic scenarios impacting the northern gauges are not associated to the stronger seismic magnitudes; such result may not be identifiable through a classical deterministic approach.

Considering the rough zonation of the *AGFZ* and the shape of the magnitude frequency distribution of the tsunami heights, we proposed to attribute to each French Gauge a tsunamis hazard equal to the modelled tsunami height "$\mu_m+2\sigma_m$" with an associated uncertainty equal to the residual "$+/-2\sigma_r$". This criterion was tested through a comparison with the tsunamis exceeding this hazard level (paragraph 4.3) and with the tsunamis generated by a Lisbon-like 1755 earthquake (paragraph 4.4). The latter representing a comparison between the global results from UQ and a more classical scenario-based approach. Globally, the *hazard level* issued by the total database seems robust and representative of the tsunamis from each single sub-zone once the uncertainty of numerical results is considered (see Figure 10), at least for French gauges off the Atlantic coast. However, it must be noted that the choice of the hazard level is arbitrary and that further studies are necessary to validate this value. Moreover, the *global database* proposed in this study was built using the Monte-Carlo



method and by considering that the seismic source input parameters are independent and uniformly distributed. Thus, an improvement of the methodology should consist in taking into account the dependencies between the seismic source parameters in the UQ and GSA steps.

    In conclusion, as reported in the introduction, the aim of this study was to illustrate the benefits of a methodology
5  through a case study, which is not an operational assessment of tsunami hazard along the French Atlantic Coast. In fact, tsunami hazard is analysed only off the French Atlantic Coast and by using, as input data for meta-model's construction, the deterministic simulations performed with a rough computation grid-resolution of 2'' degrees (~3.6 km) are not adapted to reproduce the influence of bathymetry on tsunami propagation in the vicinity of the coastal areas. As a consequence, for a real THA along the French Atlantic Coast, it should be necessary, at first, to improve the actual numerical model in order to
10  better represent the tsunamis run-up and the inundated areas, with a more accurate bathymetric grid. Moreover, it appears of interest to better test the *hazard level* in other locations where historical data from the 1755 Lisbone-tsunami are available, according to the data available in literature (Santos and Koshimura, 2015;Baptista et al., 1998).





**Acknowledgments**

This study was performed through a cooperative research between the Institut for Radioprotection and Nuclear Safety (IRSN) and the CEA, in the framework of the French ANR-TANDEM research project. The authors want to thank ENS Paris and particularly the Geology Laboratory for their technical support on this project. We acknowledge especially Eric Calais
5    for the availability of a part of team's cluster and Pierpaolo Dubernet for the help he offers to install *Promethee*. The present work also benefits from the inputs of Miss Hend Jebali and Dr. David Imbert and from the valuable technical assistance of Dr. Maria Lancieri, Dr. Yann Richet and Miss Ludmila Provost, who provided valuable technical assistance.





## APPENDIX 1 – UNIVERSAL KRIGING EQUATIONS

Originally coming from geosciences (Krige, 1951) and having become the starting point of geostatistics (Matheron, 1963), Kriging is basically a spatial interpolation method. However, starting from previous work of Sacks et al (Sacks et al., 1989), many works dedicated to Kriging as a surrogate to computer experiments have been published (Welch et al.,

1992;Koehler and Owen, 1996;O'Hagan, 2006). As a consequence, useful tools have been developed to build-up kriging meta-models in a single point for multi-variables complex problems. The theoretical background for kriging-metamodel design used in this study is fully detailed in (Roustant et al., 2012), which descript the R-packages DiceKring and DiceOptim that we used for meta-models construction. We limited here to report the basic hypothesis of the methodology for the interested reader, directly issued from (Roustant et al., 2012).

Let-us assume that the model response $y$ is assumed to be a deterministic real-valued function of the $d$-dimensional variable $x = (x_1,..., x_d) \in D \subset R^d$. $y$ is assumed to be one realization of a square-integrable random field $(Y(x))_{x \in D}$ with first and second moments known up to some parameters. Let us consider that $X = \{x^{(1)},..., x^{(n)}\}$ denotes the points where $y$ has already been evaluated, and denote $y = y(x^{(1)}, ..., x^{(n)})^T$ the corresponding outputs. For any $x \in D$, the aim of Kriging will be to optimally predict $Y_x$ by a linear combination of the observations $y$. Especially, kriging-modelling treats the

deterministic response $y(x)$ as a realization of a random function $Y(x)$, including a regression part and a ceneterd stochastic process:

$$Y(x) = f(x) + Z(x) \tag{17}$$

Where the deterministic function $f(x)$ provides the mean approximation of the computer code.

For Universal Kriging (UK) employed in this study, $Y(x)$ is assumed to be the sum of a known deterministic trend function $\mu : x \in D \rightarrow \mu(x) \in R$ and of a ceneterd square-integrable process $Z$:

$$Y(x) = \mu(x) + Z(x) \tag{18}$$

$$\mu(x) = \sum_{j=1}^{p} \beta_j f_j(x) \ (p \in N - \{0\}) \tag{19}$$

Where $f_j$ are fixed basis functions, $\beta_j$ are unknown real coefficients and $x$ is the d-dimensional variable. UK consists in deriving best linear predictions of $Y$ based on the observations $Y(X)$ while estimating the vector $\beta = (\beta_1,..., \beta_p)^T$. It must be noted that the stochastic part $Z(x)$ is a Gaussian centered process fully characterized by its covariance function. $Z$'s convariance kernel $C : (u,v) \in D^2 \rightarrow C(u,v) \in R$ is known, considering that $X = \{x^{(1)},..., x^{(n)}\}$ is known. By assuming $Z(x)$ second order stationary, the covariance can be written as $C(u,v) = \sigma^2 R(u - v; \theta)$ where the so-called correlation



function $R$ is a function of positive type with parameters $\theta$, and $\sigma^2$ is a scale parameter called the process variance. The choice of the covariance kernel $C$ is crucial for the construction of the kriging metamodel, all the more so when the trend is known or assumed constant (Roustant et al., 2012). Some of the most popular 1-dimensional stationary kernels include the Gaussian kernel, Fourier transform of the Gaussian density, as well as the Matern kernel, Fourier transform of the Student

5   density (Stein, 1999). The covariance kernels available in the version of DiceKriging used in this study (Roustant et al., 2012) can be written as follow:

$$c(h): C(u,v) = \sigma^2 \prod_{j=1}^{d} g(h_j; \theta_j) \qquad (20)$$

Where $h = (h_1, ..., h_d) := u - v$, and $g$ is a 1-dimensional covariance kernel. The parameters $\theta_j$, also called *characteristic length-scale* according to (Rasmussen and Williams, 2006), are chosen to be physically interpretable in the same unit as the corresponding variables (Roustant et al., 2012). The package proposes five equations for the covariance kernels function g(h)

10  (Gaussian, Matérn 5/2, Matérn 3/2, Exponential and Power-Exponential), according the analytical formula given by (Rasmussen and Williams, 2006). For the meta-models of this study we chosen the Matérn 5/2 expression for g(h):

$$g(h) = \left(1 + \frac{\sqrt{5}|h|}{\theta} + \frac{5h^2}{3\theta^2}\right) \exp\left(-\frac{\sqrt{5}|h|}{\theta}\right) \qquad (21)$$

Now, let us recall that the best linear unbiased predictor of *Y(x)* based on the observations *Y(X)* is obtained by finding $\lambda^*(x) \in R^n$ minimizing the mean squared error $MSE(x) := E\left[\left(Y(x) - \lambda(x)^T Y(X)\right)^2\right]$. If C is invertible, the strict convexity of MSE ensures both the existence and the uniqueness of $\lambda^*(x) \in R^n$, and the UK weights are given by

15  $\lambda^*(x) = C^{-1} c(x)$, where $C = \left(C(x^{(i)}, x^{(j)})\right)_{1 \le i, j \le n}$ is the covariance matrix of *Y(X)*, and $c(x) = \left(C(x, x^{(i)})\right)_{1 \le i \le n}$ is the vector of covariance between *Y(x)* and *Y(X)*. Substituting both the random vector *Y(X)* by its realization *y* and $\lambda(x)$ by $\lambda^*(x)$ in the MSE equation, we get the so-called UK-mean prediction:

$$m_{UK}(x) = f(x)^T \hat{\beta} + c(x)^T C^{-1}(y - F\hat{\beta}) \qquad (22)$$

Similarly, by plugging in the optimal $\lambda^*(x)$ in the expression of the MSE, one gets the so-called variance at *x*:

$$s_{UK}^2(x) = C(x, x) - c(x)^T C^{-1} c(x) + \left(f(x)^T - c(x)^T C^{-1} F\right)^T \left(F^T C^{-1} F\right)^{-1}\left(f(x)^T - c(x)^T C^{-1} F\right) \qquad (23)$$





Where $f(x)$ is the so-called vector of trend functions values at x, $F = \left( f\left( x^{(1)},..., x^{(n)} \right) \right)^{T}$ is the $n \times p$ so-called experimental matrix, and the best linear estimator of $\beta$ under correlated residual is given by the usual formula

$$\hat{\beta} := \left( F^{T} C^{-1} F \right)^{-1} F^{T} C^{-1} y .$$

Basic properties of UK include similar interpolating behaviour, with a variance vanishing at the design of experiments.

5   Furthermore, $m_{UK}(x)$ tends to the best linear fit $f(x)^{T} \hat{\beta}$ whenever the covariances c(x) vanish. Note also other interesting properties associated to the kriging meta-model, according to (Tanioka and Satake, 1996): it is flexible to any kind of functional form of the original model, by introducing less restrictive assumptions on the functional form of the simulator than a polynomial model would imply and it provides a variance estimate of the prediction, according to Eq. 23.



**Tables**

| Step for meta-models design | Classical approach | Approach used in this study |
|---|---|---|
| **Initial design database** | Space-filling as LHS-design (McKay et al., 1979) | Filtered database from (Antoshchenkova et al., 2016) based on a Monte-Carlo sampling |
| **Splitting data/base in training set and validation set** | Splitting the initial data base (Amari et al., 1997): 1) Training set (80% of initial database scenarios) 2) Validation set (20% of initial database scenarios) | |
| **Construction** | Estimation of meta-model parameters according to equations reported in section 2.4.2. | |
| **Accuracy/Optimization** | k-fold cross validation (Hastie et al., 2002), (Kohavi, 1995) and (Breiman and Spector, 1992). | |
| **Residuals on the testing database** | Residual analysis and R2 respect to the validation set. | |

**Table 1:** Steps for meta-models construction and validation

| Gauge | Longitude [degrees] | Latitude [degrees] | Depth [m] |
|---|---|---|---|
| **Saint-Malo** | -2.08 | 48.7 | 8 |
| **Brest** | -4.65 | 48.26 | 28 |
| **La Rochelle** | -1.65 | 45.93 | 40 |
| **Gastes** | -1.27 | 44.35s | 15 |

**Table 2:** Location and water depth of the French Gauges chosen for meta-model construction

| | Saint Malo | Brest | La Rochelle | Gastes |
|---|---|---|---|---|
| **Min [m]** | 0.00 | 0.00 | 0.00 | 0.00 |
| **$\mu$ [m]** | 0.21 | 0.47 | 0.37 | 0.30 |
| **$\mu + \sigma$ [m]** | 0.39 | 0.93 | 0.71 | 0.60 |
| **$\mu + 2\sigma$ [m]]** | 0.57 | 1.38 | 1.05 | 0.90 |
| **Max [m]** | 1.60 | 5.66 | 3.12 | 3.07 |

10     **Table 3:** Maximum water height associated to the construction (or design) database (data from (Antoshchenkova et al., 2016)); $\mu$, $\sigma$ and Max correspond to the mean, standard-deviation and maximum modelled values.





| Gauges | Training set | Validation set | | | | |
|---|---|---|---|---|---|---|
| | L_CV [-] | R2 [%] | $\mu_r$ [m] | $+\sigma_r$ [m] | $+2\sigma_r$ [m] | Max [m] |
| Saint-Malo | 0.004 | 87% | 0.000 | 0.066 | 0.131 | 0.498 |
| Brest | 0.012 | 94% | 0.001 | 0.112 | 0.224 | 0.982 |
| La Rochelle | 0.029 | 76% | 0.001 | 0.170 | 0.339 | 1.323 |
| Gastes | 0.021 | 80% | 0.004 | 0.148 | 0.293 | 1.392 |

**Table 4:** Summary of meta-models evaluations with the testing and the training data base; $\mu_r$, $\sigma_r$ and $Max_r$ corresponds to the mean, standard-deviation and maximum residual respectively.

| Seismic source parameter* | Design database | Oceanic database | Continental database | Gorringe bank | Horseshoe bank | Global database* |
|---|---|---|---|---|---|---|
| Strike [degrees] | 0 – 360 | 0 – 360 | 0 – 360 | 40 – 70 | 40 – 70 | 0 – 360 |
| Length [km] | 40 – 400 | 40 – 227 | 40 – 600 | 180 – 200 | 160 – 200 | 40 – 600 |
| Dip [degrees] | 1 – 90 | 60 – 90 | 10 – 90 | 10 – 40 | 10 – 40 | 10 – 90 |
| Rake [degrees] | -180 – +180 | -180 – 0 | 0 – +180 | 70 – 110 | 70 – 110 | -180 – +180 |
| Width [km] | 20 – 200 | 10 – 23 | 10 – 310 | 16 – 224 | 16 – 228 | 10 – 310 |
| Slip [m] | 1 – 25 | 1 – 25 | 1 – 25 | 5 – 25 | 5 – 25 | 1 – 25 |
| Longitude [degrees] | -18 – -7 | -18 – -10 | -10 – -7 | -12 – -11 | -10.5 – -9.5 | -18 – -7 |
| Latitude [degrees] | 34 – 40 | 34 – 40 | 34 – 40 | 36.5 – 37.5 | 35.8 – 36.5 | 34 – 40 |
| Depth [km] | 2 – 60 | 5 – 10 | 5 – 30 | 10 – 40 | 10 – 40 | 5 – 40 |
| Magnitude range $M_w$ [-] | 6.7 – 9.3 | 6.7 – 8.3 | 6.8 – 9.3 | 7.7 – 8.9 | 7.7 – 8.9 | 6.7 – 9.3 |

**Table 5:** Summary of the variation range of the seismic source input parameters for the design, the oceanic, the continental database and for the tsunami scenarios associated to the Gorringe bank and the Horseshoe bank (hypothesis from (Duarte et al., 2013) and (Grevemeyer et al., 2017)). *The global database is composed by the scenarios simulated with the meta-models (more then 50,000 tsunamis scenarios) for the construction of the oceanic and the continental database. **Seismic
10 source parameters are assumed uniformly distributed and are randomly sampled for the construction of the Global database.

| | Saint Malo | Brest | La Rochelle | Gastes |
|---|---|---|---|---|
| $\mu_m$ [m] | 0.09 | 0.18 | 0.19 | 0.13 |
| $\mu_m + \sigma_m$ [m] | 0.17 | 0.34 | 0.36 | 0.27 |
| $\mu_m + 2\sigma_m$ [m] | 0.24 | 0.51 | 0.54 | 0.41 |
| $Max_m$ [m] | 0.77 | 2.01 | 1.81 | 1.92 |

**Table 6:** Maximum water height associated to global database; $\mu_m$, $\sigma_m$ and $Max_m$ correspond to the mean, standard-deviation and maximum modelled values.





|  | HorseShoe ($\mu_m$) [m] | Gorringe ($\mu_m$) [m] | Global Database ($\mu_m + 2\sigma_m$) [m] | Meta-model uncertainty ($+/-2\sigma_r$) [m] | Max$_m$ [m] |
|---|---|---|---|---|---|
| **Saint Malo** | 0.18 | 0.25 | 0.24 | +/- 0.13 | 0.77 |
| **Brest** | 0.31 | 0.53 | 0.51 | +/- 0.22 | 2.01 |
| **La Rochelle** | 0.26 | 0.32 | 0.54 | +/- 0.34 | 1.81 |
| **Gastes** | 0.18 | 0.29 | 0.41 | +/- 0.29 | 1.92 |

**Table 7:** Numerical tsunami database and scenarios-based hazard levels. For each gauge, the uncertainty corresponds to the $+/-2\sigma_r$ also reported in Table 4. The maximum modelled tsunamis height is also reported in the next column.



**Figures**

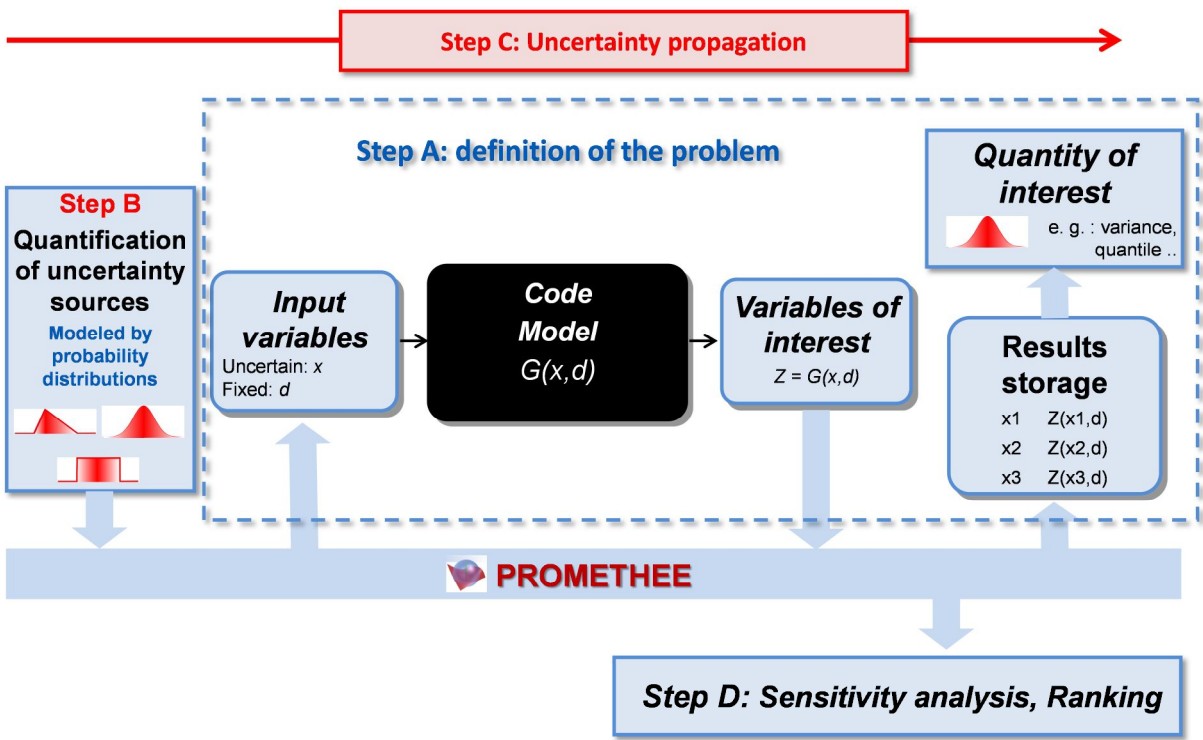

**Figure 1:** Uncertainty and Sensitivity analysis steps driven by Promethee Environment.



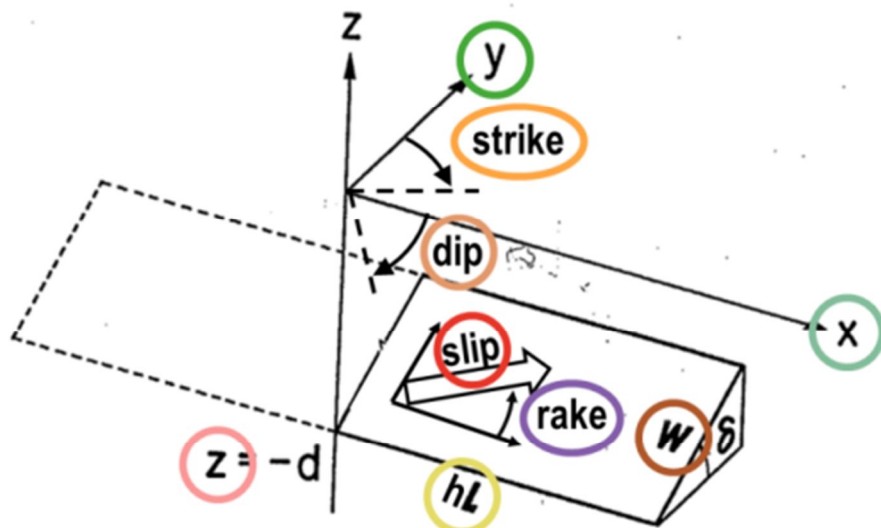

**Figure 2:** Geometrical input parameters of the tsunami-code employed in the study. $d$ [km] is the depth of the seismic source which is assumed to be at the middle of the fault, $hL$ [km] is the half fault length, $W$ [km] is the half fault width, $x$-$y$-$z$ are three-dimensional axes.



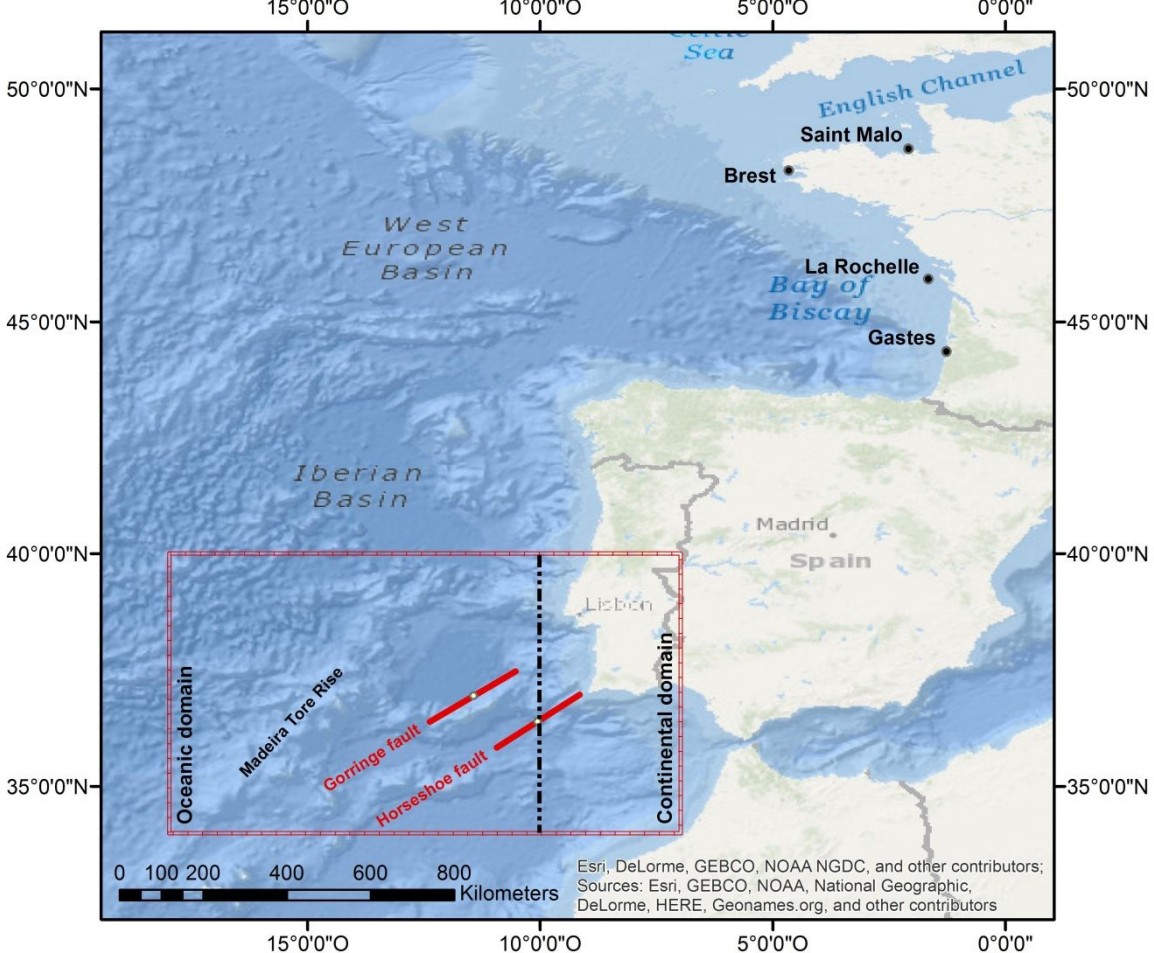

**Figure 3:** Bathymetry covering the computational domain. Red cross hatching show areas where the random source have been located for meta-models construction. The points represent the gauge locations selected for tsunami database construction along the French Atlantic Coast. Gorringe fault and Horseshoe fault are special structures at the boundary between the continental domain (on the right) and the oceanic domain (on the left).





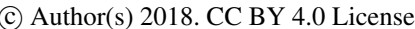

| % finer than | Saint Malo [m] | Brest [m] | La Rochelle [m] | Gastes [m] |
|---|---|---|---|---|
| 75% | 0.06 | 0.03 | 0.14 | 0.11 |
| 85% | 0.08 | 0.08 | 0.19 | 0.16 |
| 90% | 0.10 | 0.13 | 0.25 | 0.20 |
| 95% | 0.14 | 0.22 | 0.37 | 0.29 |
| 98% | 0.18 | 0.42 | 0.52 | 0.41 |
| 100% | 0.50 | 0.98 | 1.32 | 1.39 |

**Figure 4:** (a) Absolute residuals frequency distribution and (b) cumulate residual frequency distribution for French Gauges. The Table in the figure (a) summarize the absolute residuals associated to higher quantiles of the frequency distribution.





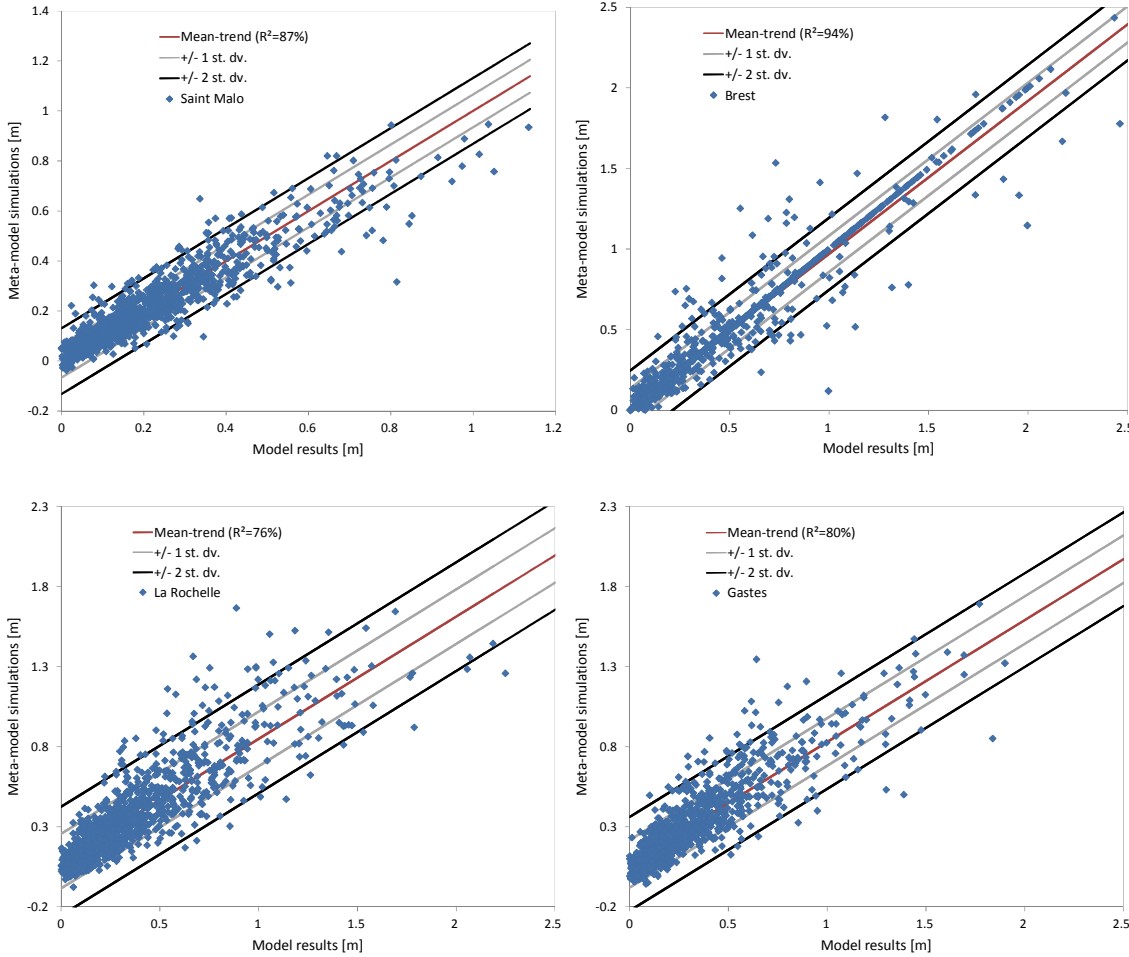

**Figure 5:** Comparison between the modelled and the meta-modelled maximum water height for the test database and the selected French Atlantic Gauges (blue points). The red line indicates the theoretical perfect correspondence between the original model and meta-model predications. Grey lines indicates the 1σ confidence interval for meta-models predictions and black lines the 2σ confidence interval for meta-models predictions.





**Figure 6:** Convergence study for the numerical tsunami database built through meta-models (only Brest and La Rochelle gauges are represented). The mobile mean of the simulated water height is stable after around two thousands simulations. The 95% confidence bounds for means correspond to "+/- $\left(\frac{2 \cdot \sigma_{m(N)}}{\sqrt{N}}\right)$" according to the central limit theorem and the chose

5       Monte-Carlo sampling method, where $\sigma_{m(N)}$ is the mobile standard deviation





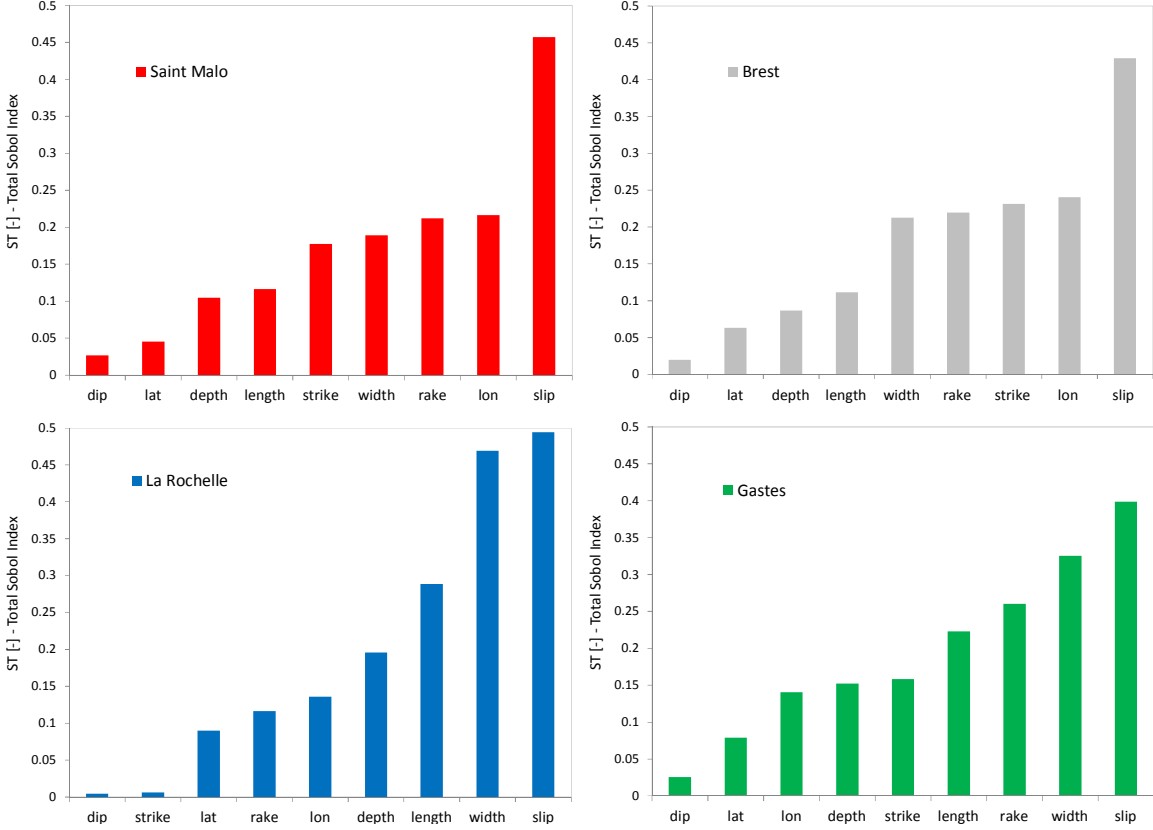

**Figure 7:** Total order Sobol Index computed for the French Atlantic Gauges.





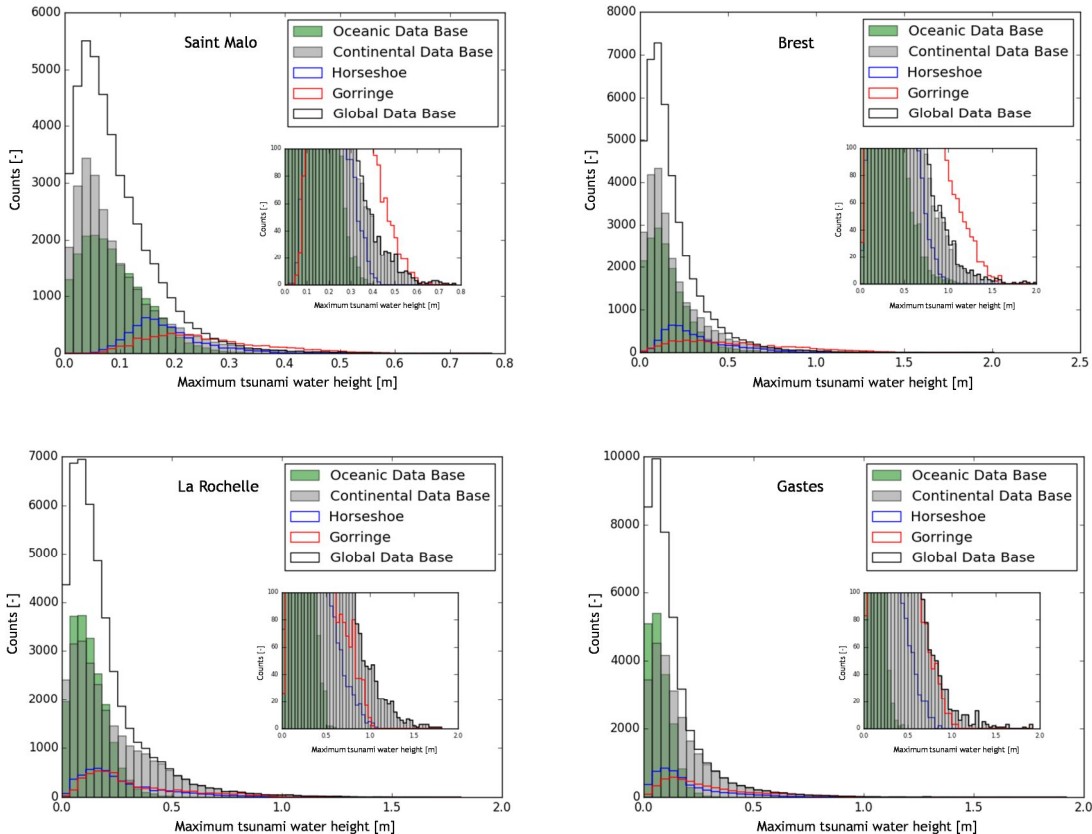

**Figure 8:** Frequency distribution of the maximum tsunami water height for tsunamis scenarios generated by the Horseshoe bank (blue line) and the Gorringe bank (red line). Comparison with scenarios associated to the oceanic shelf (green), the continental shelf (grey) and the global data base (black line).





**Figure 9:** a) Magnitude distribution of earthquake events leading to the strongest (higher than $\mu_m+2\sigma_m$) tsunamis scenarios at the French Atlantic Gauges compared to the global data set Magnitude distribution of tsunamigenic earthquakes; b) percent of strongest tsunamis scenarios (higher than $\mu_m+2\sigma_m$) associated to oceanic and continental shelves in the magnitude range $M_w$ [7.8 – 8.5]. # indicates the number of tsunami scenarios.





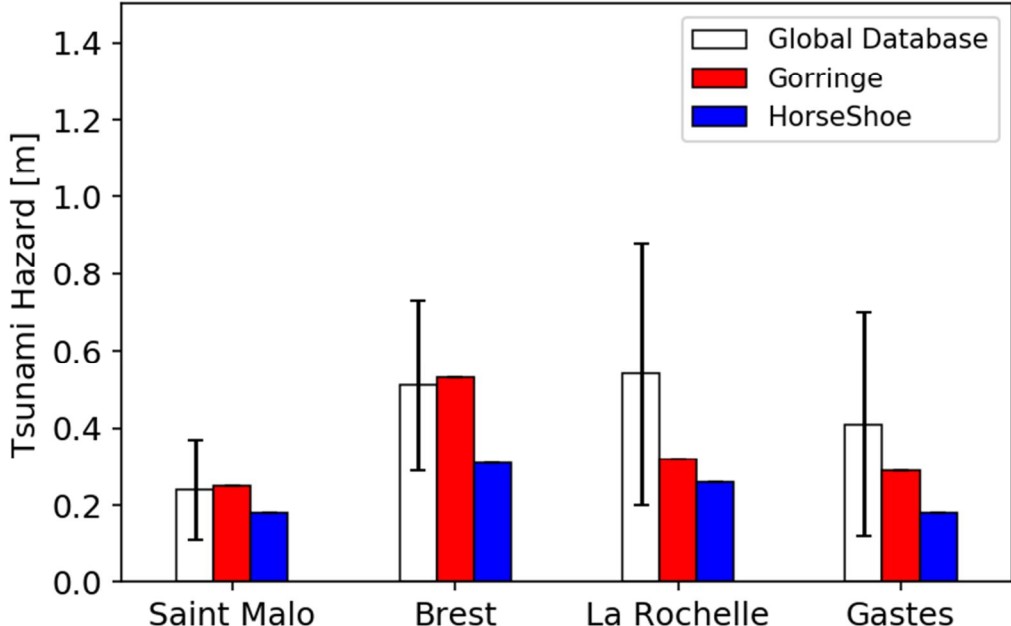

**Figure 10:** Tsunamis hazard ("$\mu_m$" modelled values) for the French Atlantic Gauges associated to the 1755-like tsunamis scenarios form the Horseshoe bank (blue) and the Gorringe bank (red). Comparison with the tsunami hazard ("$\mu_m+2\sigma_m$" modelled water height) issued from uncertainty quantification (black line). The black bars represent the uncertainty related to the use of a meta-model instead of the original model (equal to $+/-2\sigma_r$).





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
