# Peer review of "Development of a methodological framework for the assessment of seismic induced tsunami hazard through uncertainty quantification: application to the Azores-Gibraltar Fracture Zone"

_Natural Hazards and Earth System Sciences, 2018_

## Referee Comment (RC1) · L. Matias (Referee) · 13 Jun 2018

**NHESS-142 Development of a methodological framework for the assessment of seismic induced tsunami hazard through uncertainty quantification: application to the Azores-Gibraltar Fracture Zone**

Vito Bacchi, Ekaterina Antoshchenkova, Hervé Jomard, Lise Bardet, Claire-Marie Duluc, Oona Scotti, Hélène Hebert

**Comments by Luis Matias, University of Lisbon**

**Recommendation**

It is my recommendation that this paper cannot be accepted for publication. There are some major considerations that I will describe below to support my conclusion. Additional comments follow that could help the authors if they wish to resubmit a new paper.

**Major comments**

As I understand the paper, the authors use the "uncertainty quantification technique" to derive "meta-models" that could be used to make an extensive exploitation of the parameter space without requiring the time-consuming process of tsunami modelling. The authors present two applications of the methodology, one addressing the sensitivity analysis and the other as a contribution to tsunami hazard evaluation. While the methodology described seems interesting, the first application to sensitivity analysis suffers from several shortcomings and the application to tsunami hazard evaluation is seriously flawed.

As the authors mention in the paper, there are two main procedures for tsunami hazard evaluation, the probabilistic method (PTHA) and the deterministic method (DTHA). The authors explore a few scenario databases to apply the methodology that allegedly exploit randomly (with uniform probability) the range of source parameters considered. There is no probability attached to the geological realization of each scenario and so the results presented in the paper should be included in the second type of evaluations, DTHA. This is clear when the authors compare the present study with the work published for the Indian Ocean by Roshan et al. (2016). As the authors mention, DTHA studies rely on the study of "maximum credible scenarios". The databases used in the paper make a random exploitation of the model space which gives origin to a large number of scenarios (over 50,000 in one case) where only a few of them, not identified, would classify as "maximum credible scenarios". In my opinion, this approach precludes any statistical use of the results as a contribution to DTHA.

The major problem I found in the paper is related to the characterization of tsunamigenic earthquakes that is used for the Azores Gibraltar Fracture Zone (AGFZ), that I prefer to mention here as the Azores Gibraltar Plate Boundary (AGPB). The authors arbitrarily divide the AGPB into two domains, "oceanic" from 18ºW to 10ºW and "continental" from 10ºW to 7ºW. In latitude both domains span from 34ºN to 40ºN. According to the authors, the "oceanic" domain is characterized by "thin oceanic crust" while the Eastern domain is characterized by "thickened oceanic to continental crust". This classification is at odds with the knowledge of the deep structure of the Gulf of Cadiz and the Gloria Fault zone as given by Martínez-Loriente (2014) and Batista et al. (2017) that show that such a simplistic classification is inappropriate.

The authors consider that the maximum magnitude allowed for the "oceanic" domain is 8.3, being 9.3 for the "continental" domain. There is no explanation whatsoever for the arbitrary attribution of these values. There is no reference in the literature for this area that proposes such a high value of magnitude. It is in contradiction with the known studies for THA and Seismic Hazard Assessment (SHA) and it should be properly explained. The highest magnitude considered for the AGPB is the one estimated for the 1st November 1755 earthquake that, at most, as an inferred magnitude of 8.8 (e.g. Johnston, 1996). The strong discrepancy between the two zones as regards the maximum magnitude is not explained in the text. In fact, given the uncertainty today on the probable source for the 1755 earthquake, the authors recognize that the "oceanic" database does not contain one of the proposed sources for this event, the Gorringe Bank. The authors try to mitigate this problem by adding an ad-hoc extra database of scenarios for that particular source, thus violating the basic principle in their origin, make a random choice of parameters that would include all possible sources of extreme events. The "oceanic" database clearly fails to attain this objective.

The heterogeneity of the AGPB has been captured by all SHA and THA studies made in the area, most of them not considered in the paper, e.g. Vilanova and Fonseca (2007), Woessner et al. (2015) for SHA or Omira et al. (2015), Omira et al. (2016), Matias et al. (2013) for THA. The criteria used in these works take into consideration the best knowledge on seismology and seismotectonics, the same considerations that should guide the development of the tsunamigenic databases used in the paper. Instead the authors choose to build random scenarios over an all-encompassing range of source parameters. True, the geological knowledge of the area is still incomplete and there are uncertainties or disagreements, for example, on the identification of the source for the 1755 event. This fact, however, does not justify the replacement of an "educated" database supported by expert opinions by a purely random source distribution.

As regards the source of the largest historical earthquake and tsunami reported for the area, the 1st November 1755 event, the authors identify the difficulty to find in the literature a consensual solution. Some of the more relevant references to that problem were not considered in the paper (in chronological order here), Baptista et al. (2003), Terrinha et al. (2003), Gracia et al. (2003), Barkan et al. (2009) and Cunha et al. (2010.). These and other studies show how artificial is the split between "oceanic" and "continental" source areas with very different maximum magnitudes (8.3 and 9.3 respectively). The derived databases cannot be considered as representative or as including the maximum credible tsunamigenic sources in the AGPB as claimed.

Another shortcoming in building the databases is the absence of an explanation on the way that the parameters length, width, slip and magnitude are explored, since they must be related by the equations (15) and (16). Equation (15) also depends on the value attributed to the shear modulus in the equation, but no mention to that is found in the paper. Contrary to what is inferred from the text and Table 5, the magnitude is not uniformly sampled between the limits indicated. This is clear from the histogram presented in figure 9a. A clear explanation on the procedure used to relate these 5 parameters must be clearly presented, as for example is done for the AGPB by Matias et al. (2013) and Baptista et al. (2017). If other scaling laws are used in a similar process these should be clearly discussed in the paper.

As I understand, the purpose of the adopted methodology was "to build a numerical database of a wide bunch of the maximum possible tsunamis generated by the AGFZ". It is demonstrated in the paper that the most influential source parameter to the maximum water height observed at the cost is the average slip in the fault, that scales with magnitude, if earthquake physics are respected. Then we should expect that only the largest magnitude events would contribute to the desired hazard parameter. It is then hard to understand what is the contribution to the proposed investigation of events with magnitude < 8.0 that constitute 50% of the database (figure 9a). Furthermore, the authors recognize that most of the hazard is generated by "well located-oriented" faults. Since the authors explore randomly the full source parameter range, it is difficult to be sure that these "well located-oriented" faults are caught by the databases, when a small set of sources have the largest magnitudes tested. If we sample the strike every 30⁰, the dip every 10⁰, the rake every 30⁰ and the epicentre coordinates every 1⁰, the depth every 5 km, we arrive at ~400,000 possible sources for each magnitude level. If we allow for some variability in length, width and slip for each magnitude level, then more than $10^6$ scenarios are possible. With only a few thousand scenarios randomly selected we should not expect to find the "well located-oriented" faults that are responsible for the highest water level at the coast.

As regards the depth of the tsunamigenic sources (presumably below the sea bottom, but not clear) the authors mention that "the maximum oceanic depth is considered to be of the order of 10 km whereas the continental crust can attain 30 km depth". This consideration is translated into the values provided in Table 5. The authors thus disregard the evidence that recent microseismicity investigations in the area show that the brittle lithosphere extends at least to 50 km depth (Geissler et al., 2010, Grevemeyer et al., 2016, Grevemeyer et al., 2017, Silva et al., 2017). The large focal depth found in the area is interpreted as representing the rheological properties of very old oceanic lithosphere that forms most of the AGPB. This rheology as consequences for the scaling laws appropriate for the generation of large tsunamis in this domain as discussed in Matias et al., 2013 and Baptista et al., 2017. The paper does not explain also how the conflict between a shallow focus and a large vertical width is solved when selected by random?

The AGPB is an area prone to tsunamis and so the investigation of the resulting tsunami hazard is an important objective. However, the authors fail to recognize the synthesis tsunami catalogues already published for the area, while discussing the merits of the built database (Baptista et al., 2009 and Kaabouben et al., 2009).

About the methodology, as I understand it, one "meta-model" must be built for each of the desired outputs to be investigated. In the paper 4 tide gauge locations where chosen for testing and only one hazard parameter elected, the maximum tsunami water height. Nowadays, following the development of the tsunami warning systems, both national and regional, THA studies, deterministic or probabilistic, are applied to very close observation points at the coast, typically a few tens of km, to account for the high variability in tsunami propagation and coastal effects (e.g. Molinari et al., 2016, Roshan et al., 2016). Nowadays, to mitigate the massive computing power required to generate the large number of tsunami scenarios required for PTHA, authors have resorted to parallel computing using GPU and to the concept of Elementary Source or Empirical Green Function (e.g. Molinari et al., 2016 or Baptista et al. (2017). Applying the "uncertainty quantification technique" to reproduce the results that are nowadays required, with an extensive set of observation points (or forecast points) may be as computer intensive as the traditional approach of tsunami propagation for a large number of scenarios. The paper fails to present the advantages of the methodology regarding a realistic application.

The paper applies the "uncertainty quantification technique" to make a sensitivity analysis aiming to identify the source model parameters most relevant for each of the four target sites. A general classification on the most relevant parameters is given and a simple interpretation is presented. However, a more detailed analysis would be much more relevant. What are the epicentres and magnitudes that

contribute most to the highest hazard levels in each site? This is a common exercise in Probabilistic Seismic Hazard Assessment called disaggregation. My expectation is that this analysis would show that the most relevant contributions would come from unrealistic geological sources, even taking into consideration the uncertainties ascribed to the seismotectonic framework of the AGPB.

The authors use systematically the moment statistics of gaussian distribution to characterize the several variables discussed in the text. However, the authors recognize that many of these quantities have long tailed distributions, as expressed in table 4 or in figure 9a. The gaussian distribution doesn't seem the most appropriate model for the distribution of extreme values as desired. This question should be properly addressed in a more formal way in the paper.

The authors claim to obtain maximum water height estimates for the 4 locations identified in the text that are close to the coast line. The representative water depths at those points are 8, 15, 28 and 40 m. It is well known that the shallow water equations used for tsunami propagation loose their precision close to the coast and at water depths lower than 50 m. It is usually recommended that shallow water results should stop at that water depth (e.g Kamigaichi, 2009). From that point higher resolution models or empirical relationships should be used to estimate wave heights at the coast or shallower waters (ibid.). This issue should be addressed in the paper.

**Other comments**

The mathematical notation needs to be carefully revised. In particular vectors/matrices and scalars should be clearly identified, for example using bold type. The meaning of variable names and its capitalization should be kept throughout the paper.

The mention of bibliographic references should follow the journal standards and double "(())" should be avoided.

One disadvantage of the "meta-model" approach is that it adds an extra layer of uncertainty to the uncertainties already due to the source parameter space variability. This issue is addressed in the paper, but it may ne undesirable in practical applications.

The list of references is clearly incomplete for many of the subjects addressed.

The numbers on the tables do not follow the order they are referenced in the text.

The numbers on the figures do not follow the order they are referenced in the text.

The paper quotes extensively the Antoshchenkova et al. (2016) work that corresponds to an abstract to a conference. This should be avoided by presenting the main results of that work in the paper or as a supplement, otherwise many sentences in the paper cannot be verified by the reader.

**References mentioned in the comment. (*) signals works not cited in the paper**

Antoshchenkova, E., Imbert, D., Richet, Y., Bardet, L., and Duluc, C.-M.: Propagation of uncertainties for an evaluation of the azores Gibraltar fracture zone tsunamigenic potential., EGU General Assembly Conference, 2016.

(*) Baptista, M. A., Miranda, J. M., Chierici, F., & Zitellini, N. (2003). New study of the 1755 earthquake source based on multi-channel seismic survey data and tsunami modeling. Natural Hazards and Earth System Sciences, 3(5), 333-340.

(*) Baptista, M. A., & Miranda, J. M. (2009). Revision of the Portuguese catalog of tsunamis. Natural Hazards and Earth System Sciences, 9(1), 25-42.

(*) Baptista, M. A., Miranda, J. M., Matias, L., & Omira, R. (2017). Synthetic tsunami waveform catalogs with kinematic constraints. Natural Hazards and Earth System Sciences, 17(7), 1253.

(*) Barkan, R., Uri, S., & Lin, J. (2009). Far field tsunami simulations of the 1755 Lisbon earthquake: Implications for tsunami hazard to the US East Coast and the Caribbean. Marine Geology, 264(1-2), 109-122.

(*) Batista, L., Hübscher, C., Terrinha, P., Matias, L., Afilhado, A., & Lüdmann, T. (2017). Crustal structure of the Eurasia–Africa plate boundary across the Gloria Fault, North Atlantic Ocean. Geophysical Journal International, 209(2), 713-729.

(*) Cunha, T. A., Watts, A. B., Pinheiro, L. M., & Myklebust, R. (2010). Seismic and gravity anomaly evidence of large-scale compressional deformation off SW Portugal. Earth and Planetary Science Letters, 293(1-2), 171-179.

(*) Geissler, W. H., Matias, L., Stich, D., Carrilho, F., Jokat, W., Monna, S., IbenBrahim, A., Mancilla, F., Gutscher, M.-A., Sallarès, V. & Zitellini, N. (2010). Focal mechanisms for sub-crustal earthquakes in the Gulf of Cadiz from a dense OBS deployment. Geophysical Research Letters, 37(18).

Grevemeyer, I., Matias, L., & Silva, S. (2016). Mantle earthquakes beneath the South Iberia continental margin and Gulf of Cadiz–constraints from an onshore-offshore seismological network. Journal of Geodynamics, 99, 39-50.

(*) Grevemeyer, I., Lange, D., Villinger, H., Custodio, S., & Matias, L. (2017). Seismotectonics of the Horseshoe Abyssal Plain and Gorringe Bank, eastern Atlantic Ocean: Constraints from ocean bottom seismometer data. Journal of Geophysical Research: Solid Earth, 122(1), 63-78.

Johnston, A. C. (1996). Seismic moment assessment of earthquakes in stable continental regions—III. New Madrid 1811–1812, Charleston 1886 and Lisbon 1755. Geophysical Journal International, 126(2), 314-344.

(*) Gracia, E., Danobeitia, J., Vergés, J., & PARSIFAL Team. (2003). Mapping active faults offshore Portugal (36 N–38 N): implications for seismic hazard assessment along the southwest Iberian margin. Geology, 31(1), 83-86.

(*) Kaabouben, F., Baptista, M. A., Iben Brahim, A., Mouraouah, A. E., & Toto, A. (2009). On the moroccan tsunami catalogue. Natural Hazards and Earth System Sciences, 9(4), 1227-1236.

(*) Kamigaichi, O. (2009). Tsunami forecasting and warning. In Encyclopedia of complexity and systems science (pp. 9592-9618). Springer New York.

(*) Matias, L. M., Cunha, T., Annunziato, A., Baptista, M. A., & Carrilho, F. (2013). Tsunamigenic earthquakes in the Gulf of Cadiz: fault model and recurrence. Natural hazards and earth system sciences, 13(1), 1-13.

(*) Martínez-Loriente, S., Sallarès, V., Gràcia, E., Bartolome, R., Dañobeitia, J. J., & Zitellini, N. (2014). Seismic and gravity constraints on the nature of the basement in the Africa-Eurasia plate boundary: New insights for the geodynamic evolution of the SW Iberian margin. Journal of Geophysical Research: Solid Earth, 119(1), 127-149.

(*) Molinari, I., Tonini, R., Lorito, S., Piatanesi, A., Romano, F., Melini, D., Hoechner, A., Gonzàlez Vida, J.M., Maciás, J., Castro, M.J. & de la Asuncion, M. (2016). Fast evaluation of tsunami scenarios: uncertainty assessment for a Mediterranean Sea database. Natural Hazards and Earth System Sciences, 16(12), 2593.

(*) Omira, R., Baptista, M. A., & Matias, L. (2015). Probabilistic tsunami hazard in the Northeast Atlantic from near-and far-field tectonic sources. Pure and Applied Geophysics, 172(3-4), 901-920.

Omira, R., Matias, L., & Baptista, M. A. (2016). Developing an event-tree probabilistic tsunami inundation model for NE Atlantic coasts: Application to a case study. Pure and Applied Geophysics, 173(12), 3775-3794.

Roshan, A. D., Basu, P. C., & Jangid, R. S. (2016). Tsunami hazard assessment of Indian coast. Natural Hazards, 82(2), 733-762.

**(\*)** Silva, S., Terrinha, P., Matias, L., Duarte, J. C., Roque, C., Ranero, C. R., ... & Zitellini, N. (2017). Micro-seismicity in the Gulf of Cadiz: Is there a link between micro-seismicity, high magnitude earthquakes and active faults?. Tectonophysics, 717, 226-241.

**(\*)** Terrinha, P., Pinheiro, L. M., Henriet, J. P., Matias, L., Ivanov, M. K., Monteiro, J. H., ... & Rovere, M. (2003). Tsunamigenic-seismogenic structures, neotectonics, sedimentary processes and slope instability on the southwest Portuguese Margin. Marine Geology, 195(1-4), 55-73.

**(\*)** Vilanova, S. P., & Fonseca, J. F. (2007). Probabilistic seismic-hazard assessment for Portugal. Bulletin of the Seismological Society of America, 97(5), 1702-1717.

**(\*)** Woessner, J., Laurentiu, D., Giardini, D., Crowley, H., Cotton, F., Grünthal, G., ... & Hiemer, S. (2015). The 2013 European seismic hazard model: key components and results. Bulletin of Earthquake Engineering, 13(12), 3553-3596.

---

## Referee Comment (RC2) · Anonymous Referee #2 · 20 Jun 2018

**General comments**

Quantification of uncertainty is an important aspect in tsunami hazard assessment (THA) using deterministic or scenario-based methods—an aspect that is often overlooked. The authors of "Development of a methodological framework for the assessment of seismic induced tsunami hazard through uncertainty quantification: application to the Azores-Gibraltar Fracture Zone" describe sophisticated methods for uncertainty

quantification and sensitivity analysis and show how the methods can be applied for THA using potential earthquake sources along that Azores-Gibraltar Fracture Zone and targets along the French Atlantic coast. The authors provide an extensive background for the study with numerous citations. The main weakness in the paper, as described below, is the confusing description of all the different methods and how they are related. The parameters of the case study may lack some realism (as acknowledged by the authors on the bottom of pg. 10), but I believe the main objective of the case study is to demonstrate how the statistical methods can be applied.

Specific comments:

(1) This study actually discusses two aspects associated with THA: uncertainty quantification and sensitivity analysis (the latter, not included in the title of the paper). The relationship between these two aspects is confusing. Part of the problem is Figure 1 where they are related through "Promethee". "Promethee" is not mentioned until pg. 11, and only then if very briefly described. It would be much better of the reader if Figure 1 was simplified and modified to include how the methods described by authors (i.e., kriging meta-model) are incorporated into both aspects of THA (top of pg. 8). It might be better to move the meta-model explanation (Sec. 2.4) to the beginning of Section 2 and then show how uncertainty quantification and sensitivity analyses are determined from the meta-model.

(2) In the development and application of their method, the authors make several jumps in approximation without much explanation or validation:

a. To start with, it appears to be given that tsunami-modeling results can be uniquely decomposed according to Eqn. 5. Is this so, especially considering the inclusion of nonlinear advection?

b. On pg. 6 (top), the meta-model is preferred over Monte Carlo sampling because "...a large amount of simulations are necessary", even though the design database for the meta-model also includes a large number of simulations. In this case, is there an

advantage to using the meta model? Convergence appears to be achieved just with the number of starting simulations in the design database (Fig. 6).

c. On page 7, calculation of Sobol indicies jumps to total indicies to extended-Fast method to Janssen method (implemented in R) , all because of computational reasons. Is accuracy of statistical measures preserved at each step? Unclear what the variables are in the simulation cost (end of Sec. 2.3).

d. LHS is replaced by using the results from a previous modeling study. What is lost by doing this?

(3) The design database is an important part of this study. Although it looks like the essential aspects of the modeling are described in Section 3.2, the actual work is described only in abstracts (Antoshchenkova et al., 2016; Imbert et al., 2015). Is there a publication or report the fully documents the numerical modeling?

(4) The authors frequently use generic variables, whereas relating these variables to ones specific to the tsunami problem would greatly aid the reader. Please be more specific on the definition of $y$ (pg. 5), rather than just "quantitities of interest". See also Figure 1, Eqn. 5, pg. 9, Appendix, etc.

(5) Figure 4: Indicate "residuals" of what?

(6) Although the English is definitely understandable, the manuscript could use the services of a copy editor during revision.

---

## Referee Comment (RC3) · Anonymous Referee #3 · 2 Aug 2018

The manuscript aims to present a method to assess tsunami hazard and its uncertainties using sensitivity analysis and meta-modeling technique. The authors then applied their method to a case study, identifying a source area (west of Gibraltar strait) and a target area (French Atlantic coast). Even though the methodology sounds very interesting to support probabilistic tsunami hazard assessments, in my opinion the paper could not be published in its present form.

[Figure]

As major comments I found the following issues:

1) The manuscript lacks some important references on one of the main topic of the manuscript itself: quantification of uncertainties in tsunami hazard assessment. Several works faced this topic (among others, Sorensen et al, 2012, Horspool et al, 2014, Davies et al., 2016, Lorito et al., 2015, Selva et al., 2016) using probabilistic approaches and proposing methods to strongly reduce computation costs. Also approaches to develop databases based on combination of elementary sources to speed up tsunami modeling have been proposed (Miranda et al, 2015 and Baptista et al 2017), even with the quantification of the associated uncertainty (Molinary et al., 2016). I think that the manuscript should be rethought considering the existing framework.

2) I am a bit lost between the explanation of the method and how it was applied for the case study. In particular, the environment Promethee seems to me the glue of everything, so I found strange that it was quickly mentioned only in section 3. I would suggest to present what Promethee is, which module contains, how is used in this work and what the authors customized in the frame of Promethee for their scopes. For example, I did not understand subsection 2.2: why using a subsection to describe a module that was not used (if I understand well), i.e., the Monte Carlo method originally implemented in Promethee? Authors could state that uncertainty propagation used in Promethee environment for this study is modeled using an external package based on kriging. My suggestion is to rearrange section 2 and 3 in a clearer way.

3) Even though I understand that the aim of the paper is illustrative of the method, I have some doubts about the design database: the number of considered scenarios (about 5,000) seems to me relatively low with respect the number of the considered parameters and their large variability. My concern is that a too rough sampling of input parameters could introduce bias in the design of the meta-model. I would ask to authors to address this issue.

4) The work often refers to the meeting communication by Antoshchenkova et al.,
(2016) which cannot be verified by readers. This should be avoided in a peer-reviewed manuscript. When needed, important element from that work should be reported directly here.

5) About tsunami modeling: linear or non-linear shallow water equations have been used? This is not specified in the text, however depth of tide gauges is 40 m or less (and less than 30 m for 3 stations), where non linearity effects become very strong and cannot be neglected.

6) I did not understand if magnitude is uniformly sampled as stated in the text (page 14, line 17), since in figure 9a the distribution seems very different. Authors should clarify this point.

7) Even though the manuscript is illustrative, maybe to make the case study less far from something more realistic, the authors could use a magnitude distribution from catalogs

8) The numbering of figures and tables should be adjusted with respect the order in which they are mentioned in the text.

Other comments:

1) Page 2, lines 28-30: If I understand well the sentence, I disagree with the authors, since I would say the opposite: if the knowledge on the area is poor, a probabilistic approach can at least quantify this level of uncertainty, whereas deterministic approaches hardly can catch what is unknown and they could suffer of relevant bias in the hazard analysis.

2) Page 3, lines 13: The authors are referring to deterministic approaches only or to probabilistic ones as well?

3) Page 17, line 17: I cannot really understand the meaning of the sentence "for a given magnitude, a tsunami generated by a well located-oriented fault would be potentially more hazardous". Well located-oriented with respect what? I would remove the sentence, since it is already stated that the analysis suggests that position and orientation (strike/rake) for two of the stations are very relevant for hazard assessment. I

4) Figure 3: I guess that tick labels for longitude axis should be W instead O

References

Baptista, M. A., Miranda, J. M., Matias, L., & Omira, R. (2017). Synthetic tsunami waveform catalogs with kinematic constraints. Natural Hazards and Earth System Sciences, 17(7), 1253.

Davies G, Griffin J, Løvholt F, Glimsdal S, Harbitz C, Thio HK, Lorito S, Basili R, Selva J, Geist E, Baptista MA: A global probabilistic tsunami hazard assessment from earthquake sources, Geological Society, London, Special Publications, 456, 23 February 2017, https://doi.org/10.1144/SP456.5, 2016

Horspool, N., Pranantyo, I., Griffin, J., Latief, H., Natawidjaja, D. H., Kongko, W., Cipta, A., Bustaman, B., Anugrah, S. D., and Thio, H. K.: A probabilistic tsunami hazard assessment for Indonesia, Nat. Hazards Earth Syst. Sci., 14, 3105-3122, https://doi.org/10.5194/nhess-14-3105-2014, 2014.

Lorito, S., Selva, J., Basili, R., Romano, F., Tiberti, M. M., and Piatanesi, A.: Probabilistic Hazard for Seismically-Induced Tsunamis: Accuracy and Feasibility of Inundation Maps, Geophys. J. Int., 200, 574–588, https://doi.org/10.1093/gji/ggu408, 2015.

Miranda, J. M., Baptista, M. A., and Omira, R.: On the use of Green's summation for tsunami waveform estimation: a case study, Geophys. J. Int., 199, 459–464, 2014

Molinari, I., Tonini, R., Piatanesi, A., Lorito, S., Romano, F., Melini, D., Vida, J. G., Macias, J., Castro, M., and de la Asuncion, M.: Fast evaluation of tsunami scenarios: uncertainty assessment for a Mediterranean Sea database, Nat. Hazards Earth Syst. Sci., 16, 2593–2602, https://doi.org/10.5194/nhess-16-2593-2016, 2016

Selva, J., Tonini, R., Molinari, I., Tiberti, M. M., Romano, F., Grezio, A., Melini, D.,

Piatanesi, A., Basili, R., and Lorito, S.: Quantification of source uncertainties in Seismic Probabilistic Tsunami Hazard Analysis (SPTHA), Geophys. J. Int., 205, 1780–1803, https://doi.org/10.1093/gji/ggw107, 2016

Sørensen M.B., Spada M., Babeyko A., Wiemer S., Grunthal G.: Probabilistic tsunami hazard in the Mediterranean Sea, Journal of Geophysical Research, 117, https://doi.org/10.1029/2010JB008169, 2012.
* * *

---

## Referee Comment (RC4) · Anonymous Referee #4 · 8 Aug 2018

Review of "Development of a methodological framework for the assessment of seismic induced tsunami hazard through uncertainty quantification: application to the Azores-Gibraltar Fracture Zone" by Bacchi et al.

This paper describes a methodology for tsunami hazard analysis using uncertainty quantification techniques and meta-models, and then presents an application of the methodology based on an example application to the Azores-Gibraltar Fracture Zone.

[Figure]

I find the methodology that has been developed interesting and that publication of a significantly revised manuscript would be useful for other researchers interested in similar techniques. However, the demonstration application shows that the methodology is incomplete for practical purposes for reasons given below, and the authors need to acknowledge and address some of these issues.

Specific points:

1) The presented application to the Azores-Gibraltar Fracture Zone takes very little in the way of constraints or weightings from what is known about the geology, geophysics and seismology of the area. A better indication of how prior knowledge can be included in such a study is something that I think will be needed before the presented methodology can be put into practical use. Related to this, is a lack of connection to the time-frame associated with the hazard, eg the return period associated with a water level at the study sites, something that is necessary for many applications of tsunami hazard.

I think the authors are aware of the current limitations of their results, and they repeatedly mention that this is '. . . not an operational assessment . . .'. There could be some differences in interpretation about what 'operational' means, and I think it would be better if they could be more explicit about this, eg at the end of the Introduction they might consider something like: '. . . not an operational assessment of tsunami hazard along the French Atlantic Coast, and no reliance should be placed on these results for practical purposes'. With something like that, and more acknowledgement of the current limitations and areas for more work, the authors would be on safer ground that this manuscript is just an introduction to the 'meta-model' concept.

2) The methodology is hard to follow, and in particular, the relationship between the various different databases in Table 5 and how they are used in the analysis, was difficult to understand. I suggest the authors try to construct a flow diagram or other type of figure to explain this graphically (could be somewhat similar to Figure 1).

[Figure]

3) The Antoschenkova (2016) reference is key to understanding the design database, but this is just a conference presentation and I could not find the information needed to assess the suitability of this work to the purpose to which it is applied. If the information is not published elsewhere, I suggest a summary be included here – perhaps as an Appendix.

4) It seems a pity to me that all of the study sites are located along the French coast. In particular the sensitivity analysis is, in my view, one of the most interesting parts of the paper, but it would be really nice to see how the sensitivity applies to sites closer to the source region.

5) I'm presuming that the magnitudes in Table 5 are calculated after the uniform sampling of length, width and slip and applying constraints to these. And that the distribution of magnitudes is just what falls out from this process (and not linked to any assumption about earthquake statistics – see point 1). This needs clarification (unless it was mentioned and I missed it).

6) Non-uniform slip distributions have been shown to be influential in estimating tsunami hazard (see eg Geist and Dmowska, 1999), but the study here is (as far as I could tell) all based on uniform slip modelling. It is not at all obvious to me how this might be included in the methodology presented here, and it would be nice if the authors could comment on this.

Similar to this is the approximation of the rupture by a rectangular plane, which may be suitable for low magnitude earthquakes but will not be realistic for ones at the upper end of their study range. Again, this does not seem like an easy thing to incorporate.

7) The authors may find that some of their problems with finding suitable wave height statistics could be improved by working with log-normal distributions. A starting point for this would be to search for 'Aida-k parameter' (original paper Aida, 1978).

8) The paper needs extensive copy-editing to improve the English.

Aida, I., 1978. Reliability of a tsunami source model derived from fault parameters. Journal of Physics of the Earth, 26(1), pp.57-73.

Geist, E.L. and Dmowska, R., 1999. Local tsunamis and distributed slip at the source. In Seismogenic and tsunamigenic processes in shallow subduction zones (pp. 485-512). Birkhäuser, Basel.

———————————————————

---

## Author Comment (AC1) · 3 Oct 2018

**Responses to Dr. Luis Matias, University of Lisbon**

October 3th, 2018

Re: Re-submission of manuscript *"Development of a methodological framework for the assessment of seismic induced tsunami hazard through uncertainty quantification: application to the Azores-Gibraltar Fracture"* by Vito Bacchi et al. NHESS-2018-142.

Dear Dr. Luis Matias,

We appreciate your careful review and constructive suggestions. It is our belief that the manuscript is substantially improved after making the suggested edits. Following this letter are the reviewer comments with our responses in italics, including how and where the text was modified. Changes made in the manuscript are marked in blue (see attached file).

According to the reviewer comments, we propose to modify the title of the paper as follow:

*"Development of a methodological framework for the assessment of seismic induced tsunami hazard through uncertainty quantification: the test case of the Azores-Gibraltar Plate Boundary".*

Thank you for your consideration.

Sincerely,

Vito Bacchi on behalf of authors

**Major comments**

**C1.** As I understand the paper, the authors use the "uncertainty quantification technique" to derive "meta-models" that could be used to make an extensive exploitation of the parameter space without requiring the time-consuming process of tsunami modelling. The authors present two applications of the methodology, one addressing the sensitivity analysis and the other as a contribution to tsunami hazard evaluation. While the methodology described seems interesting, the first application to sensitivity analysis suffers from several shortcomings and the application to tsunami hazard evaluation is seriously flawed.

As the authors mention in the paper, there are two main procedures for tsunami hazard evaluation, the probabilistic method (PTHA) and the deterministic method (DTHA). The authors explore a few scenario databases to apply the methodology that allegedly exploit randomly (with uniform probability) the range of source parameters considered. There is no probability attached to the geological realization of each scenario and so the results presented in the paper should be included in the second type of evaluations, DTHA. This is clear when the authors compare the present study with the work published for the Indian Ocean by Roshan et al. (2016). As the authors mention, DTHA studies rely on the study of "maximum credible scenarios". The databases used in the paper make a random exploitation of the model space which gives origin to a large number of scenarios (over 50,000 in one case) where only a few of them, not identified, would classify as "maximum credible

scenarios". In my opinion, this approach precludes any statistical use of the results as a contribution to DTHA.

*We recognize that the first version of the paper was not clear and that the methodology suffers from some shortcomings in the original text. In the revised version of the paper, the methodology is presented more clearly in Section 2 and a work flow (Figure 1) is also introduced in the paper in order to better detail each step of the methodology.*

*Concerning THA, we are not sure to understand the exact meaning of the following remark "In my opinion, this approach precludes any statistical use of the results as a contribution to DTHA." The only statistical measure we defend in this paper is to account for uncertainties in DTHA by considering a mean+2 standard deviation value of the modelled maximum tsunami height at a given location off the French Atlantic Coast. The choice is arbitrary but largely discussed in section 4 and in the conclusions. Moreover, as reported in various sections of the paper (i.e. in the abstract, the introduction and the conclusions), our work is essentially methodological and does not permit assessing tsunami hazard along the French Atlantic Coast. This isn't the main objective of the paper. For an operational DTHA along the French Atlantic Coast, we should, at first improve the tsunami numerical model (bathymetry) and then explore more exhaustively scenarios from the literature depending on the location of interest.*

*The objective of our work, which was not clearly defined in the original paper, is to propose a new methodology for the improvement of DTHA, as explained in the introduction of the revised paper:*

*"Nowadays, in the classical DTHA, as reported in the cited papers, MCS is mainly focused on seismic source parameters (MCS_p) and not on the tsunamis heights at a given location (MCS_h). The great M 9.0 Tohoku-Oki subduction earthquake of 2011, the largest ever recorded in Japan (Saito et al., 2011), has clearly shown the limitations of the MCS_p approach. In fact, considering the uncertainties related to the characterization of seismic sources, using MCS_p for DTHA may lead to an underestimation of MCS_h in the area of interest.*

*In this context, the objective of our work is to propose a new methodology to go beyond the MCS_p approach and focus more on MCS_h through the evaluation of all the possible tsunamis heights at a given location. The MCS_h approach has the advantage that it accounts for other scenarios, generated by potential seismic sources not associated to the MCS_p but generating tsunamis heights potentially impacting the zone to protect. The proposed methodological framework for the improvement of DTHA is based on classical uncertainty quantification (UQ) techniques (i.e. Saltelli et al., 2000; Saltelli et al., 2008), which are robust statistical tools permitting to largely explore a given seismic area and provide all the possible tsunami heights generated by this area. This approach is a new philosophy for DTHA and can also permit to ensure that MCS_p is the MCS_h scenario for the target area."*

**C2.** The major problem I found in the paper is related to the characterization of tsunamigenic earthquakes that is used for the Azores Gibraltar Fracture Zone (AGFZ), that I prefer to mention here as the Azores Gibraltar Plate Boundary (AGPB).

*The correction was made in the text.*

**C3.** The authors arbitrarily divide the AGPB into two domains, "oceanic" from 18°W to 10°W and "continental" from 10°W to 7°W. In latitude both domains span from 34°N to 40°N. According to the authors, the "oceanic" domain is characterized by "thin oceanic crust" while the Eastern domain is characterized by "thickened oceanic to continental crust". This classification is at odds with the knowledge of the deep structure of the Gulf of Cadiz and the Gloria Fault zone as given by Martínez-Loriente (2014) and Batista et al. (2017) that show that such a simplistic classification is inappropriate.

*The main goal of our paper being methodological, we are not trying to provide a THA with a detailed state of the art seismotectonic model. The goal of our first order zonation was in fact more a way to delimitate two seismogenic domains:*

- *a western domain with mainly extensional to transtensive sources, with a limited seismogenic depth*
- *an eastern domain with mainly compressive to transpressive sources, with a deeper seismogenic depth*

*As better explained in the conclusions of the revised paper, we are testing the methodology with rough hypothesis and then checking that such rough hypothesis are consistent in terms of MCS_h by hypothesizing a conservative scenario along Gorringe and Horseshoe and verifying that tsunami heights from these scenarios are included in our database. This is a completely new philosophy for DTHA, focused on MCS_h instead of MCS_p.*

*We propose to modify section 3.3 as follow in order to better focus on the methodological goal of the paper:*

*"Even if the zonation adopted in our study is not in agreement with the actual knowledge of the deep structure of the Gulf of Cadiz and the Gloria Fault zone as given by Martínez-Loriente et al. (2014) and Batista et al. (2017), such a simplified zonation represents a first order approach of the AGPB seismotectonics, consistent with the global objective of the study being to develop a methodological way of quantifying all the possible tsunami heights impacting a given area. A more accurate consideration of the known fault parameters of the AGPB area would be more in line with the work presented by Roshan at al. (2016) or with a complete and robust deterministic MCS_p approach."*

**C4.** The authors consider that the maximum magnitude allowed for the "oceanic" domain is 8.3, being 9.3 for the "continental" domain. There is no explanation whatsoever for the arbitrary attribution of these values. There is no reference in the literature for this area that proposes such a high value of magnitude. It is in contradiction with the known studies for THA and Seismic Hazard Assessment (SHA) and it should be properly explained. The highest magnitude considered for the AGPB is the one estimated for the 1st November 1755 earthquake that, at most, as an inferred magnitude of 8.8 (e.g. Johnston, 1996). The strong discrepancy between the two zones as regards the maximum magnitude is not explained in the text.

*Magnitudes are computed from scaling relationships (section 2.1).The magnitude 9.3, for example , results from considering a length of around 575 km, a width of 290 km (dip of 11 degrees), a slip of 23 m, a depth of 30 km and a shear modulus of 30 GPA. This is an extreme scenario, without a doubt. In the database there are 133 scenarios above M9, representing less than 0.2% of the total MCS_p explored. Our choice is to not ignore these very unlikely scenarios. In fact, the objective of our work is to propose a methodology which could be applied in a context of poor knowledge and could permit to ensure that the chosen MCS_p is "really" the appropriate scenario to consider for a location,. Depending on the specific hazard target (civil or industrial facilities), and its location respect to the source zone, the end-user of this methodology will need to decide which level of water height to choose from the obtained distribution (MCS_h). In the paper, we suggest some analysis for support the end-user in its choice (in section 4).*

*We propose to modify section 3.3 of the revised accordingly to better clarify this point.*

**C5.** In fact, given the uncertainty today on the probable source for the 1755 earthquake, the authors recognize that the "oceanic" database does not contain one of the proposed sources for this event, the Gorringe Bank. The authors try to mitigate this problem by adding an ad-hoc extra database of scenarios for that particular source, thus violating the basic principle in their origin,

make a random choice of parameters that would include all possible sources of extreme events. The "oceanic" database clearly fails to attain this objective.

*We think that there may be a misunderstanding on this specific point.*

*The two "deterministic scenarios" focus on the 1755 event and present a maximum magnitude of 8.9, close to the Johnston (1996) value. These scenarios where designed in order to perform a comparison with the global database as stated in section 4.3 "it is of interest to quantify the relevance of the major known tsunami event of this zone with respect to the numerical database constructed through uncertainty quantification for the specific sites".*

*In section 4.3 of the revised paper we specify that:*

*"These scenarios have been chosen because they are representatives of the MCS_p approach for the study zone"*

*Then, in the conclusion of the paper we underline that:*

*"The data base was also tested through a comparison with the tsunamis generated by a Lisbon-like 1755 earthquake (paragraph 4.3). The latter representing a comparison between the global results from UQ (MCS_h) and a more classical MCS_p approach. Globally, the hazard level issued by the total database seems robust and representative of the tsunamis from Gorringe and Horseshoe banks once the uncertainty of numerical results is considered (see Figure 10), at least for French gauges off the Atlantic coast. Even if this work is mainly methodological, this results seem to confirm that the proposed MCS_h is adapted to the French Atlantic Coast as the tsunamis height from these scenarios are the strongest of the tsunamis height frequency distribution from the global data-base. However, to be conclusive, the same analysis should be conducted with other MCS_p hypothesis for the AGPB. Moreover, it must also be noted that the choice of the hazard level is arbitrary and it depends on the chosen area, the available information about potential tsunami sources and the analysis of the tsunamis height distribution."*

**C6.** The heterogeneity of the AGPB has been captured by all SHA and THA studies made in the area, most of them not considered in the paper, e.g. Vilanova and Fonseca (2007), Woessner et al. (2015) for SHA or Omira et al. (2015), Omira et al. (2016), Matias et al. (2013) for THA. The criteria used in these works take into consideration the best knowledge on seismology and seismotectonics, the same considerations that should guide the development of the tsunamigenic databases used in the paper. Instead the authors choose to build random scenarios over an all-encompassing range of source parameters. True, the geological knowledge of the area is still incomplete and there are uncertainties or disagreements, for example, on the identification of the source for the 1755 event. This fact, however, does not justify the replacement of an "educated" database supported by expert opinions by a purely random source distribution.

*We modify the last section of the revised paper to introduce this answer:*

*"Moreover, the authors consider that for an operational application of the proposed methodology to locations closer to the source zones (i.e. Portugal or Spain, Marocco), the sensitivity analysis will probably show that slip, length and width are the dominant parameters. In such cases, a more in-depth source analysis is necessary and the research performed in the region should be carefully considered (i.e. Vilanova and Fonseca, 2007; Woessner et al., 2015; Matias et al., 2013; Omira et al., 2016; Omira et al., 2015). However, uncertainties will always remain. Thus even in such cases, where more refined databases need to be established, the proposed approach should be of interest. In fact, as shown by the Fukushima accident, it is today necessary to go beyond the classical MCS_p approach and to focus on all the possible tsunami heights at a given location (MCS_h). In this sense, we consider that the modelled tsunamis heights of our data-base are*

*probably already representatives of the tsunamis height generated by these scenarios for the French Atlantic Coast."*

*Moreover, as reported in the introduction this work is only methodological and relies on tsunamis generated with a rough numerical model. For a real tsunamis hazard assessment, which will be the objective of a next research, the authors aim to explore the AGPB with the same methodology proposed in this study, but by introducing some essential modification:*

- *the use of a refined numerical model, in order to take into account the coastal effects and the run-up;*
- *the construction of a numerical data-base with more detailed geological hypothesis;*
- *the validation of meta-models with historical data from the 1755 Lisbon tsunami*

*This work will then allow the assessment of tsunami hazard that goes beyond the classical DTHA by introducing uncertainties.*

**C7.** As regards the source of the largest historical earthquake and tsunami reported for the area, the 1st November 1755 event, the authors identify the difficulty to find in the literature a consensual solution. Some of the more relevant references to that problem were not considered in the paper (in chronological order here), Baptista et al. (2003), Terrinha et al. (2003), Gracia et al. (2003), Barkan et al. (2009) and Cunha et al. (2010.). These and other studies show how artificial is the split between "oceanic" and "continental" source areas with very different maximum magnitudes (8.3 and 9.3 respectively). The derived databases cannot be considered as representative or as including the maximum credible tsunamigenic sources in the AGPB as claimed.

*We do not agree with this consideration, even if we recognize that the original version of the paper does not clearly present the objectives of the study and the differences between our methodology and the classical DTHA based on MCS_p. In fact, as now reported in the introduction of the revised paper, we focused on the impact of tsunamis (tsunamis height distribution) along the French coast and not on the tsunami sources, and we chose to cover all the possible tsunami scenarios from AGPB by sampling beyond the acknowledged limits of the seismic source parameters. Nevertheless, as reported in section 4.3, and figures 7 and 9, the impact of MCS_p from literature is actually covered by the numerical data-base along the French Atlantic Coast.*

*The mentioned references are integrated in the text of the revised paper.*

**C8.** Another shortcoming in building the databases is the absence of an explanation on the way that the parameters length, width, slip and magnitude are explored, since they must be related by the equations (15) and (16). Equation (15) also depends on the value attributed to the shear modulus in the equation, but no mention to that is found in the paper. Contrary to what is inferred from the text and Table 5, the magnitude is not uniformly sampled between the limits indicated. This is clear from the histogram presented in figure 9a. A clear explanation on the procedure used to relate these 5 parameters must be clearly presented, as for example is done for the AGPB by Matias et al. (2013) and Baptista et al. (2017). If other scaling laws are used in a similar process these should be clearly discussed in the paper.

*We think that there may be a misunderstanding of this aspect of the paper: magnitude is a computed quantity, whereas length, slip etc. are uniformly sampled. The shear modulus is constant for our tsunami simulations and assumed to be equal to 30 GPA. The value of the shear modulus is now in the text. In future work this value could also be explored as well as the scaling relationship used to compute magnitude.*

**C9.** As I understand, the purpose of the adopted methodology was "to build a numerical database of a wide bunch of the maximum possible tsunamis generated by the AGFZ". It is demonstrated in the

paper that the most influential source parameter to the maximum water height observed at the cost is the average slip in the fault, that scales with magnitude, if earthquake physics are respected. Then we should expect that only the largest magnitude events would contribute to the desired hazard parameter. It is then hard to understand what is the contribution to the proposed investigation of events with magnitude < 8.0 that constitute 50% of the database (figure 9a).

*The analysis reported in the section 4 of the submitted (and revised) paper is an example of how an end-user of our methodology should analyze the numerical results to derive MCS_h from UQ performed with meta-models. At first, from the distribution of tsunami heights it is necessary to choose the hazard level "MCS_h" (the mean + 2 sigma value in our paper). Then, it is of interest to understand which scenarios are stronger than the chosen hazard level. For the French Atlantic Gauges, it is reported in the text that*

*"Not surprisingly, the strongest tsunami heights correspond to the strongest seismic events of the database (Figure 9a). In particular, more than 80% of the tsunami heights higher than the chosen hazard level ("$\mu_m + 2\sigma_m$") are associated with a seismic magnitude higher than 8.5. These tsunami events completely represent the tail of the earthquake magnitude distribution (the grey histogram in the Figure 9a). Nevertheless, some discrepancies are observed between the results of the southern gauges (La Rochelle and Gastes) and the results obtained for the northern gauges (Brest and Saint-Malo). In fact, even if it seems that only the very strong earthquakes (magnitude higher than 8.5) can generate a tsunamis height higher than "$\mu_m + 2\sigma_m$" for the middle-south of Atlantic French coast (more than the 95% of the tsunamis scenarios of the sample), this is more ambiguous for the northern Gauges. In this case, nearly 30% of the tsunamis higher than the hazard level can be generated by an earthquake of magnitude in the range [7.8 – 8.5], which includes the upper range of the oceanic shelf."*

*In this sense, the objective of Figure 9 is to underline that the strongest water height (stronger than the proposed hazard level MCS_h) are not all associated to the strongest magnitudes, generally associated to the MCS_p scenarios. This is the kind of interesting result which can be obtained through UQ, and we want to underline it. Moreover, this figure also shows that for "lower" magnitudes, most of the tsunami heights could be generated by the oceanic crust (western domain of the revised paper). Thus, the location of the fault is an influent parameter for the French Atlantic Coast.*

**C10.** Furthermore, the authors recognize that most of the hazard is generated by "well located-oriented" faults. Since the authors explore randomly the full source parameter range, it is difficult to be sure that these "well located-oriented" faults are caught by the databases, when a small set of sources have the largest magnitudes tested. If we sample the strike every 30°, the dip every 10°, the rake every 30° and the epicentre coordinates every 1°, the depth every 5 km, we arrive at ~400,000 possible sources for each magnitude level. If we allow for some variability in length, width and slip for each magnitude level, then more than 106 scenarios are possible. With only a few thousand scenarios randomly selected we should not expect to find the "well located-oriented" faults that are responsible for the highest water level at the coast.

*We do not agree with this remark and we think that there may be a misunderstanding concerning our analysis and the methodology we proposed. We then tried to be clearer throughout the text.*

*Especially, in section 4 of the paper we analyze MCS_h. This analysis is mainly focused on three points:*

- *the choice of the hazard level and the analysis of the strongest scenarios of MCS_h (above the proposed hazard level);*
- *the comparison with MCS_p results*
- *the sensibility analysis using Jansen's method*

*The analysis of the strongest scenarios (stronger water height at a given location of the French Atlantic Coast) demonstrates that "oceanic shelf" seismic sources with relatively "low" magnitudes (between 7.8 and 8.2) actually contribute to the strongest tsunami heights generated in the meta-model. Hence our conclusion, that the location of seismic sources is relevant for MCS_h along the French Atlantic Coast. In other words, the French Atlantic coast is in the shadow of the tsunamis generated by the stronger continental shelf sources and more sensitive to the tsunamis generated in the oceanic shelf.*

*Then, the authors focused on sensibility analysis using the Jansen's method which permits to explore the space of the input parameters in order to assess their influence on model results. This sensibility is estimated through the Sobol indices which are a measure of how the variance of each parameter influence the global variance of the variable of interest (in this study, the maximum tsunami water height). Concretely, for the evaluation of Sobol indices through the Jansen's method, other evaluations of the meta-model (and thus, other simulations) are evaluated. The space of input parameters is sampled in an adapted way in order to compute correctly the Sobol's indices and the convergence is checked (we verify that the number of evaluation used for the sobol indices computation is sufficient to obtain a convergent value of the Sobol index). This is managed by the R-package sensitivity cited in the paper. From these results, the authors conclude that a "well located-oriented fault" can be relevant for the North Atlantic Gauges, even if the slip is still the more relevant parameter.*

*Moreover, the author underline that these results are consistent with previous results from Allgeyer et al. (2013), indicating that some areas along the French Atlantic Coast are more exposed to tsunamis from AGFZ.*

**C11.** As regards the depth of the tsunamigenic sources (presumably below the sea bottom, but not clear) the authors mention that "the maximum oceanic depth is considered to be of the order of 10 km whereas the continental crust can attain 30 km depth". This consideration is translated into the values provided in Table 5. The authors thus disregard the evidence that recent microseismicity investigations in the area show that the brittle lithosphere extends at least to 50 km depth (Geissler et al., 2010, Grevemeyer et al., 2016, Grevemeyer et al., 2017, Silva et al., 2017). The large focal depth found in the area is interpreted as representing the rheological properties of very old oceanic lithosphere that forms most of the AGPB. This rheology as consequences for the scaling laws appropriate for the generation of large tsunamis in this domain as discussed in Matias et al., 2013 and Baptista et al., 2017. The paper does not explain also how the conflict between a shallow focus and a large vertical width is solved when selected by random?

*Considering the thickness, we clearly made a mistake in the text, considering that the depth in the mode is the middle of the fault plane (as expressed in the figure caption 2). So that the thickness is in fact twice the thickness reported in the text, more in line with the available observations. The text is modified accordingly. This may also help understanding the last point mentioned, considering that we don't explore the depth of the crust but the depth of the middle of the faulting scenario (in between 5 and 10km (depth 10 to 20) for the western part, and 5 and 30km (depth 10 to 60) for the eastern part.*

*Concerning the rheology, it is clear that the rheology will influence the generation of large tsunamis, but this parameter was not taken into account in this study, considering a homogeneous standard value of 30 Gpa for mu.*

**C12.** The AGPB is an area prone to tsunamis and so the investigation of the resulting tsunami hazard is an important objective. However, the authors fail to recognize the synthesis tsunami catalogues already published for the area, while discussing the merits of the built database (Baptista et al., 2009 and Kaabouben et al., 2009).

*We think that the precisions concerning both the philosophy and the objectives of the paper as mentioned in the other answers to Dr. Luis Matias may be sufficient to answer to this specific point.*

*The mentioned references were considered in the revised paper to complete the bibliographic review.*

**C13.** About the methodology, as I understand it, one "meta-model" must be built for each of the desired outputs to be investigated. In the paper 4 tide gauge locations where chosen for testing and only one hazard parameter elected, the maximum tsunami water height. Nowadays, following the development of the tsunami warning systems, both national and regional, THA studies, deterministic or probabilistic, are applied to very close observation points at the coast, typically a few tens of km, to account for the high variability in tsunami propagation and coastal effects (e.g. Molinari et al., 2016, Roshan et al., 2016). Nowadays, to mitigate the massive computing power required to generate the large number of tsunami scenarios required for PTHA, authors have resorted to parallel computing using GPU and to the concept of Elementary Source or Empirical Green Function (e.g. Molinari et al., 2016 or Baptista et al. (2017). Applying the "uncertainty quantification technique" to reproduce the results that are nowadays required, with an extensive set of observation points (or forecast points) may be as computer intensive as the traditional approach of tsunami propagation for a large number of scenarios. The paper fails to present the advantages of the methodology regarding a realistic application.

*Thank you for this suggestion. We modified the introduction in order to present these recently published researches describing the use of meta-models in the field of tsunamis. Moreover, we also introduced a paragraph in the Conclusions of the revised paper to address the use of meta-models in THA:*

*"The meta-models are first constructed by a uniform exploration of the space of the inputs parameters of the seismic area and then tested and validated through a series of statistical indicators (correlation coefficient, residuals analysis, MSE). These performance-tests allowed us to conclude that, even if the maximum residuals (differences between model and meta-model results) are very high for the four selected locations along the French Atlantic Coast (see Table 5), the meta-models are able to reproduce the model behavior and they can be used for the construction of the numerical tsunami database. Moreover, the realism of the tsunamis height simulated with the meta-models was also ensured through a comparison with the numerical results presented in Allgeyer et al. (2013) for the French Atlantic Coast. Thus, the constructed meta-models could be employed in a further study for roughly evaluating the impact of other seismic scenarios from the AGPB and impacting the French Atlantic Coast. In the example of application presented in this work we built our meta-models using a design data-base of nearly 5 000 tsunami scenarios. The number of simulations is large and required intensive parallel computing using GPU. However, considering the large number of model evaluations necessary for UQ (more than 50 000) and GSA (more than 20 000), the meta-models appear as the more adapted tool for the proposed methodology."*

*"As perspective, the example presented in this study for the AGPB suggests that it could be of interest to introduce the meta-models in the systems developed for tsunami early warning, considering the low computational time inherent to this statistical tool"*

**C14.** The paper applies the "uncertainty quantification technique" to make a sensitivity analysis aiming to identify the source model parameters most relevant for each of the four target sites. A general classification on the most relevant parameters is given and a simple interpretation is presented. However, a more detailed analysis would be much more relevant. What are the epicentres and magnitudes that contribute most to the highest hazard levels in each site? This is a common exercise in Probabilistic Seismic Hazard Assessment called disaggregation. My expectation

is that this analysis would show that the most relevant contributions would come from unrealistic geological sources, even taking into consideration the uncertainties ascribed to the seismotectonic framework of the AGPB.

*In this study we do not ask the question "What are the epicentres and magnitudes that contribute most to the highest hazard levels in each site? ". Because it is a Deterministic approach that we are proposing. In our approach, we propose to retain for the French site a "+2sigma" value of the mean maximum water height distribution as hazard level (MCS_h). This is an arbitrary choice which seems robust for the selected French Atlantic Coast. For this value of hazard the sources contributing are located (see figure below):*

- *all over the explored space but preferentially in the continental part for the "Saint-Malo" and the "Brest" gauges,*
- *Exclusively in the continental part for the southern gauges "La Rochelle" and "Gastes".*

*The magnitudes are in the 8 to 9 range for all the gauges (see Figure below) with a mean of 8,47 for the northern gauges and of 8,67 for the southern gauges.*

*The authors recognize that this point is interesting and that without a complete disaggregation, as proposed by Dr. Luis Matias, it should not be possible to assess tsunami hazard. However, we consider that it is not necessary in this paper to make a deeper investigation of our results as we propose a simple illustration of a methodology.*

*In the revised version of the paper, we propose to add this simplified analysis in section 4.1:*

*"As reported in Figure 8, the proposed hazard level is associated to earthquake magnitudes in the 8 to 9 range for all the gauges, with a mean of 8,47 for the northern gauges and of 8,67 for the southern gauges. This value of the hazard level corresponds to nearly 200 tsunamis scenarios (+/- 0,01 m respect to the chosen value) of the data-base, located all over the AGPB but preferentially in the continental part for the "Saint-Malo" and the "Brest" gauges and exclusively in the continental part for the southern gauges for "La Rochelle" and "Gastes" gauges."*

[Figure]

*Figure 8: Location of the seismic sources associated to the proposed tsunami hazard "MCS_h" for the French Atlantic Gauges (m+2sigma). The magnitude range associated to these tsunami scenarios varies between 7.5 and 9.*

*A real application will be the objective of a further work, which will be focused on:*

- *a detailed numerical model of the French Atlantic Coast and of some coastal areas of Portugal, Spain and England with historical records of 1755-event;*
- *the construction of a numerical data-base with the methodology exposed in this paper;*
- *the test of geological hypothesis for the zone with the meta-models and the comparison of the data base with all the MCS_p from literature;*
- *the disaggregation of strongest scenarios from the data-base to verify if most relevant contributions would come from unrealistic geological sources and to check the sources associated to the most impacting scenarios for different zones.*

**C15.** The authors use systematically the moment statistics of gaussian distribution to characterize the several variables discussed in the text. However, the authors recognize that many of these quantities have long tailed distributions, as expressed in table 4 or in figure 9a. The gaussian distribution doesn't seem the most appropriate model for the distribution of extreme values as desired. This question should be properly addressed in a more formal way in the paper.

*The Gaussian distribution of water height is the results of UQ. The tsunamis height have long tail distributions because of the large range of variation of the input parameters. We discuss this point in section 4 when we discuss the hazard level to retain from UQ.*

**C16.** The authors claim to obtain maximum water height estimates for the 4 locations identified in the text that are close to the coast line. The representative water depths at those points are 8, 15, 28 and 40 m. It is well known that the shallow water equations used for tsunami propagation loose their precision close to the coast and at water depths lower than 50 m. It is usually recommended that shallow water results should stop at that water depth (e.g Kamigaichi, 2009). From that point higher resolution models or empirical relationships should be used to estimate wave heights at the coast or shallower waters (ibid.). This issue should be addressed in the paper.

*We do not agree with Dr. Luis Matias concerning this specific point. In our simulations we use the non-linear Boussinesq equation which is adapted to the simulation of tsunami propagation near to the coast-line and also for the simulation of tsunami run-up and propagation in the inundated areas, as shown in literature (i.e Poisson et al., 2009; Allgeyer et al., 2013). Moreover, in the framework of the French research project TANDEM dedicated to tsunami modelling, a series of benchmarks were set up, addressing the various stages of a tsunami event: generation, propagation, run-up and inundation (Kazolea et al., 2017). The authors present the results of five codes, involving both depth-averaged Boussinesq and fully 3D Navier-Stokes equations, aimed at being applicable to tsunami modelling. According to the authors, the five codes can be classified in two main categories: those based on the Navier-Stokes (NS) model (Thesis and EOLE) and the depth-averaged Boussinesq-type (BT) models (TUCwave, SLOWS, FUNWAVE-TVD). The authors conclude that both categories provide very similar and satisfactory results on the whole (according to experiment) test-case.*

*For the authors, the main problem of the constructed numerical model is the grid resolution employed for tsunami simulations which does not permit to correctly simulate the tsunamis height and their propagation in the vicinity of the coast. This is the main reason which led us to "only" test a new methodology and not to really assess CMS_h. This point is addressed in section 2.1.*

**Other comments**

*The ensemble of these points is also considered in the revised paper.*

The mathematical notation needs to be carefully revised. In particular vectors/matrices and scalars should be clearly identified, for example using bold type. The meaning of variable names and its capitalization should be kept throughout the paper.

The mention of bibliographic references should follow the journal standards and double "(())" should be avoided.

One disadvantage of the "meta-model" approach is that it adds an extra layer of uncertainty to the uncertainties already due to the source parameter space variability. This issue is addressed in the paper, but it may ne undesirable in practical applications.

The list of references is clearly incomplete for many of the subjects addressed.

The numbers on the tables do not follow the order they are referenced in the text.

The numbers on the figures do not follow the order they are referenced in the text.

The paper quotes extensively the Antoshchenkova et al. (2016) work that corresponds to an abstract to a conference. This should be avoided by presenting the main results of that work in the paper or as a supplement, otherwise many sentences in the paper cannot be verified by the reader.

*The citations to Antoshchenkova et al., 2016 and Imbert et al., 2015 were delated as not available in open access literature. Instead, we present only the hypothesis employed for the construction of the design data-base, which is based on Monte-Carlo sampling of the inputs parameters reported in Table 2 (of the revised paper) and also introduce the numerical domain (Figure 2 of the revised paper) and the numerical model used for the simulations of the design data-base (sec 2.1 of the revised paper). As reported in the revised paper, the equations associated to tsunami generation and propagation are reported in (Allgeyer et al., 2013).*

**Bibliography**

Allgeyer, S., Daubord, C., Hébert, H., Loevenbruck, A., Schindelé, F., and Madariaga, R.: Could a 1755-like tsunami reach the French Atlantic coastline? Constraints from twentieth century observations and numerical modeling, Pure and Applied Geophysics, 170, 1415-1431, 2013.

Batista, L., Hübscher, C., Terrinha, P., Matias, L., Afilhado, A., and Lüdmann, T.: Crustal structure of the Eurasia–Africa plate boundary across the Gloria Fault, North Atlantic Ocean, Geophysical Journal International, 209, 713-729, 2017.

Kazolea, M., Filippini, A., Ricchiuto, M., Abadie, S., Medina, M. M., Morichon, D., Journeau, C., Marcer, R., Pons, K., and LeRoy, S.: Wave propagation, breaking, and overtopping on a 2D reef: A comparative evaluation of numerical codes for tsunami modelling, European Journal of Mechanics-B/Fluids, 2017.

Martínez-Loriente, S., Sallarès, V., Gràcia, E., Bartolome, R., Dañobeitia, J. J., and Zitellini, N.: Seismic and gravity constraints on the nature of the basement in the Africa-Eurasia plate boundary: New insights for the geodynamic evolution of the SW Iberian margin, Journal of Geophysical Research: Solid Earth, 119, 127-149, 2014.

Saito, T., Ito, Y., Inazu, D., and Hino, R.: Tsunami source of the 2011 Tohoku-Oki earthquake, Japan: Inversion analysis based on dispersive tsunami simulations, Geophysical Research Letters, 38, 2011.

Saltelli, A., Chan, K., and Scott, E. M.: Sensitivity analysis, Wiley New York, 2000.

Saltelli, A., Ratto, M., Andres, T., Campolongo, F., Cariboni, J., Gatelli, D., Saisana, M., and Tarantola, S.: Global sensitivity analysis: the primer, John Wiley & Sons, 2008.

**Development of a methodological framework for the assessment of seismic induced tsunami hazard through uncertainty quantification: the test case of the Azores-Gibraltar Plate Boundary**

Vito Bacchi[1], Hervé Jomard[1], Oona Scotti[1], Ekaterina Antoshchenkova[1], Lise Bardet[1], Claire-Marie Duluc[1], Hélène Hebert[2]

[1]Institute for Radiological Protection and Nuclear Safety (IRSN), Fonteany-aux-Roses, 92262, France
[2]CEA, DAM, DIF, 91297 Arpajon Cedex, France

*Correspondence to*: Vito Bacchi (vito.bacchi@irsn.fr)

**Abstract.** The aim of this study is to propose a new methodology for deterministic tsunami hazard assessment based on the analysis of all the possible tsunami heights at a given location. With this new approach, the hazard level is evaluated through a numerical database constructed using "uncertainty quantification (UQ) techniques" which allows the exploration of the tsunamigenic potential of a seismic zone in a rigorous and complete way.. This concept goes beyond the definition of the Maximum Credible Scenario based on the definition of particular source parameters and classically reported in the literature. The proposed methodology relies on the construction and validation of "emulators", or "meta-models", that drastically reduce the computational time necessary for tsunami simulations. It is the first time, to our knowledge, that meta-models are used in this way. To test the methodology, a numerical database of nearly 50,000 tsunamis scenarios generated by the Azores-Gibraltar Plate Boundary (AGPB) and that may potentially impact the French Atlantic Coast was constructed. Tsunami heights distributions resulting from the UQ using meta-models are then discussed to illustrate the advantages of such an approach in decision-making compared to results which can be obtained with a more classical scenario-based approach. In particular, the results suggest that the most influential parameters controlling tsunami heights generated in the AGPB is not only the magnitude but also the location of the impacted zone. It must be underlined that results from this study are used to illustrate the general methodology through a case study with simplified hypothesis and should not be considered for an operational assessment of tsunami hazard along the Atlantic Coast.

**1 Introduction**

Tsunami hazard science developed intensively about 50 years ago (Zoback et al., 2013) following four complementary approaches: the (i) identification of past tsunamis in the geologic record on land, the (ii) characterization of tsunamis of different source types, the (iii) hydrodynamic modelling to predict run-up and inundation areas and the (iv) ocean buoy systems for tsunamis detection and recording. Even if it is generally accepted that large-scale tsunamis can be

generated by three main types of geologic events, namely the earthquakes, the submarine and subaerial landslides, and a variety of mechanisms associated with volcanism, in this study we only focused on tsunamis generated by earthquakes.

Earthquakes are the most common source of large-scale tsunamis, with dip-slip earthquakes (i.e. with vertical movement) being more tsunamigenic than strike-slip earthquakes (i.e. with horizontal movement). It is generally observed that only large magnitude earthquakes (M>6.5) lead to widely observable tsunamis, because earthquakes need to be large enough to be able to produce significant displacements at the sea bottom surface (Wells and Coppersmith, 1994). The typical tectonic environment able to generate tsunamigenic earthquakes is the subduction zone where major tsunami earthquakes occur on shallow inter-plate thrusts. However, they may also be generated by outer rise earthquakes, either within the subducting slab or the overlying crust (Satake and Tanioka, 1999). Other oceanic converging boundaries may also be responsible for tsunamis even if not classified as subduction zones, as the North-African coasts for instance Álvarez-Gómez et al. (2011). On the opposite, diverging plate boundaries are also able to generate tsunamis although earthquakes are in general lower in magnitudes in comparison to subduction zones (Ambraseys and Synolakis, 2010). Finally, purely strike-slip earthquakes may also affect oceanic floors, but they are in general not able to generate major tsunamis, unless a significant earthquake vertical component exists or if steep slopes are affected (Tanioka and Satake, 1996).

Two approaches are widely used by the scientific community to quantify tsunami hazard assessment: the scenario-based (or deterministic - DTHA) and the probabilistic approaches (Omira et al., 2016). The probabilistic approach (PTHA) relies on four basic steps (Geist and Lynett, 2014): (i) the specification of source parameters, including source probabilities for all relevant tsunami sources, (ii) the choice of a probability model that describes source occurrence in time (most often Poisson), (iii) the hydrodynamic modelling for each source location and set of source parameters to compute tsunami hazard characteristics at a site (or at a point) of interest and (iv) the aggregation of the modelling results and incorporation of uncertainty. PTHA calculates the likelihood of tsunami impact using a large number of possible tsunami sources and including the contribution of small and large events from both near- and far-field (Omira et al., 2016). The objectives of PTHA are to condense the complexity and the variability of tsunamis into a manageable set of parameters and to provide a synopsis of the tsunami hazard along entire coastlines in order to help identifying vulnerable locations along the coast and specific tsunamis source regions to which these vulnerable locations on the coastline are sensitive (Geist and Parsons, 2006). Early studies are based on different assumptions (i.e. Rikitake and Aida, 1998; Downes and Stirling, 2001) and a surge of PTHA studies started in the early 2000s up to present (i.e. Geist and Parsons, 2006; Annaka et al., 2007; Geist et al., 2009; González et al., 2009; Blaser et al., 2012), in particular after the major tsunamis in Indonesia in 2004 and in Japan in 2011.

In seismic areas with no (or poor) knowledge of the faults and crustal characteristics it is not trivial to associate a probability distribution to the seismic inputs parameters and DTHA may be more adapted than the probabilistic method for the assessment of tsunami hazard. A scenario-based (or DTHA) approach classically relies on the study of "maximum credible scenarios" (MCS). The tsunami hazard is assessed by means of considering particular source scenarios (the MCS with respect to the actual knowledge of the tsunamigenic seismic sources) and then estimating the coastal tsunamis impact through numerical modelling (JSCE, 2002;Lynett et al., 2016), without addressing the likelihood of occurrence of such a big

event (Omira et al., 2016). The outputs of the deterministic analysis are, in general, tsunami travel time, wave height, flow depth, run-up, and current velocity maps corresponding to the chosen scenario (Omira et al., 2016). However, DTHA could be hampered by the use of specific values of input parameters which may be subjective depending on the person or group carrying out the analysis (Roshan et al., 2016). A good example is the 1755 Lisbon tsunami, generated by an earthquake in the Azores Gibraltar Plate Boundary (AGPB). The Great 1755 Lisbon earthquake generated the most historically destructive tsunami near the Portugal coasts (Santos and Koshimura, 2015). Source location and contemporary effects of such tsunami are not precisely identified and several earthquake scenarios have already been published in the literature since last decades (Baptista et al., 1998;Grandin et al., 2007;Gutscher et al., 2006;Horsburgh et al., 2008;Johnston, 1996;Zitellini et al., 1999;Baptista et al., 2003;Terrinha et al., 2003;Gracia et al., 2003;Barkan et al., 2009;Cunha et al., 2010). All of these studies, mainly based on a comparison of several fault hypotheses based on geological properties of the AGPB, show how variable parameters of the seismic source can be, depending on the studied fault and focal mechanisms.

Nowadays, in the classical DTHA, as reported in the cited papers, MCS is mainly focused on seismic source parameters (MCS_p) and not on the tsunamis heights at a given location (MCS_h). The great M 9.0 Tohoku-Oki subduction earthquake of 2011, the largest ever recorded in Japan (Saito et al., 2011), has clearly shown the limitations of the MCS_p approach. In fact, considering the uncertainties related to the characterization of seismic sources, using MCS_p for DTHA may lead to an underestimation of MCS_h in the area of interest. In this context, the objective of our work is to propose a new methodology to go beyond the MCS_p approach and focus more on MCS_h through the evaluation of all the possible tsunamis heights at a given location. The MCS_h approach has the advantage that it accounts for other scenarios, generated by potential seismic sources not associated to the MCS_p but generating tsunamis heights potentially impacting the zone to protect. The proposed methodological framework for the improvement of DTHA is based on classical uncertainty quantification (UQ) techniques (i.e. Saltelli et al., 2000; Saltelli et al., 2008), which are robust statistical tools permitting to largely explore a given seismic area and provide all the possible tsunami heights generated by this area. This approach is a new philosophy for DTHA and can also permit to ensure that MCS_p is the MCS_h scenario for the target area.

In fact, despite the natural complexity of seismic zones prone to tsunamis, in DTHA studies the uncertainties related to model parameters are generally not taken into account in a rigorous and robust way (Lynett et al., 2016;JSCE, 2002), considering the mathematical framework available in the literature (Saltelli et al., 2004;Saltelli et al., 2008;Faivre et al., 2013;Iooss and Lemaître, 2015), and the classical approach to deal with uncertainties consists to perform a limited number of deterministic simulations with penalized values of the seismic sources. For instance, Allgeyer et al. (Allgeyer et al., 2013) modelled the impact of three different scenarios for the 1755 earthquake on the French Atlantic Coast, with a focus on La Rochelle harbour. The authors show that, depending on the source hypothesis and tide conditions, several areas (western part of the island of Re´ and northern coast of the island of Oléron) may have experienced a moderate impact from 0.5 to 1 m for this tsunami event. Recently, Roshan et al. (2016) tried to improve the DTHA procedure detailed in Yanagisawa et al. (2007) in order to better evaluate the effects of the seismic source uncertainties. The authors focused on a possible range of parameters which could produce more robust estimates of hazard instead of using single value of each seismic source

parameter (Roshan et al., 2016). Basically, their methodology is based on the classical steps of an uncertainty propagation study: (i) the identification and characterization of tsunamigenic earthquake sources, (ii) the estimation of rupture parameters and associated parametric variations, (iii) the exploration of a tsunami numerical model and, finally, (iv) the parametric study capturing the uncertainties through the numerical simulations of the tsunami waves associated to different model inputs

5    parameters varying within a given range. The deterministic numerical results are then processed and hazard maps of mean simulated water height are produced to assess the tsunami hazard (Roshan et al., 2016). Finally, the mean simulated water height of the different tsunami scenarios is employed to design flood level due to tsunami along Indian coast (Roshan et al., 2016). The objective of this exploratory and innovating methodological work was not directly the analysis of the benefits of uncertainty quantification on DTHA (Roshan et al., 2016), even if the authors presented an improvement of the classical

10   MCS approach by considering the uncertainties related to a limited number of seismic source parameters (the dip angle, the strike and the source location), leading to a limited number of tsunami scenarios (around 320). In this sense, they were not interested, for instance, to estimate the robustness of the mean simulated tsunami height considering the low number of simulations. Moreover, the sensitivity of model results to the input parameters was not studied (Roshan et al., 2016). Finally, the other seismic source parameters (e.g. the mean rupture length and width, the slip and the magnitude) were considered as

15   constant for each tsunami scenario and their uncertainty was not evaluated, thus strongly limiting the impact of their study with respect to a more classical uncertainty quantification approach.

In this study, because of the computational constraints associated with UQ, the proposed methodology relies mainly on the use of a meta-model, an emulator of the tsunami numerical model, that make it possible to perform a large number of tsunami scenarios with reduced computational time and, consequently, to intensively explore a tsunamigenic area for which

20   geological and geophysical datasets are in general limited. Statistical emulators have already been employed in the field of tsunamis, for the analysis of landslide-generated tsunamis (Sarri et al., 2012) or for the evaluation of the uncertainty in the friction parameterization used in tsunamis simulations (Sraj et al., 2014). More recently, Rohmer et al. (2018) proposed a Bayesian procedure to infer (i.e. learn) the probability distribution of the source parameters of the earthquake, based on the combination of a kriging-based metamodelling technique, to overcome the high computation time cost of the numerical

25   simulator, and an Approximate Bayesian Computation (ABC) procedure, to perform the Bayesian inference. The authors applied the procedure to the Ligurian (North West of Italy) 1887 tsunami case. All the mentioned studies show that it is of interest to use meta-models for tsunamis analysis in many contexts and at least for reducing the computational time necessary for tsunami simulations (Sarri et al., 2012).

In this research work, the constructed meta-models are exploited with the statistical criteria classically employed in

30   UQ studies (Saltelli et al., 2000;Saltelli, 2002;Saltelli et al., 2008;Iooss and Lemaître, 2015) for the assessment of DTHA. To our knowledge, it is the first time that meta-models are tested for the assessment of DTHA. The methodology was hereafter applied to the AGPB area by the construction and the validation of four meta-models able to reproduce the maximum tsunami heights along the French Atlantic Coast. The AGPB area was chosen as an exercise because of the large available literature, because it represents one of the most important sources of potential earthquake-related tsunamis for the French

Atlantic coast (as attested by the impact of the 1755 tsunami along the French coast, http://tsunamis.brgm.fr/) and because the results of our constructed meta-models could be compared with previous deterministic simulations of tsunamis from AGPB along the French Atlantic Coast (Allgeyer et al., 2013).

Finally, in section 4 we present how the numerical data-base constructed through UQ techniques should be employed for DTHA along the French Atlantic Coast and its critical analysis. The analysis is focused on the hazard-level which should be retained from UQ (MCS_h) and on the comparison between the tsunamis data-base with tsunamis heights obtained with a more classical approach (MCS_p). It must be noted that the aim of this study is to show how a numerical database constructed using UQ "techniques" can be a useful tool for the analysis of the tsunamigenic potential of a seismic zone. As a consequence, this paper illustrates a methodology through a case study, and is not to be considered as an operational assessment of tsunami hazard along the French Atlantic Coast.

**2 Presentation of the methodology: theoretical background**

In this study, we propose a general methodological framework for the exploration of the tsunamigenic potential of a given seismic area and the quantification of the associated uncertainties. This methodology should permit to identify all the possible tsunamis heights at a given location through UQ and then to assess a hazard level (MCS_h). It should be suitable in two situations: (i) for poorly known tsunamigenic areas with no (or poor) information related to seismic sources or (ii) to evaluate the robustness of the MCS_p, by ensuring that this scenario is really the more impacting for the area to protect. The proposed approach relies on three main steps, as reported in Figure 1. STEP 1 consists in the construction of a numerical model able to reproduce the tsunami heights generated by a given seismic area and impacting a target area. In STEP 2, the numerical model simulates a regular set of physical tsunamigenic scenarios (so called design data-base) that are used for the construction and the validation of an emulator (the meta-model) able to reproduce the original model results in the target zone. In STEP 3, the validated meta-models are used for DTHA. The UQ performed using meta-models instead of the original model permits to explore intensively the tsunamigenic area with nearly zero computational time. The distribution of tsunami heights resulting from UQ (step 3.1) can then be analysed and discussed in order to define a given hazard level for the zone to protect (CMS_h), as presented in section 4. Finally, in order to identify the model parameters contributing the most to the tsunamis height at a given location a global sensitivity analysis (step 3.2) is performed. The mathematical hypothesis of each step are described in the next paragraphs, with a focus on the AGPB. This methodology is general and can be used in different hydraulic fields.

**2.1 Step 1: numerical tool for tsunami simulations**

The first step consists in the construction of a tsunami numerical model of the area to explore. In this study, the tsunami numerical simulations were performed by using the CEA tsunami code which exploits two models, one for tsunami initialization and the other one for tsunami propagation. The initial seabed deformation caused by an earthquake is generated

with the Okada model (Okada, 1985) and is transmitted instantaneously to the surface of the water. This analytical model satisfies the expression of the seismic moment $M_o$:

$$M_O = \mu \cdot D \cdot L \cdot W \qquad (1)$$

where μ denotes the shear modulus, D [m] the average slip along the rupture of length L [m] and width W [m]. Then, the seismic magnitude "$M_w$" is directly computed through equation 2, as follow:

$$M_W = \frac{2}{3} \cdot \log_{10}(M_O) - 6,07 \qquad (2)$$

5    The following parameters are also required for tsunami initiation: longitude, latitude, and depth [km] of the center of the source, strike [degrees], dip [degrees], and rake [degrees]. A conceptual scheme of the input parameters for the tsunami-code and the numerical domain used in this study are reported in Figure 2. Then, the computation of the tsunami propagation is based on hydrodynamic equations, under the nonlinear shallow water approximation (the Boussinesq equations as reported in Allgeyer et al., 2013). Shallow water equations are discretized using a finite-differences method in space and time

10   (FDTD). Pressure and velocity fields are evaluated on uniform separate grids according to Arakawa's C-grid (Arakawa, 1972). Partial derivatives are approximated using upwind finite-differences (Mader, 2004). Time integration is performed using the iterated Crank-Nicholson scheme. No viscosity terms are taken into account in our simulations. The only parameters of this second model are the bathymetry (space step and depth resolution) and the time step.

In this study, all the simulations were performed on the same bathymetric grid with a space resolution of 2'

15   (~3.6 km). The numerical model was not directly validated by the comparison with similar simulations from literature. Considering the rough bathymetrical grid resolution, the developed numerical model is not adapted to the estimation of the tsunamis run-up and the inundation areas and it can't be used for a real assessment of the tsunami hazard along the French Atlantic Coast. However, this work being methodological, the authors consider that the numerical results are consistent with the objectives of the study. Moreover, the tsunami-code was largely validated through extensive benchmarks in the

20   framework of the work package 1 of the TANDEM research project (Violeau et al., 2016) by ensuring its ability to reproduce tsunamis generation and propagation. As a consequence, the order of magnitude of the tsunami heights computed in this study should be realistic and adapted to the test of the methodology.

Finally, in order to perform the numerical simulations needed for the meta-model construction and validation (see section 2.2), the CEA code was coupled with the IRSN Promethee bench. Promethee is an environment for parametric

25   computation that allows carrying out UQ studies, when coupled (or warped) to a code. This software is freely distributed by IRSN (http://promethee.irsn.org/doku.php) and allows the parameterization with any numerical code and is optimized for intensive computing. Promethee was first linked to the numerical code by means of a set of software links (similar to bash scripts). In this way, numerical simulations were directly lunched by the IRSN environment. Then, the statistical analysis, such as the Monte-Carlo simulations used for the meta-model construction (see section 2.2.1) and UQ (see section 2.3.1) or

the Sobol indices computation (see section 2.3.2) were also driven by Promethee, which integrates R statistical computing environment by permitting this kind of analysis (R Core Team, 2016).

**2.2 Step 2: Meta-model design and validation**

For the evaluation of numerical codes in industrial applications, where the computational code is expensive in computation time, so that it cannot be evaluated intensively (e.g. only several hundred calculations are possible), it is usually not easy to estimate some numerical parameters (i.e. the sensitivity Sobol indices), requiring for each input several hundreds or thousands of evaluations of the model (Iooss and Lemaître, 2015). As a consequence, a meta-model instead of the model is generally used in the estimation procedure. A meta-model is a mathematical model of approximation of the numerical model, built on a learning basis (Fang et al., 2006). The meta-model solution is a current engineering practice also for estimating sensitivity indices (Gratiet et al., 2016). In this study, we use the Gaussian process meta-model (also called kriging) (Sacks et al., 1989), which good predictive capacities have been demonstrated in many practical situations (see Marrel et al., 2008, for example). As a consequence, kriging meta-model will be useful both for sensitivity analysis and for a numerical database construction through uncertainty quantification (as reported in the next section). The classical steps for meta-models construction and validation are reported in various studies (i.e. Faivre et al., 2013;Saltelli, 2002; Saltelli et al., 2008), and can be shortly summarized as follow (see Table 1).

The mathematical construction of the meta-model obviously depends on the chosen meta-model. The theoretical background for kriging meta-model design used in this study is fully detailed in Roustant et al. (2012). The associated equations are implemented in the R-packages DiceKring and DiceOptim and reported in the APPENDIX 1 for the interested reader.

**2.2.1 The initial design for meta-model fitting**

For computer experiments, selecting a *design database* (or space filling design) is a key issue in building an efficient and informative meta-model (Iooss et al., 2010). The design of the experience is the choice of the input parameters (and, consequently, of the initial simulations) to use in order to build the most accurate meta-model at a minimum computational coast (minimum number of model simulations).It can depend on the chosen meta-model (Kleijnen, 2005). For instance, for Gaussian emulators as kriging, one of the most popular design type is the Latin Hypercube Sampling (LHS) (McKay et al., 1979), which offers flexible design size for any number of simulation inputs (Kleijnen, 2005). An equivalent approach, used in this paper, is the Monte-Carlo sampling technique which also permits a regular and uniform exploration of the space of the model input parameters. In fact, except for some numerical tests reported in previous studies (Iooss et al., 2010), at the moment, no theoretical results gives the type of initial design leading to the best fitted meta-model in terms of meta-model predictions.

**2.2.2 The partitioning of the original data base**

It is generally assumed that the simulations performed to build the meta-model (the *design database*) can be partitioned in a *training set* (data used for the estimation of the meta-model) and a *validation set* (simulations performed to test the ability of the meta-model to reproduce the model results with other data). According to Amari et al. (1997), to obtain an optimum unbiased meta-model, nearly 80% of the simulations of the *design database* must be used for the *training set* and 20% for the *validation set*. This ratio is used in our study.

**2.2.3 The validation of the chosen type of meta-model: training set**

In order to assess if a given meta-model (i.e. kriging, gam method, random forest, chaos polynomials, and so on…) is adapted to reproduce the model behaviour for the design data base, probably the simplest and most widely used method is the *cross-validation* or *K-fold* cross-validation method (Hastie et al., 2002). This step is performed with the *training set*. The principle of cross-validation is to split the data into $K$ folds of approximately equal size $A_1 A1,...,A_k AK$. For $k = 1$ to $K$, a model $\hat{Y}^{(-k)}$ is fitted from the data $U_{j \neq k} A_k$ (all the data except the $A_k$ fold) and this model is validated on the fold $A_k$. Given a criterion of quality $L$ as the Mean Square Error (Eq. 3):

$$L = MSE = \frac{1}{n} \sum\nolimits_{i=1}^{n} (\hat{y}_i - y_i)^2 \tag{3}$$

The quantity used for the "evaluation" of the model is computed as follow:

$$L_k = \frac{1}{n/K} \sum_{i \in A_k} L\left( y_i Y^{(-k)}(x_i) \right) \tag{4}$$

Where $\hat{y}_i$ and $y_i$ are, respectively, the meta-model and the model response and $n$ is the number of simulations in the *kth* sample. Finally, the cross-validation used in this study is evaluated through the mean of the quantity $L_k$ computed for each fold:

$$L\_CV = \frac{1}{K} \sum\nolimits_{k=1}^{K} L_k \tag{5}$$

It must be noted that when $K$ is equal to the number of simulations of the training set, the cross-validation method corresponds to the leave-one-out technique (Oliveira, 2008). The methodology employed is described in the DiceEval R-package reference-manual (Dupuy et al., 2015). We considered $K = 10$ and compute the MSE between meta-model approximation and the k-fold observations for each fold ($k$ times). The mean MSE is an index of the mean differences between meta-model predictions and model results, thus representing the ability of the chosen meta-model to reproduce the original model. The more this index approaches zero, the more the meta-model can be considered accurate.

**2.2.4 Quantification of the meta-model uncertainty: residuals analysis**

This last operation is performed on the *validation-set* (composed of the 20% of the *design database*). The objectives are (i) to evaluate the meta-models built on the entire *testing set* (after cross-validation) outside their construction domain (*training set*) and (ii) to estimate the uncertainty related to the use of the meta-model instead of the original model. The first statistical parameter we chose for evaluating the meta-models performance is the correlation coefficient $R^2$ (eq. 6), classically used in statistic:

$$R^2 = 1 - \frac{\sum_{i=1}^{n}(\hat{y}_i - y_i)^2}{\sum_{i=1}^{n}(\hat{y}_i - \bar{y}_i)^2}$$

(6)

Where $\bar{y}_i$ is the mean model response.

Finally, a particular attention was devoted to the analysis of the distribution of the residuals (the difference between model results and meta-model predictions). Especially, the residual plots can help avoiding inadequate meta-models and help adjusting the meta-model for better results. In general, it is assumed that the simplest meta-model that produces random residuals is a good candidate for being a relatively precise and unbiased model. In this study, residuals $\varepsilon_i$ between model $y_i$ and meta-model $\hat{y}_i$ predictions are evaluated as follow:

$$\varepsilon_i = y_i - \hat{y}_i$$

(7)

Two kind of analysis are possible, both performed in this study: (i) the analysis of the distribution of the residuals, which requires residuals to be normally distributed with mean zero (Jarque and Bera, 1980) and (ii) the analysis of the cumulative frequency distribution of the modulus of residuals (reported in the next section). This latter analysis provides an order of magnitude of the total uncertainty related to meta-model prediction, as discussed in the next sections.

**2.3 Step 3: Uncertainty Quantification and Global Sensitivity Analysis**

Considering the variety and the complexity of the geophysical mechanisms involved in tsunami generation, tsunami hazard assessment is generally associated with strong uncertainties (aleatory and epistemic). In PTHA, uncertainties are classically integrated in a rigorous way (Selva et al., 2016;Sørensen et al., 2012;Horspool et al., 2014) and quantified using the logic-tree approach (Horspool et al., 2014) and/or random simulations performed using the Monte-Carlo sampling of probability density functions of geological parameters (Horspool et al., 2014;Sørensen et al., 2012). An alternative and interesting approach is recently proposed by Selva et al. (2016), consisting in the use of an event tree approach and ensemble modelling (Marzocchi et al., 2015). Moreover, a new procedure was recently proposed by Molinari et al. (2016) for the quantification of uncertainties related to the construction of a tsunami data-base based on the quantification of elementary effects.

In this work we propose a classical methodology that could be adapted to analyse tsunamigenic regions with poor (or no) information on crustal characteristics and based on the classical uncertainty study steps (Saltelli et al., 2008;Saltelli et al., 2004;Faivre et al., 2013;Iooss and Lemaître, 2015). This methodology was already tested in other hydraulic context in recent years (Abily et al., 2016;Nguyen et al., 2015). At first, the variation intervals and the probability density functions (PDF) for the inputs affected by uncertainty must be defined for UQ. We remind that, for a random variable X defined on an $[a,b]$ interval, a PDF $f_X(X)$ is defined as follows:

$$P(X \in [a,b]) = \int_a^b f_X(X)dx \tag{8}$$

Most of the time, as in the present study, the probability distributions of random inputs is unknown. Hence, expert's knowledge or the experimental evidence available is used as the basis for the PDF definition. The step 3.1 of our methodology requires propagating in the model the uncertainty associated to the input parameters with the objective of computing the PDF $f_y(Y)$ of the resulting quantities of interest y, as the maximum tsunami height employed in this study.

Finally, a better understanding of uncertainties can be achieved by analysing the contribution of the different sources of uncertainty to the uncertainty of the variables of interest (Step 3.2). Post-treatment procedures are used in order to rank the sources of uncertainty. It is important to note that this last step plays a crucial role. Indeed, the ranking results highlight the variables that truly determine the relevancy of the final results of the study. The methods employed for Step 3.1 and 3.2 are described in the next paragraphs.

**2.3.1 Step 3.1: Uncertainty Propagation using Monte-Carlo method**

The Monte-Carlo method used in this study requires random generation of input variables from their probability distributions. This step is the most complex and computationally demanding. In fact, the number of simulations increases with the increase of the uncertainty of each parameter. For this reason, alternative numerical techniques could also be used to reduce the computational time and yet preserve the quality of the statistical results (i.e. quasi Monte-Carlo screening methods). For instance, in this study, we used kriging meta-models (see section 2.2) instead of the original model to perform UQ using Monte-Carlo simulations.

The resulted sampling of a given size N is a $N \times V$ matrix, V being the number of uncertain parameters. Each row of the matrix $x^i = (x_1,..., x_V)^i$ represents a possible configuration of the input parameters of the model, that are, in this study, the fault parameters presented in Figure 2. Corresponding realizations of the output "Y" (the maximum tsunami water height in this study) are generated by successive deterministic simulations with each configuration of the inputs. Statistical estimators of the response $Y = (Y_1,..., Y_N) = (G(x^i))_{i \in \{1,...,N\}}$ can therefore be computed from the output as follows:

$$E[Y] = \mu_Y = \frac{1}{N} \sum_{i=1}^{N} G(x^i) \tag{9}$$

$$Var(Y) = \frac{1}{N-1} \sum_{i=1}^{N} \left[ G(x^i) - \mu_Y \right]^2 \qquad (10)$$

$$\sigma_Y = \sqrt{Var(Y)} \qquad (11)$$

Where $E[Y] = \mu_Y$, $Var(Y)$ and $\sigma_Y$ are, respectively, the mean, the variance and the standard deviation of the response $Y$ given by the model $G$. These statistical moments are useful for both the uncertainty propagation and the uncertainty sensitivity analysis (see next paragraph). It must be noted that $\sigma_Y$ is classically employed for the quantification of the global uncertainty issued from Monte-Carlo simulations.

5    The convergence order of the Monte-Carlo sampling method is given by the Central Limit Theorem (Faivre et al., 2013) as $O\left( \frac{1}{\sqrt{N}} \right)$, implying that a large amount of simulations are necessary to obtain convergent statistics.

**2.3.2 Step 3.2: Global sensitivity analysis**

The role of the sensitivity analysis is to determine the strength of the relation between a given uncertain input parameters and the model outputs (Saltelli et al., 2004). In this study, we focused our attention on Global Sensitivity Analysis
10   approaches which rely on sampling based methods for uncertainty propagation, willing to fully map the space of possible model predictions from the various model uncertain input parameters and then, allowing to rank the significance of the input parameter uncertainty contribution to the model output variability (Baroni and Tarantola, 2014). The objectives with this approach are mostly to identify the parameter or set of parameters which significantly impact models outputs (Volkova et al., 2008;Iooss et al., 2008).
15   In this study, the GSA approach we focused on is the Sobol index computation, that considers the output hyperspace *(x)* as a function *(Y(x))* and performs a functional decomposition (Iooss and Lemaître, 2015;Iooss, 2011) or a Fourier decomposition (FAST method) of the variance (Saltelli, 2002;Saltelli et al., 1999). With respect to deterministic methods (i.e. Morris, 1991), probabilistic GSA approaches relying on Sobol index computation go one step further, by quantifying the contribution to the output variance of the main effect of each input parameter (Sobol, 2001;Saltelli et al., 1999;Saint-Geours,
20   2012;Sobol, 1993). The definition of Sobol Indices is a result of the ANOVA (ANalysis Of VAriance) variance decomposition. To be able to write the variance decomposition, some mathematical hypothesis are required (see i.e. Saltelli et al., 2008, or Faivre et al., 2013), briefly recalled here.

Let us consider an integrable model $G$ on the domain $A = A_1 \times ... \times A_V$. It exists an unique decomposition of the model:

$$G(Y) = f_0 + f_1(Y_1) + ... + f_m(Y_m) + f_{1,2}(Y_1, X_2) + ... + f_{v-1}(Y_{v-1}, Y_v) + ... + f_{1,...,v}(Y_1, ..., Y_v) \qquad (12)$$

Where all the functions fi are mutually orthogonal, implying that $\langle f_i | f_j \rangle = 0$ if $i \neq j$, with the scalar product defined as follows:

$$\langle f | g \rangle = \int_A f(x) g(x) \pi(x) dX \tag{13}$$

Where $\pi$ is a probability distribution defined on $A$. Under these assumptions, given a set of $V$ independent uncertain parameters $X$, the variance of a response $Y = G(X)$ can be calculated, using the total variance theorem, as follows:

$$Var(Y) = \sum_{i=1}^{V} D_i(Y) + \sum_{i<j}^{V} D_{ij}(Y) + ... + D_{12..V}(Y) \tag{14}$$

5    Where $D_i(Y) = Var[E(Y|X_i)]$ and $D_{ij}(Y) = Var[E(Y|X_i X_j)] - D_i(Y) - D_j(Y)$ and so on for higher order interactions. $E(Y|X_i)$ is the $Y$ conditional expectation with the condition that $X_i$ remains constant. The so-called "Soblol' indices" are obtained as follows:

$$S_i = \frac{D_i(Y)}{Var(Y)}, \quad S_{ij} = \frac{D_{ij}(Y)}{Var(Y)}, \quad ... \tag{15}$$

These indices express the share of variance of $Y$ (the maximum tsunami height) that is due to a given combination of input parameters (the seismic sources reported in Figure 2). For instance, $S_1$ is the first order Sobol index. The number of
10   indices grows in an exponential way with the number $V$ of dimensions. So, for computational time and interpretation reasons, (Homma and Saltelli, 1996) introduced the so-called "total indices" or "total effects" that write as follows:

$$S_{Ti} = S_i + \sum_{i<j} S_{ij} + \sum_{j \neq i, k \neq i, j<k} S_{ijk} + ... = \sum_{l \in \neq i} S_l \tag{16}$$

Where $\neq i$ are all the subsets of $\{1,...,V\}$ including i.

GSA approaches are robust, have a wide range of applicability, and provide accurate sensitivity information for most models (Adetula and Bokov, 2012). Moreover, even if they are theoretically defined for linear mathematical systems, it
15   was demonstrated that they are well suited to be applied with models having nonlinear behaviour and when interaction among parameters occur (Saint-Geours, 2012), as in the present study. For these reasons, these indices were already adopted for the analysis of bi-dimensional hydrodynamic simulations in urban areas (Abily et al., 2016) or of complex coastal models including interactions between waves, current and vegetation (Kalra et al., 2018) and they seem well suited for the present work. For the computation of Sobol' indices, a large variety of methodology are available, as the so-called "extended-FAST"
20   method (Saltelli et al., 1999), already used in previous studies by IRSN (Nguyen et al., 2015). In this study, we used the methodology proposed by Jansen et al. (1994) already implemented in the open source sensitivity-package R (Pujol et al.,

2016). This method estimates first order and total Sobol' indices for all the factors "$v$" at higher total cost of "$v \times (p + 2)$" simulations (Faivre et al., 2013).

**3 Example of application: building a numerical data-base of tsunamis generated by the Azores Gibraltar Plate Boundary (AGPB) and impacting the French Atlantic Coast**

**3.1 Design data-base for meta-models construction**

In this methodological work, we considered a widened AGPB tsunamigenic area as a test case for DTHA. We chose to explore as largely as possible the potential tsunami height along the French Atlantic Coast generated by earthquakes from 34° to 40°N and from 18 to 7°W, encompassing to the East the southern part of Portugal down to Morocco, and reaching the oceanic sea-floor west of the Madeira tore rise (as reported in Figure 3). Because the design database is a learning base for meta-modelling, the range of variation of the inputs parameters (column "*Design database*" in Table 2) need to be large in order to cover both a wide range of earthquake scenarios as well as most of the supposed possible sources of the 1755 Lisbon event and earthquakes coming from the following catalogues: http://www.isc.ac.uk; http://earthquake.usgs.gov; http://www.globalcmt.org; http://www.bo.ingv.it/RCMT/. Thus, if correctly estimated, meta-models will be able to reproduce the model behaviour for a large range of variations of the seismic inputs parameters, including physical scenarios from geological studies of the zone.

In order to build the *design database*, fault parameters as defined in section 2.1 and Table 2 were sampled randomly and independently with the Monte-Carlo method and supposing uniform distributions. The uniform distribution was chosen in order to build meta-models able to reproduce tsunamis heights generated by various tsunamigenic sources with the same accuracy. The resulting earthquake magnitudes are computed using the sampled parameters with equation 2. The shear modulus chosen for the magnitude estimation is a constant value assumed to be equal to 30 GPA. This design database is a matrix which associates to a given combination of fault parameters estimates the maximum simulated water height at each point of the numerical grid and also at four selected locations along the French Atlantic Coast called *gauges* (Table 3 and Figure 3), namely, from North to South, "Saint-Malo", "Brest", "La Rochelle" and "Gastes". The maximum tsunami water height is the relevant parameter when estimating tsunami hazard. The *design database* contains 5839 scenarios split in a *training set* of 4672 scenarios and a *testing set* of 1167 tsunamis scenarios, used for the meta-model validation and the residual analysis. The water height characteristics associated to these scenarios are reported in Table 4.

[revised manuscript text omitted]

**3.2.2 Comparison with Allgeyer et al. (2013) results along the French Atlantic Coast**

Another possible validation of the meta-models consistency is the comparison with a previous study from Allgeyer et al. (2013). This comparison is of interest in order to confirm the ability of the constructed meta-models to reproduce the order of magnitude of the modelled tsunami height at a given location. In their study, the authors analysed the impact of a Lisbon-like tsunami on the French Atlantic Coast through numerical modelling. Especially, the authors focused on the simulated maximum tsunami water height in the North Atlantic associated to three different sources for the 1755 events derived from Johnston (1996), Baptista et al. (2003) and Gutscher et al. (2006), for a total of five tsunami scenarios. The same scenarios were simulated with our meta-models for the four French Atlantic Gauges. Even if the results obtained with meta-models (see Table 6) cannot be really compared with results from the authors (Allgeyer et al., 2013), which are obtained with a more refined grid, spacing from 1' to 0.3" near to the "La Rochelle" harbour, the general pattern reported in Allgeyer et al. (2013) is confirmed. Especially, the authors pointed out that the coastal area offshore French Brittany (from 46 °N to 48 °N) were impacted by amplitudes of 0.6 to 0.8 m for the Johnston source (1996) and not exceeding the 0.5 m for the Baptista et al. (2003) and the Gutscher et al. (2006) sources. These results are consistent with those obtained with our meta-models for the *Brest* and *La Rochelle* gauges which represent this coastal area (Table 6). As a consequence, the results obtained with meta-models by using some physical sources by literature seem reasonable, physically speaking.

**3.3 Construction of a tsunami data-base for the French Atlantic Gauges**

Starting from the validated meta-models we finally build a database of tsunamis generated by the considered AGPB area at four French Atlantic Gauges. This numerical data-base should permit to cover all the possible tsunamis height impacting a given area and then to assess the tsunami hazard level (MCS_h). Moreover, in a context with strong uncertainties on the seismic source parameters, this methodology should also permit to verify if the tsunamis height associated with the MCS_p parameters are robust with respect to the area to protect.

AGPB is a zone prone to tsunamis and its investigation already lead to the publication of tsunami catalogues for the zone (Baptista and Miranda, 2009;Kaabouben et al., 2009). However, for the purpose of this methodological paper, the AGPB area modelled here, which encompass different seismotectonic domains, was split in a very simplistic way into two main seismic source zones. This was done to represent a western domain where normal to transtensive earthquakes mainly occur within an oceanic crust and an eastern domain to the East where reverse to transpressive earthquakes mainly occur on an oceanic crust and a thicker continental crust (Buforn et al., 1988;Molinari and Morelli, 2011;Cunha et al., 2012). As a first order analysis, we considered the *Western domain* west of the 10°W meridian and the *Eastern domain* east of this meridian, coinciding roughly with the base of the continental slope facing the Portuguese coastline. Even if the zonation adopted in our study is not in agreement with the actual knowledge of the deep structure of the Gulf of Cadiz and the Gloria Fault zone as

given by Martínez-Loriente et al. (2014) and Batista et al. (2017), such a simplified zonation represents a first order approach of the AGPB seismotectonics, consistent with the global objective of the study aimed at illustrating a methodological approach to quantifying all possible tsunamis heights that can potentially impact a given area. A more accurate consideration of faults parameters that are better constrained in some regions of the AGPB area would be more in line with the work presented by Roshan at al. (2016) or with a complete and robust deterministic MCS_p approach.

The main fault characteristics considered to perform earthquake scenarios in both eastern and western domains are reported in Table 2, representing our interpretation of data contained in (Buforn et al., 1988;Molinari and Morelli, 2011;Cunha et al., 2012). The considered seismogenic thickness takes into account the depth of the observed seismicity and crustal structure for each zone as well as a part of the upper mantle that can potentially be mobilised during major earthquakes: the Atlantic shelf seismogenic thickness is considered to be of the order of 20 km (after Baptista et al., 2017), and 60 km for the European shelf (after Silva et al., 2017). The fault parameters are considered uniformly distributed and are randomly sampled in their range of variation. We finally filtered the resulting database according to an aspect-ratio criterion, allowing the ratio between length and the width of the faults not to exceed the value of 10, which correspond to an upper bound of what is observed in nature (Mc Calpin, 2009). The final database contains nearly 50,000 tsunami scenarios, resulting in earthquake magnitudes varying from 6.7 to 9.3 (Table 2), depending on the explored earthquake sources characteristics and calculated from Eq 1. This range of magnitudes is consistent with the magnitude range of the *design database*. It must be noted that the higher magnitudes considered for the AGPB (higher than the maximum value estimated for the 1st November 1755 earthquake of 8.8, according to Johnston (1996), correspond to some extreme scenarios which represent less than 0.2% of the total scenarios of the numerical data-base. However, our choice is to not ignore these very unlikely scenarios. In fact, the objective of this work is to propose a methodology which could be applied in a context of poor knowledge and could permit to ensure that MCS_p is "really" the appropriate scenario to consider for a given location. MCS_h is deduced from the tsunamis height distribution at a given location. Depending on the specific hazard target (civil or industrial facilities), and its location respect to the source zone, the end-user of this methodology will need to decide which level of water height to choose from the obtained distribution (MCS_h).

The resulting *global database* (Table 2) is composed by the union of the *western* and the *eastern* databases scenarios. The tsunamis height characteristics associated to the four gauges are reported in Table 7. It must be noted that the maximum tsunami height of the *global database* are lower than the tsunamis height of the *design database* and that the convergence of statistics (mean water height) is largely achieved, as reported in Figure 6. The achievement of the convergence of numerical results is an important aspect of the adopted methodology and suggests that it is not necessary to further explore the space of inputs parameters in order to assess CMS_h: statistically, the distribution of tsunami heights is representative of the source variability in the defined space of parameters.

In conclusion, the built tsunamis database (the *global database*) should represent a wide sample of tsunami heights potentially generated by the *AGPB* (the *Western* and the *Eastern domains*). The database is analysed in the next section with a focus on DTHA. At first, the resulting tsunami height distribution is analysed in order to propose a hazard level for the

French Atlantic Gauges (CMS_h). Then, this choice is discussed in the light of tsunami heights higher than the proposed hazard level (section 4.2) and of tsunamis heights generated by a MCS_p- like scenarios (section 4.3).

**4 Analysis of the numerical tsunamis database**

**4.1 Assessment of a hazard level for the French Atlantic Coast (MCS_h)**

5      The distribution of tsunami heights resulting from each considered tsunamigenic source is reported and compared in Figure 7 for each gauge. It can be observed that the tsunami heights generated by the *western shelf* are globally lower than those generated by other sources. This is not surprising given the different depths explored for the two regions (see Table2). Indeed, the highest tsunamis heights resulting from the global database are globally generated by the *eastern shelf* for both the *southern* and the *northern gauges*.

10     From an engineering point of view, the tsunami hazard "MCS_h" can be directly deducted from Figure 7 (black line). Roshan et al. (2016) proposed to use a hazard reference level based on the mean tsunami height resulting from their distribution. However, as reported in section 3.3, the constructed tsunami data-base integrates all the "likely" and "unlikely" tsunami scenarios from AGPB. As a consequence, we prefer to set the tsunami hazard reference level at the value "$\mu_m+2\sigma_m$" of the modelled water height for each gauge and to consider an uncertainty associated to this level equal to the $+/-2\sigma_r$

15     residual (summarized in Table 8 for each gauge). This choice needs to be justified on a case by case basis. In this specific example the choice is guided, firstly, by considering the shape of the tsunami heights distribution resulting from UQ (Figure 7, black line). In fact, the distributions for each gauge are centred on the lower tsunami heights while the maximum modelled tsunamis heights are very large with respect to the tsunamis statistical parameters of the distribution (reported in Table 8). As a consequence, this choice appears numerically reasonable in a context of DTHA as it considers both the low tsunamis

20     height and the maximum modelled tsunamis height. Secondly, considering the rough approximation of the earthquake input parameters (and the consequently wide range of seismic magnitude distributions) used for the construction of the global database, this hazard level appears more conservative in a context of DTHA.

       The proposed hazard level corresponds to nearly 200 tsunamis scenarios (+/- 0,01 m respect to the chosen value) of the data-base, located all over the AGPB but preferentially in the eastern part for the "Saint-Malo" and the "Brest" gauges

25     and exclusively in the eastern part for the southern gauges for "La Rochelle" and "Gastes" gauges (Figure 8). The MCS_h scenarios are associated with earthquake magnitudes in the 7.5 to 9 range for all the gauges, with a mean of 8.47 for the northern gauges and of 8.67 for the southern gauges. The robustness of this choice is analysed in the next sections. Especially, it is of interest to analyse the strongest tsunamis of the numerical database (see section 4.2) and to evaluate if this hazard level is robust with respect to results which could be obtained with a more classical scenario-based approach

30     "MCS_p" (see section 4.3).

**4.2 Focus on the strongest tsunamis of the numerical database**

As reported in Table 8, there are strong differences, for all the gauges of the numerical database, between the maximum simulated tsunami height and the "$\mu_m + 2\sigma_m$" water height and the hazard level chosen in this study (MCS_h),. As a consequence, from a methodological point of view, and with the objective of assessing the tsunami hazard level from the UQ study, it is of interest to un understand which scenarios are stronger than the proposed hazard level.

With this aim, we first analysed the magnitude distribution "$M_w$" of the tsunami scenarios of the global database (grey curve in Figure 9a) generating tsunamis heights higher than the proposed *hazard level at* each gauge (coloured lines in Figure 9a). Not surprisingly, the strongest tsunami heights correspond to the strongest seismic events of the database (Figure 9a). In particular, more than 80% of the tsunami heights higher than the chosen *hazard level* ("$\mu_m + 2\sigma_m$") are associated with a seismic magnitude higher than *8.5*. These tsunami events completely represent the tail of the earthquake magnitude distribution (the grey histogram in the Figure 9a). Nevertheless, some discrepancies are observed between the results of *the southern* gauges (*La Rochelle* and *Gastes*) and the results obtained for the *northern gauges* (*Brest* and *Saint-Malo*). In fact, even if it seems that only the very strong earthquakes (magnitude higher than *8.5*) can generate a tsunamis height higher than "$\mu_m + 2\sigma_m$" for the middle-south of Atlantic French coast (more than the *95%* of the tsunamis scenarios of the sample), this is more ambiguous for the *northern Gauges*. In this case, nearly *30%* of the tsunamis higher than the *hazard level* can be generated by an earthquake of magnitude in the range [7.8 – 8.5], which includes the upper range of the western shelf.

As shown in Figure 9b, 50% of the simulated tsunamis in this [7.8 – 8.5] magnitude range generating tsunami heights greater than chosen hazard level "$\mu_m + 2\sigma_m$" originate in the western shelf. In fact, the *northern gauges* (*Saint-Malo* and *Brest*) are practically the only ones exposed to tsunamis generated by the western shelf.

These results are interesting and show that the strongest water height obtained through UQ (stronger than MCS_h) are not all associated with the strongest magnitudes, which are typically considered in the MCS_p approach. On the contrary, most of the strongest tsunamis heights at the northern gauges could be generated by the western shelf, for seismic magnitudes in the range [7.8 – 8.5]. In conclusion, these results also suggest that, beyond earthquake magnitudes, the position and the orientation of the faults are influent parameters. In terms of *hazard level*, which is the original objective of this analysis, one can consider, at first, that tsunamis higher than the chosen hazard level are mainly generated by seismic magnitudes stronger than *8.5*, even if some rare combinations of tsunami input parameters can lead to higher tsunami heights.

**4.3 Comparison with a classical MCS_p approach associated to a Lisbon-like 1755 tsunami scenario**

As reported in the introduction, DTHA is classically assessed by means of considering particular source scenarios (usually maximum credible scenario) and the associated maximum tsunami height is generally retained as hazard level (MCS_p). A more sophisticated method was recently proposed by Roshan et al. (2016). The authors tested around 300 tsunamis scenarios in a [8 – 9.5] magnitude range associated with various faults potentially impacting the Indian coast.

Finally, the authors suggested that an appropriate water level for hazard assessment (e.g. mean value or mean plus sigma value) should be retained. They proposed the mean value of the simulated water heights, as test, by considering that this value may need to be revisited in the future.

In the case of the tsunamis hazard associated with the AGPB, the 1755 Lisbon tsunami is the classical reference scenario. For the critical analysis of the global numerical database we decided to complement our study with two specific and nearly deterministic scenarios considered as the most-likely sources generating the Lisbon 1755 tsunami, namely the Gorringe and Horseshoe structures (Buforn et al., 1988;Stich et al., 2007;Cunha et al., 2012;Duarte et al., 2013;Grevemeyer et al., 2017). Both structures were modelled taking into account available maps (Cunha et al., 2012;Duarte et al., 2013) and fault parameters (Stich et al., 2007;Grevemeyer et al., 2017) summarized in Table 2. These scenarios have been chosen because they are representatives of the MCS_p approach for the study zone. They permit to evaluate the global tsunami database, which was designed by roughly zoning European (eastern) and Atlantic (western) shelves. In fact, the tsunami hazard induced by the Horseshoe scenario, located at the limit between these two zones of the global database, may be partly contained in the European contribution of the database. On the contrary, the Gorringe scenario represents a pure "European like" scenario within the Atlantic zone, hence excluded by the global database.

For the computation of the tsunami heights associated with these scenarios, fault parameters are considered uniformly distributed and are randomly sampled in their range of variation (Table 2), for a total of nearly 10,000 tsunami scenarios. The magnitude range associated with these tsunami scenarios varies from 7.7 to 8.9 and the hazard level associated to these faults corresponds to the "$\mu_m$" modelled water height. This hazard level for MCS_p approach was arbitrarily chosen, considering that the philosophy of this kind of approach is to take into account only the worst possible scenarios sources. In this sense, this value is in agreement with the choice reported in Roshan et al. (2016).

In Table 8 we summarized the tsunami hazard height from the global database and from Gorringe and Horseshoe faults for each gauge, with the associated uncertainty (corresponding to the residuals +/- $2\sigma_r$). These results suggest that the strongest tsunamis are globally generated by the Gorringe bank and that the proposed *hazard level* from the total database (MCS_h) is representative of the scenarios based tsunamis (MCS_p), once the uncertainty associated to meta-model predictions is considered (see Figure 10). In conclusion, the proposed hazard level resulting from UQ seems reasonable in light of the results obtained through a scenario-based-like approach, at least for the tsunamis impact along the French Atlantic coast due to sources off of Portugal.

**4.4 Results from sensitivity analysis: the influence of the seismic-source parameters: application of Step 3.2**

Homma and Saltelli (1996) introduced the total sensitivity index which measures the influence of a variable jointly with all its interactions. If the total sensitivity index of a variable is zero, this variable can be removed because neither the variable nor its interactions at any order have an influence on the results. This statistical index (called Sobol index "ST" in this paper, Eq. 16), is here of particular interest in order to highlight the earthquake source parameters that mostly control the tsunamis height at each tested gauge. In Figure 11, we reported the total Sobol index for the four meta-models of the French

Atlantic Gauges computed with the methodology proposed by Jansen et al. (1994) using the sensitivity-package R (Pujol et al., 2016). The accuracy of Sobol indices performed with Jansen's method depends on the number of model evaluations. For instance, in this study, we performed nearly 20 000 simulations using meta-models for the computation of Sobol indices. Results show that the slip parameter is globally the most-influencing parameter for all the French Atlantic Gauges meta-models, which is quite obvious considering that the fault-slip directly conditions the ocean floor deformation and hence the tsunami amplitude. However, Figure 11 suggests that the most influencing parameters for the four gauges are slightly different, depending on their location. One can differentiate results obtained for the southern gauges (i.e. La Rochelle & Gastes) from those obtained at northern gauges (i.e. Saint Malo & Brest):

- For the southern gauges, *width* is the second most relevant parameter, especially at La Rochelle where it is almost as important as the *slip* parameter. Other important parameters are *length* and *rake,* suggesting that for these gauges, fault source parameters in terms of magnitudes (depending on *width*, *slip* and *length* according to equations 1 and 2) and kinematics are the most important in generating hazard;

- For the northern gauges, apart from *slip*, *strike*, *width* and *rake* are also important, but slightly less than the *longitude* parameter. This means that the location of the source is here of major importance. A possible physical reason could be associated to the lack of the natural barrier composed by the north of Spain and Portugal which protects southern gauges from AGPB related tsunamis compared to northern gauges. As a consequence, the northern gauges should be more exposed to hazard in comparison to southern gauges, located in the shadow of Portugal and Spain.

In conclusion, these results also confirm that, beyond earthquake magnitudes, the position and the orientation of the faults are influents parameters, in agreement with the analysis of the previous paragraph (see 4.2). Thus, for a given magnitude, a tsunami generated by a "*well located-oriented*" fault would be potentially more hazardous, at least for the sites along the northern French Atlantic Coast. In other words, the French Atlantic coast is in the shadow of the tsunamis generated by the stronger continental shelf sources and more sensitive to the tsunamis generated in the oceanic shelf.

**5 Conclusions and perspectives**

The methodological research work presented in this paper was performed in order to test the interest of UQ for the assessment of the DTHA generated by earthquakes. We propose a new methodology for DTHA, consisting in the assessment of tsunami hazard through the analysis of all the possible tsunami heights at a given location (MCS_h). This concept goes beyond the definition of the Maximum Credible Scenario source parameters (MCS_p) classically reported in the literature (JSCE, 2002;Lynett et al., 2016) and employed for DTHA. MCS_h is evaluated through UQ allows the exploration of the tsunamigenic potential of a seismic zone in a rigorous and complete way. It can be considered as an extension of the previous innovative work from Roshan et al. (2016), which first improved the classical deterministic approach by focusing its analysis on some selected earthquake source parameters.

The methodology is based on the use of kriging meta-models that intensively explore all the possible tsunamis generated by a given seismic area. It is the first time, to our knowledge, that meta-models are used in this way in the context of DTHA (but also for PTHA). The meta-models are first constructed through a uniform exploration of the space of the inputs parameters of the seismic area and then tested and validated through a series of statistical indicators (correlation

5    coefficient, residuals analysis, MSE). These performance-tests allowed us to conclude that, even if the maximum residuals (differences between model and meta-model results) are very high for the four selected locations along the French Atlantic Coast (see Table 5), the meta-models are able to reproduce the model behaviour and they can be used for the construction of the numerical tsunami database. Nevertheless, when meta-models are used instead of the original model for the predictions of the tsunami heights, we propose to add an uncertainty corresponding to the "$+/-2\sigma_r$" residual value measured at each

10   gauge. Moreover, the realism of the tsunamis height simulated with the meta-models was also ensured through a comparison with the numerical results presented in Allgeyer et al. (2013) for the French Atlantic Coast. Thus, the constructed meta-models could be employed in a further study to roughly evaluate the impact of other seismic scenarios from the AGPB and impacting the French Atlantic Coast.

In the example of application presented in this work we built our meta-models using a design data-base of nearly 5 000

15   tsunami scenarios. The number of simulations is large and required intensive parallel computing using GPU. However, considering the large number of model evaluations necessary for UQ (more than 50 000) and GSA (more than 20 000), the meta-models appear as the more adapted tool for the proposed methodology.

We test this methodology by building a numerical database (of nearly 50,000 tsunamis scenarios) of tsunamis potentially generated by the AGPB and impacting the French Atlantic Coast at four sites of interest. As frequently reported

20   in the paper, this is only a test-case and all the results and the analysis presented in section 4 are an example of how an end-user of the methodology should use these results for a DTHA application. This database should permit to explore the numerous uncertainties related to the AGPB characteristics. Considering the convergence of the mean modelled tsunami height, it is not necessary to further explore the AGPB inputs parameters in order to refine the assessment of the CMS_h for the French Atlantic Gauges since statistically we have shown that the resulting tsunamis height distribution is representative

25   of the source variability. This may not be the case, however, for sites located closer to the AGPB region. In such regions a more refined zonation is necessary. However, the proposed methodology remains unchanged.

Tsunami hazard (MCS_h) was assessed directly from the tsunamis height distribution of the database (see section 4.1). Considering the rough zonation of the AGPB and the shape of the tsunami heights distribution we propose in this case to attribute to each French Gauge a tsunamis hazard level equal to the modelled tsunami height "$\mu_m+2\sigma_m$" with an

30   associated uncertainty equal to the residual "$+/-2\sigma_r$". To validate this choice, we analyse the strongest tsunamis (higher than the proposed *hazard level*) of the distribution. In our example, this analysis permits to conclude that, globally, the position and the orientation of the fault are influent parameters. In other words, for a given magnitude, a tsunami generated by a "*well located-oriented*" fault would be potentially more hazardous, at least for the sites located along the northern French Atlantic Coast. In this sense, the UQ study permitted to explore some rare combinations of seismic inputs parameters generating

higher tsunamis level (see Figure 9b). Moreover, the UQ permits to underline that globally the tsunamis generated by the *western shelf* are largely less impacting than the tsunamis generated by the *eastern shelf*, on average. Furthermore, GSA performed with Jansen's method permits to confirm these first results. Especially, the magnitude generating the tsunamis in the AGPB (depending on width, slip and length) is more relevant depending on the location of the gauge (the impacted

5    zone). For instance, only the earthquakes inducing large displacements of the sea-floor (large *slip*) are potentially capable to affect the middle/south of France. However, concerning the gauges located to the north of the French Atlantic Coast (*Brest* and *Saint-Malo*), other influencing parameters are, apart from the slip, the location (*longitude*) and the orientation (*strike* and *rake* parameters) of the fault sources. These results are also in agreement with results presented in (Allgeyer et al., 2013).

        The data base was also tested through a comparison with the tsunamis generated by a Lisbon-like 1755 earthquake

10   (paragraph 4.3). The latter representing a comparison between the global results from UQ (MCS_h) and a more classical MCS_p approach. Globally, the *hazard level* issued by the total database seems robust and representative of the tsunamis from *Gorringe* and *Horseshoe* banks once the uncertainty of numerical results is considered (see Figure 10), at least for French gauges off the Atlantic coast. Even if this work is mainly methodological, this results seems to confirm that the proposed MCS_h is adapted to the French Atlantic Coast as the tsunamis height from these scenarios are the strongest of the

15   tsunamis heights resulting from the global data-base. However, to be conclusive, the same analysis should be conducted with other MCS_p hypothesis for the AGPB. Moreover, it must also be noted that the choice of the hazard level is arbitrary and it depends on the chosen area, the available information about potential tsunami sources and the analysis of the obtained tsunami heights distribution.

        Tsunami hazard is analysed only off the French Atlantic Coast and by using, as input data for meta-model's

20   construction, the deterministic simulations performed with a rough computation grid-resolution of 2'' degrees (~3.6 km) which are not adapted to reproduce the influence of bathymetry on tsunami propagation in the vicinity of the coastal areas. For an operational DTHA along the French Atlantic Coast, it should be necessary, at first, to improve the actual numerical model in order to better represent the tsunamis run-up and the inundated areas, with a more accurate bathymetric grid. Then, the meta-models could be constructed with the methodology reported in section 2 and with a design data-base permitting to

25   explore as largely as possible the tsunamigenic potential from AGPB. Once validated, these meta-models could be used for DTHA as reported in section 3.3. However, in order to really assess the hazard level for the French Atlantic Coast, the constructed global data-base should be compared, at least, to the main MCS_p hypothesis for this area, to ensure that the chosen hazard level is not underestimated with respects to these scenarios.

        In perspective, for an operational application of the proposed methodology to locations closer to the source zones

30   (i.e. Portugal or Spain, Morocco), the sensitivity analysis will probably show that slip, length and width are the dominant parameters. In such cases, a more in-depth source analysis is necessary and the research performed in the region should be carefully considered (i.e. (Vilanova and Fonseca, 2007;Woessner et al., 2015;Matias et al., 2013;Omira et al., 2016;Omira et al., 2015)). However, uncertainties will always remain. Thus even in such cases, where more refined databases need to be established, the proposed approach should be of interest. In fact, as shown by the Fukushima accident, it is today necessary

to go beyond the classical MCS_p approach and to focus on all the possible tsunami heights at a given location (MCS_h). In this sense, we consider that the modelled tsunamis heights of our data-base are probably already representatives of the tsunamis heights generated by these scenarios for the French Atlantic Coast.

Finally, other analysis would be of interest for the improvement and the application of the methodology. At first, it should be possible to build more meta-models, in order to cover other area in locations where historical data from the 1755 Lisbone-tsunami are available (i.e. Santos and Koshimura, 2015;Baptista et al., 1998), thus permitting to further validate the constructed meta-models. Then, it would be of interest to test other hypothesis during the UQ step necessary for the assessment of the hazard (MCS_h), as different probability distribution of inputs parameters, a more realistic non-uniform slip distribution, considering its impact on tsunami heights (Geist and Dmowska, 1999), or, to analyze other approximation of the fault geometry.

As perspective, the example presented in this study for the AGPB suggests that it could be of interest to introduce the meta-models in the systems developed for tsunami early warning, considering the low computational time inherent to this statistical tool.

[revised manuscript text omitted]

**Tables**

| Step for meta-models design | Description |
|---|---|
| Initial design database | Monte-Carlo sampling in the distribution of the model input parameters. |
| Splitting data/base in training set and validation set | Splitting the initial data base (Amari et al., 1997): 1) Training set (80% of initial database scenarios) 2) Validation set (20% of initial database scenarios) |
| Construction | Estimation of meta-model parameters according to equations reported in section 2.4.2. |
| Accuracy/Optimization | k-fold cross validation (Hastie et al., 2002), (Kohavi, 1995) and (Breiman and Spector, 1992). |
| Meta-model uncertainty | Residual analysis and $R^2$ for the validation set. |

**Table 1:** Steps for meta-models construction and validation

| Seismic source parameter* | Design database | Western database | Eastern database | Gorringe bank | Horseshoe bank | Global database* |
|---|---|---|---|---|---|---|
| Strike [degrees] | 0 – 360 | 0 – 360 | 0 – 360 | 40 – 70 | 40 – 70 | 0 – 360 |
| Length [km] | 40 – 400 | 40 – 227 | 40 – 600 | 180 – 200 | 160 – 200 | 40 – 600 |
| Dip [degrees] | 1 – 90 | 60 – 90 | 10 – 90 | 10 – 40 | 10 – 40 | 10 – 90 |
| Rake [degrees] | -180 – +180 | -180 – 0 | 0 – +180 | 70 – 110 | 70 – 110 | -180 – +180 |
| Width [km] | 20 – 200 | 10 – 23 | 10 – 310 | 16 – 224 | 16 – 228 | 10 – 310 |
| Slip [m] | 1 – 25 | 1 – 25 | 1 – 25 | 5 – 25 | 5 – 25 | 1 – 25 |
| Longitude [degrees] | -18 – -7 | -18 – -10 | -10 – -7 | -12 – -11 | -10.5 – -9.5 | -18 – -7 |
| Latitude [degrees] | 34 – 40 | 34 – 40 | 34 – 40 | 36.5 – 37.5 | 35.8 – 36.5 | 34 – 40 |
| Depth [km] | 2 – 60 | 5 – 10 | 5 – 30 | 10 – 40 | 10 – 40 | 5 – 40 |
| Magnitude range $M_w$ [-] | 6.7 – 9.3 | 6.7 – 8.3 | 6.8 – 9.3 | 7.7 – 8.9 | 7.7 – 8.9 | 6.7 – 9.3 |

**Table 2:** Summary of the variation range of the seismic source input parameters for the design, the western, the eastern database and for the tsunami scenarios associated to the Gorringe bank and the Horseshoe bank (hypothesis from (Duarte et al., 2013) and (Grevemeyer et al., 2017)). *The global database is composed by the scenarios simulated with the meta-models (more then 50,000 tsunamis scenarios) for the construction of the *western* and the *eastern* database. **Seismic source parameters are assumed uniformly distributed and are randomly sampled for the construction of the Global database.

| Gauge | Longitude [degrees] | Latitude [degrees] | Depth [m] |
|---|---|---|---|
| Saint-Malo | -2.08 | 48.7 | 8 |
| Brest | -4.65 | 48.26 | 28 |
| La Rochelle | -1.65 | 45.93 | 40 |
| Gastes | -1.27 | 44.35s | 15 |

**Table 3:** Location and water depth of the French Gauges chosen for meta-model construction

| | Saint Malo | Brest | La Rochelle | Gastes |
|---|---|---|---|---|
| Min [m] | 0.00 | 0.00 | 0.00 | 0.00 |
| μ [m] | 0.21 | 0.47 | 0.37 | 0.30 |
| μ + σ [m] | 0.39 | 0.93 | 0.71 | 0.60 |
| μ + 2σ [m]] | 0.57 | 1.38 | 1.05 | 0.90 |
| Max [m] | 1.60 | 5.66 | 3.12 | 3.07 |

**Table 4:** Maximum water height associated with the design database; μ, σ and Max correspond to the mean, standard-deviation and maximum modelled values.

| Gauges | Training set | Validation set | | | | |
|---|---|---|---|---|---|---|
| | L_CV [-] | R2 [%] | $\mu_r$ [m] | $+\sigma_r$[m] | $+2\sigma_r$[m] | Max [m] |
| Saint-Malo | 0.004 | 87% | 0.000 | 0.066 | 0.131 | 0.498 |
| Brest | 0.012 | 94% | 0.001 | 0.112 | 0.224 | 0.982 |
| La Rochelle | 0.029 | 76% | 0.001 | 0.170 | 0.339 | 1.323 |
| Gastes | 0.021 | 80% | 0.004 | 0.148 | 0.293 | 1.392 |

**Table 5:** Summary of meta-model evaluations with the testing and the training data base; $\mu_r$, $\sigma_r$ and $Max_r$ correspond to the mean, standard-deviation and maximum residual respectively. L_CV is defined in Eq. 5.

| Hypothesis reported in Allgeyer et al. (2013) | | | | | | | | | | Results from meta-models constructed in this study | | | |
|---|---|---|---|---|---|---|---|---|---|---|---|---|---|
| Source | Strike [°] | Length [km] | Dip [°] | Rake [°] | Width [km] | Slip [m] | Lon [°] | Lat [°] | Depth [km] | Saint Malo [m] | Brest [m] | La Rochelle [m] | Gastes [m] |
| Johnston (1996) | 60 | 200 | 40 | 90 | 80 | 13.1 | 11.45 | 36.95 | 27 | 0.37 | 0.85 | 0.57 | 0.41 |
| Baptista et al. (2003) | 250 | 155 | 45 | 90 | 55 | 20 | -8.7 | 36.1 | 20.5 | 0.20 | 0.45 | 0.38 | 0.33 |
| Gutscher et al. (2006) | 21.7 | 96 | 24 | 90 | 55 | 20 | -10 | 36.8 | 20.5 | 0.24 | 0.39 | 0.42 | 0.28 |
| | 346 | 180 | 5 | 90 | 197 | 20 | -7.5 | 35.5 | 15.1 | 0.35 | 0.54 | 0.78 | 0.40 |
| | 346 | 180 | 30 | 90 | 12 | 20 | -8.6 | 35.3 | 3.2 | 0.10 | 0.21 | 0.10 | 0.10 |

**Table 6:** Maximum tsunamis height computed with meta-models using seismic sources simulated in (Allgeyer et al., 2013) for the 1755 Lisbon-tsunami.

|  | Saint Malo | Brest | La Rochelle | Gastes |
|---|---|---|---|---|
| $\mu_m$ [m] | 0.09 | 0.18 | 0.19 | 0.13 |
| $\mu_m + \sigma_m$ [m] | 0.17 | 0.34 | 0.36 | 0.27 |
| $\mu_m + 2\sigma_m$ [m] | 0.24 | 0.51 | 0.54 | 0.41 |
| $Max_m$ [m] | 0.77 | 2.01 | 1.81 | 1.92 |

**Table 7:** Maximum water height associated with the global database; $\mu_m$, $\sigma_m$ and $Max_m$ correspond to the mean, standard-deviation and maximum modelled values.

|  | HorseShoe ($\mu_m$) [m] | Gorringe ($\mu_m$) [m] | Global Database ($\mu_m + 2\sigma_m$) [m] | Meta-model uncertainty ($+/-2\sigma_r$) [m] | $Max_m$ [m] |
|---|---|---|---|---|---|
| Saint Malo | 0.18 | 0.25 | 0.24 | +/- 0.13 | 0.77 |
| Brest | 0.31 | 0.53 | 0.51 | +/- 0.22 | 2.01 |
| La Rochelle | 0.26 | 0.32 | 0.54 | +/- 0.34 | 1.81 |
| Gastes | 0.18 | 0.29 | 0.41 | +/- 0.29 | 1.92 |

5 **Table 8:** Numerical tsunami database and scenarios-based hazard levels. For each gauge, the uncertainty corresponds to the $+/-2\sigma_r$ also reported in Table 5. The maximum modelled tsunamis height is also reported in the next column.

**Figures**

| STEP 1: PROBLEM SPECIFICATION |
|---|
| **Goal:** enable the modelling of maximum tsunami heights generated by a chosen tsunamigenic area |
| **Method:** generation of tsunamis according to Okada (1985) surface deformation model and 2D propagation of tsunamis using Boussinesq equations as reported in Allgeyer et al. (2013) |

| STEP 2: META-MODELS CONSTRUCTINO AND VALIDATION |
|---|
| **Goal:** replace the numerical model, reduce the computational coast, extensive exploration of the seismic area |
| **Method:** kriging meta-model (see section 2.2 and APPENDIX 1) |

[Figure]

| STEP 3: DTHA ASSESSMENT THROUGH UQ AND GSA | |
|---|---|
| **Goal:** (i) estimation of all the possible tsunami heights generated by a seismic area and impacting the target zone; (ii) assessment of MCS_h (see sections 3.3, 4.1, 4.2 and 4.3) | **Goal:** critical analysis of the relevance of the seismic source parameters on tsunami heights at a given location (see section 4.4) |
| **Method:** Monte-Carlo simulations using kriging meta-model instead of the original model | **Method:** Computation of Sobol Indices using the methodology proposed by Jansen et al. (1994). See section 2.3.2 |

**Figure 1:** Global methodology proposed in this study.

[Figure]

**Figure 2:** (a) Numerical domain used for the deterministic simulations performed in this study and (b) geometrical input parameters of the tsunami-code employed in the study. *d* [km] is the depth of the seismic source which is assumed to be at the middle of the fault, *hL* [km] is the half fault length, *W* [km] is the half fault width, *x-y-z* are three-dimensional axes.

[Figure]

**Figure 3:** Bathymetry covering the computational domain. Red cross hatching show location of tsunamigenic sources used for meta-models construction. The points represent the gauge locations selected for tsunami database construction along the French Atlantic Coast. Gorringe fault and Horseshoe fault are special structures at the boundary between the Eastern domain (on the right) and the Western domain (on the left).

[Figure]

**Figure 4:** (a) Absolute residuals frequency distribution and (b) cumulate residual distribution for French Gauges computed as the difference between the modelled and meta-modelled tsunamis height for the *testing database*. The Table in figure (a) summarize the absolute residuals associated with higher quantiles of the distribution.

[Figure]

**Figure 5:** Comparison between the modelled and the meta-modelled maximum water height for the test database and the selected French Atlantic Gauges (blue points). The red line indicates the theoretical perfect correspondence between the original model and meta-model predications. Grey lines indicates the $1\sigma$ confidence interval for meta-models predictions and black lines the $2\sigma$ confidence interval for meta-models predictions.

[Figure]

**Figure 6:** Convergence study for the numerical tsunami database built through meta-models (only Brest and La Rochelle gauges are represented). The mobile mean of the simulated water height is stable after around two thousands simulations. The 95% confidence bounds for means correspond to "+/- $\left(\frac{2 \cdot \sigma_{m(N)}}{\sqrt{N}}\right)$" according to the central limit theorem and the chose Monte-Carlo sampling method, where $\sigma_{m(N)}$ is the mobile standard deviation

[Figure]

**Figure 7:** Distribution of the maximum tsunami water heights for tsunamis scenarios generated by the Horseshoe bank (blue line) and the Gorringe bank (red line). Comparison with scenarios associated to the western shelf (green), the eastern shelf (grey) and the global data base (black line).

[Figure]

**Figure 8:** Location of the seismic sources associated with the proposed tsunami hazard "MCS_h" ($\mu_m+2\sigma_m$) for the French Atlantic Gauges; the Magnitude associated with these tsunami scenarios varies between 7.5 and 9.

[Figure]

**Figure 9:** a) Magnitude distribution of earthquake events leading to the strongest (higher than $\mu_m+2\sigma_m$) tsunamis scenarios at the French Atlantic Gauges compared to the global data set; b) percentage of strongest tsunami scenarios (higher than $\mu_m+2\sigma_m$) associated with western and eastern domains in the magnitude range $M_w$ [7.8 – 8.5], in which western and eastern tsunami scenarios overlap. # indicates the number of tsunami scenarios.

[Figure]

**Figure 10:** Tsunamis hazard ("$\mu_m$" modelled values) for the French Atlantic Gauges associated with the 1755-like tsunamis scenarios form the Horseshoe bank (blue) and the Gorringe bank (red). Comparison with the tsunami hazard ("$\mu_m+2\sigma_m$" modelled water height) resulting from uncertainty quantification (black line). The black bars represent the uncertainty related to the use of a meta-model instead of the original model (equal to +/-$2\sigma_r$).

[Figure]

**Figure 11:** Total order Sobol Index computed for the French Atlantic Gauges.

---

## Author Comment (AC4) · 3 Oct 2018

**Responses to RC4**

October 3th, 2018

Re: Re-submission of manuscript "Development of a methodological framework for the assessment of seismic induced tsunami hazard through uncertainty quantification: application to the Azores-Gibraltar Fracture" by Vito Bacchi et al. NHESS-2018-142.

Dear RC4,

We appreciate your careful review and constructive suggestions. It is our belief that the manuscript is substantially improved after making the suggested edits. Following this letter are the reviewer comments with our responses in italics, including how and where the text was modified. Changes made in the manuscript are marked in blue (see attached file).

According to the comments from Dr. Luis Matias (RC1), we propose to modify the title of the paper as follow:

"Development of a methodological framework for the assessment of seismic induced tsunami hazard through uncertainty quantification: the test case of the Azores-Gibraltar Plate Boundary".

Thank you for your consideration.

Sincerely, Vito Bacchi on behalf of authors

**Specific points:**

(1) The presented application to the Azores-Gibraltar Fracture Zone takes very little in the way of constraints or weightings from what is known about the geology, geophysics and seismology of the area. A better indication of how prior knowledge can be included in such a study is something that I think will be needed before the presented methodology can be put into practical use. Related to this, is a lack of connection to the time-frame associated with the hazard, eg the return period associated with a water level at the study sites, something that is necessary for many applications of tsunami hazard.

The authors recognize that the objective of our work was not clearly defined in the original paper. Our objective is to propose a new methodology for the improvement of Deterministic Hazard Assessment (DTHA), classically based on the definition of a Maximum Credible Scenario (MCS). In the introduction of the revised paper we introduce the sentence below:

"Nowadays, in the classical DTHA, as reported in the cited papers, MCS is mainly focused on seismic source parameters (MCS\_p) and not on the impact (MCS\_h) (i.e. the tsunamis heights at a given location). This approach, even if classically exploited, has already shown its limitations, as for the Fukushima accident of 2011 (REF). In fact, in a context of strong uncertainties on the characterization of seismic sources, using MCS\_p for DTHA may lead to an underestimation of MCS\_h in the area of interest. In this context, the objective of our work is to propose a new methodology to go beyond the MCS\_p approach and focus on MCS\_h through the evaluation of all the possible tsunamis heights at a given location. MCS\_h has also the advantage that it accounts for other scenarios, generated by potential seismic sources not recognized in the seismic area but generating tsunamis heights potentially impacting the zone to protect. This methodological framework for the improvement of DTHA is based on classical uncertainty quantification (UQ) techniques (i.e. (Saltelli et al., 2000;Saltelli et al., 2008)), which are robust statistical tools permitting to largely explore a given seismic area and provide all the possible tsunamis heights generated by this area. This approach is a new philosophy for DTHA and can permit to go beyond the MCS\_p approach and ensure that MCS\_p is the MCS\_h scenario."

Our methodology is not related to the Probabilistic Tsunami Hazard Assessment (PTHA) and the objective of this paper is not to do a probabilistic evaluation. For this reason, we do not introduce some basic concepts as the prior knowledge or the time-frame associated to the hazard.

I think the authors are aware of the current limitations of their results, and they repeatedly mention that this is '... not an operational assessment ...'. There could be some differences in interpretation about what 'operational' means, and I think it would be better if they could be more explicit about this, eg at the end of the Introduction they might consider something like: '... not an operational assessment of tsunami hazard along the French Atlantic Coast, and no reliance should be placed on these results for practical purposes'. With something like that, and more acknowledgement of the current limitations and areas for more work, the authors would be on safer ground that this manuscript is just an introduction to the 'meta-model' concept.

As you noticed, our study does not permit an operational assessment of DTHA along the French Atlantic Coast and we tried to remind this aspect along the manuscript. In particular, we underline in the abstract that:

"It must be underlined that the results from this study are the illustration of a general methodology through a case study with simplified hypothesis, which is not an operational assessment of tsunami hazard along the French Atlantic Coast."

And at the end of the introduction we indicate that:

"It must be noted that the aim of this study is to show how a numerical database constructed using "uncertainty quantification techniques" can be a useful tool for the analysis of the tsunamigenic potential of a seismic zone. As a consequence, this paper illustrates a methodology through a case study, which is not an operational assessment of tsunami hazard along the French Atlantic Coast"

Moreover, in the conclusion of the revised paper we also presented the improvement of the actual study which would be necessary for DTHA (see track changes in blue).

(2) The methodology is hard to follow, and in particular, the relationship between the various different databases in Table 5 and how they are used in the analysis, was difficult to understand. I suggest the authors try to construct a flow diagram or other type of figure to explain this graphically (could be somewhat similar to Figure 1).

We agree with RC4 that the structure of the paper was definitely not clear. As a consequence, the organization of the paper was changed and the methodology more detailed. A flow diagram of the main three steps of the methodology is reported on the new Figure 1 (the ancient was removed). Moreover, the section 2 is reorganized in order to introduce the proposed steps for the application of the methodology and provide further details. In the revised paper, we first introduce the main three steps of the proposed methodology (beginning of the section 2), and then we detail each step. The meta-model explanation was moved to the beginning of Section 2 (section 2.2) after an introduction of the numerical model used for tsunami modelling (section 2.1).

(3) The Antoschenkova (2016) reference is key to understanding the design database, but this is just a conference presentation and I could not find the information needed to assess the suitability of this work to the purpose to which it is applied. If the information is not published elsewhere, I suggest a summary be included here - perhaps as an Appendix.

We agree with this comment. As a consequence, the citations to Antoshchenkova et al., 2016 and Imbert et al., 2015 were deleted as these references are not available in open-access literature. Instead, we present only the hypothesis employed for the construction of the design data-base, which is based on Monte-Carlo sampling of the inputs parameters reported in Table 2 (of the revised paper) and also introduce the numerical domain (Figure 2 of the revised paper) and the numerical model used for the simulations of the design data-base (sec 2.1 of the revised paper). As reported in the revised paper, all the equations associated to tsunami generation and propagation are reported in Allgeyer et al. (2013).

(4) It seems a pity to me that all of the study sites are located along the French coast. In particular the sensitivity analysis is, in my view, one of the most interesting parts of the paper, but it would be really nice to see how the sensitivity applies to sites closer to the source region.

We agree with this comment. It would be very interesting to focus on some sites located near the source zone (i.e. Portugal, Spain, Morocco). The authors preview a further research for a real tsunamis hazard assessment. Especially, we aim to explore the AGFZ (called Azores Gibraltar Plate Boundary in the revised paper - AGPB) with the same methodology proposed in this study, but by introducing some essential modification:

- the use of a refined numerical model, in order to take into account the coastal effects and the run-up;
- the construction of a numerical data-base with more detailed geological hypothesis from literature;
- the validation of meta-models with historical data from the 1755 Lisbon tsunami

This work will permit the assessment of tsunamis hazard and test our new philosophy for DTHA (the concept of the MCS\_h) on a real application case.

(5) I'm presuming that the magnitudes in Table 5 are calculated after the uniform sampling of length, width and slip and applying constraints to these. And that the distribution of magnitudes is just what falls out from this process (and not linked to any assumption about earthquake statistics - see point 1). This needs clarification (unless it was mentioned and I missed it).

We first would like to underline that, as you presumed, in this study there are not earthquake statistics in our calculations as this methodology is only focused on DTHA. However, we agree that a clarification is needed and we inserted in the section 3.1 (Design data-base for meta-models construction) of the revised paper the following sentence to clarify this issue:

"In order to build the design database, fault parameters as defined in section 2.1 and Table 2 were sampled randomly and independently with Monte-Carlo method and assuming uniform distributions. The uniform distribution was chosen in order to build meta-models able to reproduce tsunamis heights generated by various seismic sources with the same accuracy. The resulting earthquake magnitudes are computed using the sampled parameters with equation 2. The shear modulus chosen for the magnitude estimation is a constant value assumed to be equal to 30 GPA."

(6) Non-uniform slip distributions have been shown to be influential in estimating tsunami hazard (see eg Geist and Dmowska, 1999), but the study here is (as far as I could tell) all based on uniform slip modelling. It is not at all obvious to me how this might be included in the methodology presented here, and it would be nice if the authors could comment on this.

Similar to this is the approximation of the rupture by a rectangular plane, which may be suitable for low magnitude earthquakes but will not be realistic for ones at the upper end of their study range. Again, this does not seem like an easy thing to incorporate.

We agree with these comments and we consider that the introduction of the variability of these parameters in our study would complete our investigation.

In fact, we expect that the parameters mentioned by RC4 should have a strong impact near the source zone but a lower impact on the French Atlantic Coast considering its location with respect to the AGPB. Moreover, considering the methodological objective of the paper and the rough approximation related to the tsunami model, we consider that this aspect can be neglected for the simple illustration of the methodology.

However, we introduce this discussion of the revised paper (see conclusions), as follow:

"Finally, other analysis would be of interest for the application of the methodology. At first, it should be possible to build more meta-models, in order to cover other area in locations where historical data from the 1755 Lisbon-tsunami are available (i.e. (Santos and Koshimura, 2015; Baptista et al., 1998)), thus permitting to further validate the constructed meta-models. Then, it would be necessary to test other hypothesis during the UQ step necessary for the assessment of the hazard (MCS\_h), as different probability distribution of the inputs parameters, a more realistic non-uniform slip distribution, considering that this parameter can strongly influence the tsunami hazard (Geist and Dmowska, 1999), or, by testing other approximation of the rupture surface, which is rectangular in our simulations."

(7) The authors may find that some of their problems with finding suitable wave height statistics could be improved by working with log-normal distributions. A starting point for this would be to search for 'Aida-k parameter' (original paper Aida, 1978).

We consider that RC4 addresses a very interesting issue. However, given the objectives of the study (methodological) and the rough numerical model used for tsunamis simulation, we neglected this aspect in the present work. These aspects will be integrated in a further research activity focused on a real DTHA.

**Bibliography:**

Baptista, M., Miranda, P., Miranda, J., and Victor, L. M.: Constrains on the source of the 1755 Lisbon tsunami inferred from numerical modelling of historical data on the source of the 1755 Lisbon tsunami, Journal of Geodynamics, 25, 159-174, 1998.

Geist, E. L., and Dmowska, R.: Local tsunamis and distributed slip at the source, in: Seismogenic and tsunamigenic processes in shallow subduction zones, Springer, 485-512, 1999.

Saltelli, A., Chan, K., and Scott, E. M.: Sensitivity analysis, Wiley New York, 2000.

Saltelli, A., Ratto, M., Andres, T., Campolongo, F., Cariboni, J., Gatelli, D., Saisana, M., and Tarantola, S.: Global sensitivity analysis: the primer, John Wiley & Sons, 2008.

Santos, A., and Koshimura, S.: The historical review of the 1755 Lisbon Tsunami, J. Geodesy Geomat. Eng, 1, 38-52, 2015.

**Development of a methodological framework for the assessment of seismic induced tsunami hazard through uncertainty quantification: the test case of the Azores-Gibraltar Plate Boundary**

Vito Bacchi1, Hervé Jomard1, Oona Scotti1, Ekaterina Antoshchenkova1, Lise Bardet1, Claire-Marie Duluc1, Hélène Hebert2

1Institute for Radiological Protection and Nuclear Safety (IRSN), Fonteany-aux-Roses, 92262, France 2CEA, DAM, DIF, 91297 Arpajon Cedex, France

Correspondence to: Vito Bacchi (vito.bacchi@irsn.fr)

Abstract. The aim of this study is to propose a new methodology for deterministic tsunami hazard assessment based on the
 analysis of all the possible tsunami heights at a given location. With this new approach, the hazard level is evaluated through a numerical database constructed using "uncertainty quantification (UQ) techniques" which allows the exploration of the tsunamigenic potential of a seismic zone in a rigorous and complete way.. This concept goes beyond the definition of the Maximum Credible Scenario based on the definition of particular source parameters and classically reported in the literature. The proposed methodology relies on the construction and validation of "emulators", or "meta-models", that drastically

- 15 reduce the computational time necessary for tsunami simulations. It is the first time, to our knowledge, that meta-models are used in this way. To test the methodology, a numerical database of nearly 50,000 tsunamis scenarios generated by the Azores-Gibraltar Plate Boundary (AGPB) and that may potentially impact the French Atlantic Coast was constructed. Tsunami heights distributions resulting from the UQ using meta-models are then discussed to illustrate the advantages of such an approach in decision-making compared to results which can be obtained with a more classical scenario-based
- 20 approach. In particular, the results suggest that the most influential parameters controlling tsunami heights generated in the AGPB is not only the magnitude but also the location of the impacted zone. It must be underlined that results from this study are used to illustrate the general methodology through a case study with simplified hypothesis and should not be considered for an operational assessment of tsunami hazard along the Atlantic Coast.

**1** Introduction**

[revised manuscript text omitted]

- 5 the Azores Gibraltar Plate Boundary (AGPB). The Great 1755 Lisbon earthquake generated the most historically destructive tsunami near the Portugal coasts (Santos and Koshimura, 2015). Source location and contemporary effects of such tsunami are not precisely identified and several earthquake scenarios have already been published in the literature since last decades (Baptista et al., 1998;Grandin et al., 2007;Gutscher et al., 2006;Horsburgh et al., 2008;Johnston, 1996;Zitellini et al., 1999;Baptista et al., 2003;Terrinha et al., 2003;Gracia et al., 2003;Barkan et al., 2009;Cunha et al., 2010). All of these
- 10 studies, mainly based on a comparison of several fault hypotheses based on geological properties of the AGPB, show how variable parameters of the seismic source can be, depending on the studied fault and focal mechanisms.

Nowadays, in the classical DTHA, as reported in the cited papers, MCS is mainly focused on seismic source parameters (MCS\_p) and not on the tsunamis heights at a given location (MCS\_h). The great M 9.0 Tohoku-Oki subduction earthquake of 2011, the largest ever recorded in Japan (Saito et al., 2011), has clearly shown the limitations of the MCS\_p approach. In fact, considering the uncertainties related to the characterization of seismic sources, using MCS\_p for DTHA may lead to an underestimation of MCS\_h in the area of interest. In this context, the objective of our work is to propose a new methodology to go beyond the MCS\_p approach and focus more on MCS\_h through the evaluation of all the possible tsunamis heights at a given location. The MCS\_h approach has the advantage that it accounts for other scenarios, generated by potential seismic sources not associated to the MCS\_p but generating tsunamis heights potentially impacting the zone to

- 20 protect. The proposed methodological framework for the improvement of DTHA is based on classical uncertainty quantification (UQ) techniques (i.e. Saltelli et al., 2000; Saltelli et al., 2008), which are robust statistical tools permitting to largely explore a given seismic area and provide all the possible tsunami heights generated by this area. This approach is a new philosophy for DTHA and can also permit to ensure that MCS\_p is the MCS\_h scenario for the target area.
- In fact, despite the natural complexity of seismic zones prone to tsunamis, in DTHA studies the uncertainties related to model parameters are generally not taken into account in a rigorous and robust way (Lynett et al., 2016;JSCE, 2002), considering the mathematical framework available in the literature (Saltelli et al., 2004;Saltelli et al., 2008;Faivre et al., 2013;Iooss and Lemaître, 2015), and the classical approach to deal with uncertainties consists to perform a limited number of deterministic simulations with penalized values of the seismic sources. For instance, Allgeyer et al. (Allgeyer et al., 2013) modelled the impact of three different scenarios for the 1755 earthquake on the French Atlantic Coast, with a focus on La
- 30 Rochelle harbour. The authors show that, depending on the source hypothesis and tide conditions, several areas (western part of the island of Re´ and northern coast of the island of Oléron) may have experienced a moderate impact from 0.5 to 1 m for this tsunami event. Recently, Roshan et al. (2016) tried to improve the DTHA procedure detailed in Yanagisawa et al. (2007) in order to better evaluate the effects of the seismic source uncertainties. The authors focused on a possible range of parameters which could produce more robust estimates of hazard instead of using single value of each seismic source

parameter (Roshan et al., 2016). Basically, their methodology is based on the classical steps of an uncertainty propagation study: (i) the identification and characterization of tsunamigenic earthquake sources, (ii) the estimation of rupture parameters and associated parametric variations, (iii) the exploration of a tsunami numerical model and, finally, (iv) the parametric study capturing the uncertainties through the numerical simulations of the tsunami waves associated to different model inputs

- 5 parameters varying within a given range. The deterministic numerical results are then processed and hazard maps of mean simulated water height are produced to assess the tsunami hazard (Roshan et al., 2016). Finally, the mean simulated water height of the different tsunami scenarios is employed to design flood level due to tsunami along Indian coast (Roshan et al., 2016). The objective of this exploratory and innovating methodological work was not directly the analysis of the benefits of uncertainty quantification on DTHA (Roshan et al., 2016), even if the authors presented an improvement of the classical
- 10 MCS approach by considering the uncertainties related to a limited number of seismic source parameters (the dip angle, the strike and the source location), leading to a limited number of tsunami scenarios (around 320). In this sense, they were not interested, for instance, to estimate the robustness of the mean simulated tsunami height considering the low number of simulations. Moreover, the sensitivity of model results to the input parameters was not studied (Roshan et al., 2016). Finally, the other seismic source parameters (e.g. the mean rupture length and width, the slip and the magnitude) were considered as
- 15 constant for each tsunami scenario and their uncertainty was not evaluated, thus strongly limiting the impact of their study with respect to a more classical uncertainty quantification approach.

In this study, because of the computational constraints associated with UQ, the proposed methodology relies mainly on the use of a meta-model, an emulator of the tsunami numerical model, that make it possible to perform a large number of tsunami scenarios with reduced computational time and, consequently, to intensively explore a tsunamigenic area for which

- 20 geological and geophysical datasets are in general limited. Statistical emulators have already been employed in the field of tsunamis, for the analysis of landslide-generated tsunamis (Sarri et al., 2012) or for the evaluation of the uncertainty in the friction parameterization used in tsunamis simulations (Sraj et al., 2014). More recently, Rohmer et al. (2018) proposed a Bayesian procedure to infer (i.e. learn) the probability distribution of the source parameters of the earthquake, based on the combination of a kriging-based metamodelling technique, to overcome the high computation time cost of the numerical
- 25 simulator, and an Approximate Bayesian Computation (ABC) procedure, to perform the Bayesian inference. The authors applied the procedure to the Ligurian (North West of Italy) 1887 tsunami case. All the mentioned studies show that it is of interest to use meta-models for tsunamis analysis in many contexts and at least for reducing the computational time necessary for tsunami simulations (Sarri et al., 2012).

In this research work, the constructed meta-models are exploited with the statistical criteria classically employed in 30 UQ studies (Saltelli et al., 2000;Saltelli, 2002;Saltelli et al., 2008;Iooss and Lemaître, 2015) for the assessment of DTHA. To our knowledge, it is the first time that meta-models are tested for the assessment of DTHA. The methodology was hereafter applied to the AGPB area by the construction and the validation of four meta-models able to reproduce the maximum tsunami heights along the French Atlantic Coast. The AGPB area was chosen as an exercise because of the large available literature, because it represents one of the most important sources of potential earthquake-related tsunamis for the French Atlantic coast (as attested by the impact of the 1755 tsunami along the French coast, http://tsunamis.brgm.fr/) and because the results of our constructed meta-models could be compared with previous deterministic simulations of tsunamis from AGPB along the French Atlantic Coast (Allgeyer et al., 2013).

- Finally, in section 4 we present how the numerical data-base constructed through UQ techniques should be employed for DTHA along the French Atlantic Coast and its critical analysis. The analysis is focused on the hazard-level which should be retained from UQ (MCS\_h) and on the comparison between the tsunamis data-base with tsunamis heights obtained with a more classical approach (MCS\_p). It must be noted that the aim of this study is to show how a numerical database constructed using UQ "techniques" can be a useful tool for the analysis of the tsunamigenic potential of a seismic zone. As a consequence, this paper illustrates a methodology through a case study, and is not to be considered as an
- 10 operational assessment of tsunami hazard along the French Atlantic Coast.

**2 Presentation of the methodology: theoretical background**

In this study, we propose a general methodological framework for the exploration of the tsunamigenic potential of a given seismic area and the quantification of the associated uncertainties. This methodology should permit to identify all the possible tsunamis heights at a given location through UQ and then to assess a hazard level (MCS\_h). It should be suitable in

- 15 two situations: (i) for poorly known tsunamigenic areas with no (or poor) information related to seismic sources or (ii) to evaluate the robustness of the MCS\_p, by ensuring that this scenario is really the more impacting for the area to protect. The proposed approach relies on three main steps, as reported in Figure 1. STEP 1 consists in the construction of a numerical model able to reproduce the tsunami heights generated by a given seismic area and impacting a target area. In STEP 2, the numerical model simulates a regular set of physical tsunamigenic scenarios (so called design data-base) that are used for the
- 20 construction and the validation of an emulator (the meta-model) able to reproduce the original model results in the target zone. In STEP 3, the validated meta-models are used for DTHA. The UQ performed using meta-models instead of the original model permits to explore intensively the tsunamigenic area with nearly zero computational time. The distribution of tsunami heights resulting from UQ (step 3.1) can then be analysed and discussed in order to define a given hazard level for the zone to protect (CMS\_h), as presented in section 4. Finally, in order to identify the model parameters contributing the
- 25 most to the tsunamis height at a given location a global sensitivity analysis (step 3.2) is performed. The mathematical hypothesis of each step are described in the next paragraphs, with a focus on the AGPB. This methodology is general and can be used in different hydraulic fields.

**2.1 Step 1: numerical tool for tsunami simulations**

The first step consists in the construction of a tsunami numerical model of the area to explore. In this study, the 30 tsunami numerical simulations were performed by using the CEA tsunami code which exploits two models, one for tsunami initialization and the other one for tsunami propagation. The initial seabed deformation caused by an earthquake is generated with the Okada model (Okada, 1985) and is transmitted instantaneously to the surface of the water. This analytical model satisfies the expression of the seismic moment  $M_0$ :

$$M_0 = \mu \cdot D \cdot L \cdot W \tag{1}$$

where  $\mu$  denotes the shear modulus, D [m] the average slip along the rupture of length L [m] and width W [m]. Then, the seismic magnitude " $M_w$ " is directly computed through equation 2, as follow:

$$M_W = \frac{2}{3} \cdot \log_{10}(M_0) - 6,07 \tag{2}$$

5 The following parameters are also required for tsunami initiation: longitude, latitude, and depth [km] of the center of the source, strike [degrees], dip [degrees], and rake [degrees]. A conceptual scheme of the input parameters for the tsunami-code and the numerical domain used in this study are reported in Figure 2. Then, the computation of the tsunami propagation is based on hydrodynamic equations, under the nonlinear shallow water approximation (the Boussinesq equations as reported in Allgeyer et al., 2013). Shallow water equations are discretized using a finite-differences method in space and time (FDTD). Pressure and velocity fields are evaluated on uniform separate grids according to Arakawa's C-grid (Arakawa, 1972). Partial derivatives are approximated using upwind finite-differences (Mader, 2004). Time integration is performed using the iterated Crank-Nicholson scheme. No viscosity terms are taken into account in our simulations. The only parameters of this second model are the bathymetry (space step and depth resolution) and the time step.

In this study, all the simulations were performed on the same bathymetric grid with a space resolution of 2' (~3.6 km). The numerical model was not directly validated by the comparison with similar simulations from literature. Considering the rough bathymetrical grid resolution, the developed numerical model is not adapted to the estimation of the tsunamis run-up and the inundation areas and it can't be used for a real assessment of the tsunami hazard along the French Atlantic Coast. However, this work being methodological, the authors consider that the numerical results are consistent with the objectives of the study. Moreover, the tsunami-code was largely validated through extensive benchmarks in the 20 framework of the work package 1 of the TANDEM research project (Violeau et al., 2016) by ensuring its ability to

reproduce tsunamis generation and propagation. As a consequence, the order of magnitude of the tsunami heights computed in this study should be realistic and adapted to the test of the methodology.

Finally, in order to perform the numerical simulations needed for the meta-model construction and validation (see section 2.2), the CEA code was coupled with the IRSN Promethee bench. Promethee is an environment for parametric

25 computation that allows carrying out UQ studies, when coupled (or warped) to a code. This software is freely distributed by IRSN (http://promethee.irsn.org/doku.php) and allows the parameterization with any numerical code and is optimized for intensive computing. Promethee was first linked to the numerical code by means of a set of software links (similar to bash scripts). In this way, numerical simulations were directly lunched by the IRSN environment. Then, the statistical analysis, such as the Monte-Carlo simulations used for the meta-model construction (see section 2.2.1) and UQ (see section 2.3.1) or

the Sobol indices computation (see section 2.3.2) were also driven by Promethee, which integrates R statistical computing environment by permitting this kind of analysis (R Core Team, 2016).

**2.2 Step 2: Meta-model design and validation**

- For the evaluation of numerical codes in industrial applications, where the computational code is expensive in
  computation time, so that it cannot be evaluated intensively (e.g. only several hundred calculations are possible), it is usually not easy to estimate some numerical parameters (i.e. the sensitivity Sobol indices), requiring for each input several hundreds or thousands of evaluations of the model (Iooss and Lemaître, 2015). As a consequence, a meta-model instead of the model is generally used in the estimation procedure. A meta-model is a mathematical model of approximation of the numerical model, built on a learning basis (Fang et al., 2006). The meta-model solution is a current engineering practice also for
  estimating sensitivity indices (Gratiet et al., 2016). In this study, we use the Gaussian process meta-model (also called kriging) (Sacks et al., 1989), which good predictive capacities have been demonstrated in many practical situations (see Marrel et al., 2008, for example). As a consequence, kriging meta-model will be useful both for sensitivity analysis and for a numerical database construction through uncertainty quantification (as reported in the next section). The classical steps for meta-models construction and validation are reported in various studies (i.e. Faivre et al., 2013;Saltelli, 2002; Saltelli et al., 2008), and can be shortly summarized as follow (see Table 1).
- 15 2008), and can be shortly summarized as follow (see Table 1).

The mathematical construction of the meta-model obviously depends on the chosen meta-model. The theoretical background for kriging meta-model design used in this study is fully detailed in Roustant et al. (2012). The associated equations are implemented in the R-packages DiceKring and DiceOptim and reported in the APPENDIX 1 for the interested reader.

**20 2.2.1 The initial design for meta-model fitting**

For computer experiments, selecting a *design database* (or space filling design) is a key issue in building an efficient and informative meta-model (Iooss et al., 2010). The design of the experience is the choice of the input parameters (and, consequently, of the initial simulations) to use in order to build the most accurate meta-model at a minimum computational coast (minimum number of model simulations). It can depend on the chosen meta-model (Kleijnen, 2005). For instance, for Gaussian emulators as kriging, one of the most popular design type is the Latin Hypercube Sampling (LHS) (McKay et al., 1979), which offers flexible design size for any number of simulation inputs (Kleijnen, 2005). An equivalent approach, used in this paper, is the Monte-Carlo sampling technique which also permits a regular and uniform exploration of the space of the model input parameters. In fact, except for some numerical tests reported in previous studies (Iooss et al., 2010), at the moment, no theoretical results gives the type of initial design leading to the best fitted meta-model in terms of

30 meta-model predictions.

**2.2.2 The partitioning of the original data base**

5

10

It is generally assumed that the simulations performed to build the meta-model (the *design database*) can be partitioned in a *training set* (data used for the estimation of the meta-model) and a *validation set* (simulations performed to test the ability of the meta-model to reproduce the model results with other data). According to Amari et al. (1997), to obtain an optimum unbiased meta-model, nearly 80% of the simulations of the *design database* must be used for the *training set* and 20% for the *validation set*. This ratio is used in our study.

2.2.3 The validation of the chosen type of meta-model: training set

In order to assess if a given meta-model (i.e. kriging, gam method, random forest, chaos polynomials, and so on...) is adapted to reproduce the model behaviour for the design data base, probably the simplest and most widely used method is the *cross-validation* or *K-fold* cross-validation method (Hastie et al., 2002). This step is performed with the *training set*. The principle of cross-validation is to split the data into *K* folds of approximately equal size  $A_IAI,...,A_kAK$ . For k = 1 to *K*, a model  $\hat{Y}^{(-k)}$  is fitted from the data  $U_{j\neq k}A_k$  (all the data except the  $A_k$  fold) and this model is validated on the fold  $A_k$ . Given a criterion of quality *L* as the Mean Square Error (Eq. 3):

$$L = MSE = \frac{1}{n} \sum_{i=1}^{n} (\hat{y}_i - y_i)^2$$
(3)

The quantity used for the "evaluation" of the model is computed as follow:

$$L_{k} = \frac{1}{n/K} \sum_{i \in A_{k}} L(y_{i}Y^{(-k)}(x_{i}))$$
(4)

15 Where  $\hat{y}_i$  and  $y_i$  are, respectively, the meta-model and the model response and *n* is the number of simulations in the *kth* sample. Finally, the cross-validation used in this study is evaluated through the mean of the quantity  $L_k$  computed for each fold:

$$L_{-}CV = \frac{1}{K} \sum_{k=1}^{K} L_k \tag{5}$$

It must be noted that when K is equal to the number of simulations of the training set, the cross-validation method corresponds to the leave-one-out technique (Oliveira, 2008). The methodology employed is described in the DiceEval R-

20 package reference-manual (Dupuy et al., 2015). We considered K = 10 and compute the MSE between meta-model approximation and the k-fold observations for each fold (k times). The mean MSE is an index of the mean differences between meta-model predictions and model results, thus representing the ability of the chosen meta-model to reproduce the original model. The more this index approaches zero, the more the meta-model can be considered accurate.

**2.2.4 Quantification of the meta-model uncertainty: residuals analysis**

This last operation is performed on the *validation-set* (composed of the 20% of the *design database*). The objectives are (i) to evaluate the meta-models built on the entire *testing set* (after cross-validation) outside their construction domain (*training set*) and (ii) to estimate the uncertainty related to the use of the meta-model instead of the original model. The first statistical parameter we chose for evaluating the meta-models performance is the correlation coefficient  $R^2$  (eq. 6), classically used in statistic:

$$R^{2} = 1 - \frac{\sum_{i=1}^{n} (\hat{y}_{i} - y_{i})^{2}}{\sum_{i=1}^{n} (\hat{y}_{i} - \overline{y}_{i})^{2}}$$
(6)

Where  $\overline{y}_i$  is the mean model response.

5

Finally, a particular attention was devoted to the analysis of the distribution of the residuals (the difference between model results and meta-model predictions). Especially, the residual plots can help avoiding inadequate meta-models and help adjusting the meta-model for better results. In general, it is assumed that the simplest meta-model that produces random residuals is a good candidate for being a relatively precise and unbiased model. In this study, residuals  $\mathcal{E}_i$  between model  $y_i$  and meta-model  $\hat{y}_i$  predictions are evaluated as follow:

$$\varepsilon_i = y_i - \hat{y}_i \tag{7}$$

Two kind of analysis are possible, both performed in this study: (i) the analysis of the distribution of the residuals, which requires residuals to be normally distributed with mean zero (Jarque and Bera, 1980) and (ii) the analysis of the cumulative frequency distribution of the modulus of residuals (reported in the next section). This latter analysis provides an order of

15 frequency distribution of the modulus of residuals (reported in the next section). This latter analysis p magnitude of the total uncertainty related to meta-model prediction, as discussed in the next sections.

**2.3 Step 3: Uncertainty Quantification and Global Sensitivity Analysis**

Considering the variety and the complexity of the geophysical mechanisms involved in tsunami generation, tsunami hazard assessment is generally associated with strong uncertainties (aleatory and epistemic). In PTHA, uncertainties are
classically integrated in a rigorous way (Selva et al., 2016;Sørensen et al., 2012;Horspool et al., 2014) and quantified using the logic-tree approach (Horspool et al., 2014) and/or random simulations performed using the Monte-Carlo sampling of probability density functions of geological parameters (Horspool et al., 2014;Sørensen et al., 2012). An alternative and interesting approach is recently proposed by Selva et al. (2016), consisting in the use of an event tree approach and ensemble modelling (Marzocchi et al., 2015). Moreover, a new procedure was recently proposed by Molinari et al. (2016) for the quantification of uncertainties related to the construction of a tsunami data-base based on the quantification of elementary effects.

In this work we propose a classical methodology that could be adapted to analyse tsunamigenic regions with poor (or no) information on crustal characteristics and based on the classical uncertainty study steps (Saltelli et al., 2008;Saltelli et al., 2004; Faivre et al., 2013; Jooss and Lemaître, 2015). This methodology was already tested in other hydraulic context in recent years (Abily et al., 2016; Nguyen et al., 2015). At first, the variation intervals and the probability density functions (PDF) for the inputs affected by uncertainty must be defined for UQ. We remind that, for a random variable X defined on an [a,b] interval, a PDF  $f_x(X)$  is defined as follows:

$$P(X \in [a,b]) = \int_{a}^{b} f_{X}(X) dx$$
(8)

Most of the time, as in the present study, the probability distributions of random inputs is unknown. Hence, expert's knowledge or the experimental evidence available is used as the basis for the PDF definition. The step 3.1 of our methodology requires propagating in the model the uncertainty associated to the input parameters with the objective of computing the PDF  $f_v(Y)$  of the resulting quantities of interest y, as the maximum tsunami height employed in this study.

Finally, a better understanding of uncertainties can be achieved by analysing the contribution of the different sources of uncertainty to the uncertainty of the variables of interest (Step 3.2). Post-treatment procedures are used in order to rank the sources of uncertainty. It is important to note that this last step plays a crucial role. Indeed, the ranking results highlight the variables that truly determine the relevancy of the final results of the study. The methods employed for Step 3.1 and 3.2 are described in the next paragraphs.

15

5

10

20

**2.3.1 Step 3.1: Uncertainty Propagation using Monte-Carlo method**

The Monte-Carlo method used in this study requires random generation of input variables from their probability distributions. This step is the most complex and computationally demanding. In fact, the number of simulations increases with the increase of the uncertainty of each parameter. For this reason, alternative numerical techniques could also be used to reduce the computational time and yet preserve the quality of the statistical results (i.e. quasi Monte-Carlo screening methods). For instance, in this study, we used kriging meta-models (see section 2.2) instead of the original model to perform UQ using Monte-Carlo simulations.

The resulted sampling of a given size N is a  $N \times V$  matrix, V being the number of uncertain parameters. Each row of the matrix  $x^i = (x_1, ..., x_V)^i$  represents a possible configuration of the input parameters of the model, that are, in this study, the fault parameters presented in Figure 2. Corresponding realizations of the output "Y" (the maximum tsunami water 25 height in this study) are generated by successive deterministic simulations with each configuration of the inputs. Statistical estimators of the response  $Y = (Y_1, ..., Y_N) = (G(x^i))_{i \in \{1,...,N\}}$  can therefore be computed from the output as follows:

$$E[Y] = \mu_Y = \frac{1}{N} \sum_{i=1}^{N} G(x^i)$$
(9)

$$Var(Y) = \frac{1}{N-1} \sum_{i=1}^{N} \left[ G(x^{i}) - \mu_{Y} \right]^{2}$$

$$\sigma_{Y} = \sqrt{Var(Y)}$$
(10)
(11)

Where  $E[Y] = \mu_Y$ , Var(Y) and  $\sigma_Y$  are, respectively, the mean, the variance and the standard deviation of the response Y given by the model G. These statistical moments are useful for both the uncertainty propagation and the uncertainty sensitivity analysis (see next paragraph). It must be noted that  $\sigma_Y$  is classically employed for the quantification of the global uncertainty issued from Monte-Carlo simulations.

The convergence order of the Monte-Carlo sampling method is given by the Central Limit Theorem (Faivre et al., 2013) as  $O\left(\frac{1}{\sqrt{N}}\right)$ , implying that a large amount of simulations are necessary to obtain convergent statistics.

**2.3.2 Step 3.2: Global sensitivity analysis**

5

The role of the sensitivity analysis is to determine the strength of the relation between a given uncertain input parameters and the model outputs (Saltelli et al., 2004). In this study, we focused our attention on Global Sensitivity Analysis approaches which rely on sampling based methods for uncertainty propagation, willing to fully map the space of possible model predictions from the various model uncertain input parameters and then, allowing to rank the significance of the input parameter uncertainty contribution to the model output variability (Baroni and Tarantola, 2014). The objectives with this approach are mostly to identify the parameter or set of parameters which significantly impact models outputs (Volkova et al., 2008;Iooss et al., 2008).

- In this study, the GSA approach we focused on is the Sobol index computation, that considers the output hyperspace (x) as a function (Y(x)) and performs a functional decomposition (Iooss and Lemaître, 2015; Iooss, 2011) or a Fourier decomposition (FAST method) of the variance (Saltelli, 2002; Saltelli et al., 1999). With respect to deterministic methods (i.e. Morris, 1991), probabilistic GSA approaches relying on Sobol index computation go one step further, by quantifying the contribution to the output variance of the main effect of each input parameter (Sobol, 2001; Saltelli et al., 1999; Saint-Geours,
- 20 2012;Sobol, 1993). The definition of Sobol Indices is a result of the ANOVA (ANalysis Of VAriance) variance decomposition. To be able to write the variance decomposition, some mathematical hypothesis are required (see i.e. Saltelli et al., 2008, or Faivre et al., 2013), briefly recalled here.

Let us consider an integrable model G on the domain  $A = A_1 \times ... \times A_V$ . It exists an unique decomposition of the model:

$$G(Y) = f_0 + f_1(Y_1) + \dots + f_m(Y_m) + f_{1,2}(Y_1, X_2) + \dots + f_{\nu-1}(Y_{\nu-1}, Y_{\nu}) + \dots + f_{1,\dots,\nu}(Y_1, \dots, Y_{\nu})$$
(12)

Where all the functions fi are mutually orthogonal, implying that  $\langle f_i | f_j \rangle = 0$  if  $i \neq j$ , with the scalar product defined as follows:

$$\langle f | g \rangle = \int_{A} f(x)g(x)\pi(x)dX$$
 (13)

Where  $\pi$  is a probability distribution defined on *A*. Under these assumptions, given a set of *V* independent uncertain parameters *X*, the variance of a response Y = G(X) can be calculated, using the total variance theorem, as follows:

$$Var(Y) = \sum_{i=1}^{V} D_{i}(Y) + \sum_{i< j}^{V} D_{ij}(Y) + \dots + D_{12.V}(Y)$$
(14)

5 Where  $D_i(Y) = Var[E(Y|X_i)]$  and  $D_{ij}(Y) = Var[E(Y|X_iX_j)] - D_i(Y) - D_j(Y)$  and so on for higher order interactions.  $E(Y|X_i)$  is the Y conditional expectation with the condition that  $X_i$  remains constant. The so-called "Soblol" indices" are obtained as follows:

$$S_i = \frac{D_i(Y)}{Var(Y)}, \quad S_{ij} = \frac{D_{ij}(Y)}{Var(Y)}, \quad \dots$$
(15)

These indices express the share of variance of Y (the maximum tsunami height) that is due to a given combination of input parameters (the seismic sources reported in Figure 2). For instance,  $S_1$  is the first order Sobol index. The number of indices grows in an exponential way with the number V of dimensions. So, for computational time and interpretation reasons, (Homma and Saltelli, 1996) introduced the so-called "total indices" or "total effects" that write as follows:

$$S_{Ti} = S_i + \sum_{i < j} S_{ij} + \sum_{j \neq i, k \neq i, j < k} S_{ijk} + \dots = \sum_{l \in \neq i} S_l$$
(16)

Where  $\neq i$  are all the subsets of  $\{1,...,V\}$  including i.

GSA approaches are robust, have a wide range of applicability, and provide accurate sensitivity information for most models (Adetula and Bokov, 2012). Moreover, even if they are theoretically defined for linear mathematical systems, it
15 was demonstrated that they are well suited to be applied with models having nonlinear behaviour and when interaction among parameters occur (Saint-Geours, 2012), as in the present study. For these reasons, these indices were already adopted for the analysis of bi-dimensional hydrodynamic simulations in urban areas (Abily et al., 2016) or of complex coastal models including interactions between waves, current and vegetation (Kalra et al., 2018) and they seem well suited for the present work. For the computation of Sobol' indices, a large variety of methodology are available, as the so-called "extended-FAST"

20 method (Saltelli et al., 1999), already used in previous studies by IRSN (Nguyen et al., 2015). In this study, we used the methodology proposed by Jansen et al. (1994) already implemented in the open source sensitivity-package R (Pujol et al.,

2016). This method estimates first order and total Sobol' indices for all the factors "v" at higher total cost of "v x (p + 2)" simulations (Faivre et al., 2013).

**3** Example of application: building a numerical data-base of tsunamis generated by the Azores Gibraltar Plate Boundary (AGPB) and impacting the French Atlantic Coast**

**5 3.1 Design data-base for meta-models construction**

In this methodological work, we considered a widened AGPB tsunamigenic area as a test case for DTHA. We chose to explore as largely as possible the potential tsunami height along the French Atlantic Coast generated by earthquakes from 34° to 40°N and from 18 to 7°W, encompassing to the East the southern part of Portugal down to Morocco, and reaching the oceanic sea-floor west of the Madeira tore rise (as reported in Figure 3). Because the design database is a learning base for

- 10 meta-modelling, the range of variation of the inputs parameters (column "Design database" in Table 2) need to be large in order to cover both a wide range of earthquake scenarios as well as most of the supposed possible sources of the 1755 Lisbon event and earthquakes coming from the following catalogues: http://www.isc.ac.uk; http://earthquake.usgs.gov; http://www.globalcmt.org; http://www.bo.ingv.it/RCMT/. Thus, if correctly estimated, meta-models will be able to reproduce the model behaviour for a large range of variations of the seismic inputs parameters, including physical scenarios
- 15 from geological studies of the zone.

In order to build the *design database*, fault parameters as defined in section 2.1 and Table 2 were sampled randomly and independently with the Monte-Carlo method and supposing uniform distributions. The uniform distribution was chosen in order to build meta-models able to reproduce tsunamis heights generated by various tsunamigenic sources with the same accuracy. The resulting earthquake magnitudes are computed using the sampled parameters with equation 2. The shear

- 20 modulus chosen for the magnitude estimation is a constant value assumed to be equal to 30 GPA. This design database is a matrix which associates to a given combination of fault parameters estimates the maximum simulated water height at each point of the numerical grid and also at four selected locations along the French Atlantic Coast called *gauges* (Table 3 and Figure 3), namely, from North to South, "Saint-Malo", "Brest", "La Rochelle" and "Gastes". The maximum tsunami water height is the relevant parameter when estimating tsunami hazard. The *design database* contains 5839 scenarios split in a
- 25 *training set* of 4672 scenarios and a *testing set* of 1167 tsunamis scenarios, used for the meta-model validation and the residual analysis. The water height characteristics associated to these scenarios are reported in Table 4.

[revised manuscript text omitted]

**3.2.2 Comparison with Allgeyer et al. (2013) results along the French Atlantic Coast**

- Another possible validation of the meta-models consistency is the comparison with a previous study from Allgeyer 5 et al. (2013). This comparison is of interest in order to confirm the ability of the constructed meta-models to reproduce the order of magnitude of the modelled tsunami height at a given location. In their study, the authors analysed the impact of a Lisbon-like tsunami on the French Atlantic Coast through numerical modelling. Especially, the authors focused on the simulated maximum tsunami water height in the North Atlantic associated to three different sources for the 1755 events derived from Johnston (1996), Baptista et al. (2003) and Gutscher et al. (2006), for a total of five tsunami scenarios. The
- 10 same scenarios were simulated with our meta-models for the four French Atlantic Gauges. Even if the results obtained with meta-models (see Table 6) cannot be really compared with results from the authors (Allgeyer et al., 2013), which are obtained with a more refined grid, spacing from 1' to 0.3" near to the "La Rochelle" harbour, the general pattern reported in Allgeyer et al. (2013) is confirmed. Especially, the authors pointed out that the coastal area offshore French Brittany (from 46 °N to 48 °N) were impacted by amplitudes of 0.6 to 0.8 m for the Johnston source (1996) and not exceeding the 0.5 m for
- 15 the Baptista et al. (2003) and the Gutscher et al. (2006) sources. These results are consistent with those obtained with our meta-models for the *Brest* and *La Rochelle* gauges which represent this coastal area (Table 6). As a consequence, the results obtained with meta-models by using some physical sources by literature seem reasonable, physically speaking.

**3.3 Construction of a tsunami data-base for the French Atlantic Gauges**

- Starting from the validated meta-models we finally build a database of tsunamis generated by the considered AGPB 20 area at four French Atlantic Gauges. This numerical data-base should permit to cover all the possible tsunamis height impacting a given area and then to assess the tsunami hazard level (MCS\_h). Moreover, in a context with strong uncertainties on the seismic source parameters, this methodology should also permit to verify if the tsunamis height associated with the MCS\_p parameters are robust with respect to the area to protect.
- AGPB is a zone prone to tsunamis and its investigation already lead to the publication of tsunami catalogues for the 25 zone (Baptista and Miranda, 2009;Kaabouben et al., 2009). However, for the purpose of this methodological paper, the AGPB area modelled here, which encompass different seismotectonic domains, was split in a very simplistic way into two main seismic source zones. This was done to represent a western domain where normal to transtensive earthquakes mainly occur within an oceanic crust and an eastern domain to the East where reverse to transpressive earthquakes mainly occur on an oceanic crust and a thicker continental crust (Buforn et al., 1988;Molinari and Morelli, 2011;Cunha et al., 2012). As a first
- 30 order analysis, we considered the *Western domain* west of the 10°W meridian and the *Eastern domain* east of this meridian, coinciding roughly with the base of the continental slope facing the Portuguese coastline. Even if the zonation adopted in our study is not in agreement with the actual knowledge of the deep structure of the Gulf of Cadiz and the Gloria Fault zone as

given by Martínez-Loriente et al. (2014) and Batista et al. (2017), such a simplified zonation represents a first order approach of the AGPB seismotectonics, consistent with the global objective of the study aimed at illustrating a methodological approach to quantifying all possible tsunamis heights that can potentially impact a given area. A more accurate consideration of faults parameters that are better constrained in some regions of the AGPB area would be more in line with the work presented by Roshan at al. (2016) or with a complete and robust deterministic MCS p approach.

5

The main fault characteristics considered to perform earthquake scenarios in both eastern and western domains are reported in Table 2, representing our interpretation of data contained in (Buforn et al., 1988; Molinari and Morelli, 2011;Cunha et al., 2012). The considered seismogenic thickness takes into account the depth of the observed seismicity and crustal structure for each zone as well as a part of the upper mantle that can potentially be mobilised during major

- earthquakes: the Atlantic shelf seismogenic thickness is considered to be of the order of 20 km (after Baptista et al., 2017), 10 and 60 km for the European shelf (after Silva et al., 2017). The fault parameters are considered uniformly distributed and are randomly sampled in their range of variation. We finally filtered the resulting database according to an aspect-ratio criterion, allowing the ratio between length and the width of the faults not to exceed the value of 10, which correspond to an upper bound of what is observed in nature (Mc Calpin, 2009). The final database contains nearly 50,000 tsunami scenarios,
- resulting in earthquake magnitudes varying from 6.7 to 9.3 (Table 2), depending on the explored earthquake sources 15 characteristics and calculated from Eq 1. This range of magnitudes is consistent with the magnitude range of the design database. It must be noted that the higher magnitudes considered for the AGPB (higher than the maximum value estimated for the 1st November 1755 earthquake of 8.8, according to Johnston (1996), correspond to some extreme scenarios which represent less than 0.2% of the total scenarios of the numerical data-base. However, our choice is to not ignore these very
- unlikely scenarios. In fact, the objective of this work is to propose a methodology which could be applied in a context of 20 poor knowledge and could permit to ensure that MCS\_p is "really" the appropriate scenario to consider for a given location. MCS\_h is deduced from the tsunamis height distribution at a given location. Depending on the specific hazard target (civil or industrial facilities), and its location respect to the source zone, the end-user of this methodology will need to decide which level of water height to choose from the obtained distribution (MCS\_h).
- 25 The resulting *global database* (Table 2) is composed by the union of the *western* and the *eastern* databases scenarios. The tsunamis height characteristics associated to the four gauges are reported in Table 7. It must be noted that the maximum tsunami height of the global database are lower than the tsunamis height of the design database and that the convergence of statistics (mean water height) is largely achieved, as reported in Figure 6. The achievement of the convergence of numerical results is an important aspect of the adopted methodology and suggests that it is not necessary to
- further explore the space of inputs parameters in order to assess CMS h: statistically, the distribution of tsunami heights is 30 representative of the source variability in the defined space of parameters.

In conclusion, the built tsunamis database (the *global database*) should represent a wide sample of tsunami heights potentially generated by the AGPB (the Western and the Eastern domains). The database is analysed in the next section with a focus on DTHA. At first, the resulting tsunami height distribution is analysed in order to propose a hazard level for the French Atlantic Gauges (CMS\_h). Then, this choice is discussed in the light of tsunami heights higher than the proposed hazard level (section 4.2) and of tsunamis heights generated by a MCS\_p- like scenarios (section 4.3).

**4 Analysis of the numerical tsunamis database**

**4.1 Assessment of a hazard level for the French Atlantic Coast (MCS\_h)**

- 5 The distribution of tsunami heights resulting from each considered tsunamigenic source is reported and compared in Figure 7 for each gauge. It can be observed that the tsunami heights generated by the *western shelf* are globally lower than those generated by other sources. This is not surprising given the different depths explored for the two regions (see Table2). Indeed, the highest tsunamis heights resulting from the global database are globally generated by the *eastern shelf* for both the *southern* and the *northern gauges*.
- From an engineering point of view, the tsunami hazard "MCS\_h" can be directly deducted from Figure 7 (black line). Roshan et al. (2016) proposed to use a hazard reference level based on the mean tsunami height resulting from their distribution. However, as reported in section 3.3, the constructed tsunami data-base integrates all the "likely" and "unlikely" tsunami scenarios from AGPB. As a consequence, we prefer to set the tsunami hazard reference level at the value "μm+2σm" of the modelled water height for each gauge and to consider an uncertainty associated to this level equal to the +/-2σr residual (summarized in Table 8 for each gauge). This choice needs to be justified on a case by case basis. In this specific
- example the choice is guided, firstly, by considering the shape of the tsunami heights distribution resulting from UQ (Figure 7, black line). In fact, the distributions for each gauge are centred on the lower tsunami heights while the maximum modelled tsunamis heights are very large with respect to the tsunamis statistical parameters of the distribution (reported in Table 8). As a consequence, this choice appears numerically reasonable in a context of DTHA as it considers both the low tsunamis
- 20 height and the maximum modelled tsunamis height. Secondly, considering the rough approximation of the earthquake input parameters (and the consequently wide range of seismic magnitude distributions) used for the construction of the global database, this hazard level appears more conservative in a context of DTHA.

The proposed hazard level corresponds to nearly 200 tsunamis scenarios (+/- 0,01 m respect to the chosen value) of the data-base, located all over the AGPB but preferentially in the eastern part for the "Saint-Malo" and the "Brest" gauges and exclusively in the eastern part for the southern gauges for "La Rochelle" and "Gastes" gauges (Figure 8). The MCS\_h scenarios are associated with earthquake magnitudes in the 7.5 to 9 range for all the gauges, with a mean of 8.47 for the northern gauges and of 8.67 for the southern gauges. The robustness of this choice is analysed in the next sections. Especially, it is of interest to analyse the strongest tsunamis of the numerical database (see section 4.2) and to evaluate if this hazard level is robust with respect to results which could be obtained with a more classical scenario-based approach "MCS p" (see section 4.3).

**4.2 Focus on the strongest tsunamis of the numerical database**

As reported in Table 8, there are strong differences, for all the gauges of the numerical database, between the maximum simulated tsunami height and the " $\mu_m + 2\sigma_m$ " water height and the hazard level chosen in this study (MCS\_h), As a consequence, from a methodological point of view, and with the objective of assessing the tsunami hazard level from the UQ study, it is of interest to un understand which scenarios are stronger than the proposed hazard level.

5

With this aim, we first analysed the magnitude distribution " $M_w$ " of the tsunami scenarios of the global database (grev curve in Figure 9a) generating tsunamis heights higher than the proposed *hazard level* at each gauge (coloured lines in Figure 9a). Not surprisingly, the strongest tsunami heights correspond to the strongest seismic events of the database (Figure 9a). In particular, more than 80% of the tsunami heights higher than the chosen hazard level (" $\mu_m + 2\sigma_m$ ") are associated

- 10 with a seismic magnitude higher than 8.5. These tsunami events completely represent the tail of the earthquake magnitude distribution (the grey histogram in the Figure 9a). Nevertheless, some discrepancies are observed between the results of the southern gauges (La Rochelle and Gastes) and the results obtained for the northern gauges (Brest and Saint-Malo). In fact, even if it seems that only the very strong earthquakes (magnitude higher than 8.5) can generate a tsunamis height higher than " $\mu_m + 2\sigma_m$ " for the middle-south of Atlantic French coast (more than the 95% of the tsunamis scenarios of the sample), this is
- more ambiguous for the northern Gauges. In this case, nearly 30% of the tsunamis higher than the hazard level can be 15 generated by an earthquake of magnitude in the range [7.8 - 8.5], which includes the upper range of the western shelf.

As shown in Figure 9b, 50% of the simulated tsunamis in this [7.8 - 8.5] magnitude range generating tsunami heights greater than chosen hazard level " $\mu_m + 2\sigma_m$ " originate in the western shelf. In fact, the northern gauges (Saint-Malo and *Brest*) are practically the only ones exposed to tsunamis generated by the western shelf.

20 These results are interesting and show that the strongest water height obtained through UO (stronger than MCS h) are not all associated with the strongest magnitudes, which are typically considered in the MCS\_p approach. On the contrary, most of the strongest tsunamis heights at the northern gauges could be generated by the western shelf, for seismic magnitudes in the range [7.8 - 8.5]. In conclusion, these results also suggest that, beyond earthquake magnitudes, the position and the orientation of the faults are influent parameters. In terms of *hazard level*, which is the original objective of this analysis, one can consider, at first, that tsunamis higher than the chosen hazard level are mainly generated by seismic 25 magnitudes stronger than 8.5, even if some rare combinations of tsunami input parameters can lead to higher tsunami heights.

**4.3 Comparison with a classical MCS p approach associated to a Lisbon-like 1755 tsunami scenario**

As reported in the introduction, DTHA is classically assessed by means of considering particular source scenarios 30 (usually maximum credible scenario) and the associated maximum tsunami height is generally retained as hazard level (MCS\_p). A more sophisticated method was recently proposed by Roshan et al. (2016). The authors tested around 300 tsunamis scenarios in a [8 - 9.5] magnitude range associated with various faults potentially impacting the Indian coast. Finally, the authors suggested that an appropriate water level for hazard assessment (e.g. mean value or mean plus sigma value) should be retained. They proposed the mean value of the simulated water heights, as test, by considering that this value may need to be revisited in the future.

- In the case of the tsunamis hazard associated with the AGPB, the 1755 Lisbon tsunami is the classical reference 5 scenario. For the critical analysis of the global numerical database we decided to complement our study with two specific and nearly deterministic scenarios considered as the most-likely sources generating the Lisbon 1755 tsunami, namely the Gorringe and Horseshoe structures (Buforn et al., 1988;Stich et al., 2007;Cunha et al., 2012;Duarte et al., 2013;Grevemeyer et al., 2017). Both structures were modelled taking into account available maps (Cunha et al., 2012;Duarte et al., 2013) and fault parameters (Stich et al., 2007;Grevemeyer et al., 2017) summarized in Table 2. These scenarios have been chosen
- 10 because they are representatives of the MCS\_p approach for the study zone. They permit to evaluate the global tsunami database, which was designed by roughly zoning European (eastern) and Atlantic (western) shelves. In fact, the tsunami hazard induced by the Horseshoe scenario, located at the limit between these two zones of the global database, may be partly contained in the European contribution of the database. On the contrary, the Gorringe scenario represents a pure "European like" scenario within the Atlantic zone, hence excluded by the global database.
- For the computation of the tsunami heights associated with these scenarios, fault parameters are considered uniformly distributed and are randomly sampled in their range of variation (Table 2), for a total of nearly 10,000 tsunami scenarios. The magnitude range associated with these tsunami scenarios varies from 7.7 to 8.9 and the hazard level associated to these faults corresponds to the " $\mu_m$ " modelled water height. This hazard level for MCS\_p approach was arbitrarily chosen, considering that the philosophy of this kind of approach is to take into account only the worst possible scenarios sources. In this sense, this value is in agreement with the choice reported in Roshan et al. (2016).

In Table 8 we summarized the tsunami hazard height from the global database and from Gorringe and Horseshoe faults for each gauge, with the associated uncertainty (corresponding to the residuals +/- 2*σr*). These results suggest that the strongest tsunamis are globally generated by the Gorringe bank and that the proposed *hazard level* from the total database (MCS\_h) is representative of the scenarios based tsunamis (MCS\_p), once the uncertainty associated to meta-model predictions is considered (see Figure 10). In conclusion, the proposed hazard level resulting from UQ seems reasonable in light of the results obtained through a scenario-based-like approach, at least for the tsunamis impact along the French Atlantic coast due to sources off of Portugal.

**4.4 Results from sensitivity analysis: the influence of the seismic-source parameters: application of Step 3.2**

Homma and Saltelli (1996) introduced the total sensitivity index which measures the influence of a variable jointly 30 with all its interactions. If the total sensitivity index of a variable is zero, this variable can be removed because neither the variable nor its interactions at any order have an influence on the results. This statistical index (called Sobol index "ST" in this paper, Eq. 16), is here of particular interest in order to highlight the earthquake source parameters that mostly control the tsunamis height at each tested gauge. In Figure 11, we reported the total Sobol index for the four meta-models of the French Atlantic Gauges computed with the methodology proposed by Jansen et al. (1994) using the sensitivity-package R (Pujol et al., 2016). The accuracy of Sobol indices performed with Jansen's method depends on the number of model evaluations. For instance, in this study, we performed nearly 20 000 simulations using meta-models for the computation of Sobol indices. Results show that the slip parameter is globally the most-influencing parameter for all the French Atlantic Gauges meta-

- 5 models, which is quite obvious considering that the fault-slip directly conditions the ocean floor deformation and hence the tsunami amplitude. However, Figure 11 suggests that the most influencing parameters for the four gauges are slightly different, depending on their location. One can differentiate results obtained for the southern gauges (i.e. La Rochelle & Gastes) from those obtained at northern gauges (i.e. Saint Malo & Brest):
  - For the southern gauges, *width* is the second most relevant parameter, especially at La Rochelle where it is almost as important as the *slip* parameter. Other important parameters are *length* and *rake*, suggesting that for these gauges, fault source parameters in terms of magnitudes (depending on *width*, *slip* and *length* according to equations 1 and 2) and kinematics are the most important in generating hazard;
  - For the northern gauges, apart from *slip*, *strike*, *width* and *rake* are also important, but slightly less than the *longitude* parameter. This means that the location of the source is here of major importance. A possible physical reason could be associated to the lack of the natural barrier composed by the north of Spain and Portugal which protects southern gauges from AGPB related tsunamis compared to northern gauges. As a consequence, the northern gauges should be more exposed to hazard in comparison to southern gauges, located in the shadow of Portugal and Spain.
- In conclusion, these results also confirm that, beyond earthquake magnitudes, the position and the orientation of the 20 faults are influents parameters, in agreement with the analysis of the previous paragraph (see 4.2). Thus, for a given magnitude, a tsunami generated by a "*well located-oriented*" fault would be potentially more hazardous, at least for the sites along the northern French Atlantic Coast. In other words, the French Atlantic coast is in the shadow of the tsunamis generated by the stronger continental shelf sources and more sensitive to the tsunamis generated in the oceanic shelf.

**5** Conclusions and perspectives**

10

- The methodological research work presented in this paper was performed in order to test the interest of UQ for the assessment of the DTHA generated by earthquakes. We propose a new methodology for DTHA, consisting in the assessment of tsunami hazard through the analysis of all the possible tsunami heights at a given location (MCS\_h). This concept goes beyond the definition of the Maximum Credible Scenario source parameters (MCS\_p) classically reported in the literature (JSCE, 2002;Lynett et al., 2016) and employed for DTHA. MCS\_h is evaluated through UQ allows the exploration of the
- 30 tsunamigenic potential of a seismic zone in a rigorous and complete way. It can be considered as an extension of the previous innovative work from Roshan et al. (2016), which first improved the classical deterministic approach by focusing its analysis on some selected earthquake source parameters.

The methodology is based on the use of kriging meta-models that intensively explore all the possible tsunamis generated by a given seismic area. It is the first time, to our knowledge, that meta-models are used in this way in the context of DTHA (but also for PTHA). The meta-models are first constructed through a uniform exploration of the space of the inputs parameters of the seismic area and then tested and validated through a series of statistical indicators (correlation

- 5 coefficient, residuals analysis, MSE). These performance-tests allowed us to conclude that, even if the maximum residuals (differences between model and meta-model results) are very high for the four selected locations along the French Atlantic Coast (see Table 5), the meta-models are able to reproduce the model behaviour and they can be used for the construction of the numerical tsunami database. Nevertheless, when meta-models are used instead of the original model for the predictions of the tsunami heights, we propose to add an uncertainty corresponding to the "+/- $2\sigma_r$ " residual value measured at each
- 10 gauge. Moreover, the realism of the tsunamis height simulated with the meta-models was also ensured through a comparison with the numerical results presented in Allgeyer et al. (2013) for the French Atlantic Coast. Thus, the constructed metamodels could be employed in a further study to roughly evaluate the impact of other seismic scenarios from the AGPB and impacting the French Atlantic Coast.
- In the example of application presented in this work we built our meta-models using a design data-base of nearly 5 000 15 tsunami scenarios. The number of simulations is large and required intensive parallel computing using GPU. However, considering the large number of model evaluations necessary for UQ (more than 50 000) and GSA (more than 20 000), the meta-models appear as the more adapted tool for the proposed methodology.

We test this methodology by building a numerical database (of nearly 50,000 tsunamis scenarios) of tsunamis potentially generated by the AGPB and impacting the French Atlantic Coast at four sites of interest. As frequently reported

- 20 in the paper, this is only a test-case and all the results and the analysis presented in section 4 are an example of how an enduser of the methodology should use these results for a DTHA application. This database should permit to explore the numerous uncertainties related to the AGPB characteristics. Considering the convergence of the mean modelled tsunami height, it is not necessary to further explore the AGPB inputs parameters in order to refine the assessment of the CMS\_h for the French Atlantic Gauges since statistically we have shown that the resulting tsunamis height distribution is representative
- 25 of the source variability. This may not be the case, however, for sites located closer to the AGPB region. In such regions a more refined zonation is necessary. However, the proposed methodology remains unchanged.

Tsunami hazard (MCS\_h) was assessed directly from the tsunamis height distribution of the database (see section 4.1). Considering the rough zonation of the AGPB and the shape of the tsunami heights distribution we propose in this case to attribute to each French Gauge a tsunamis hazard level equal to the modelled tsunami height " $\mu_m + 2\sigma_m$ " with an

- 30
  - associated uncertainty equal to the residual "+/- $2\sigma_r$ ". To validate this choice, we analyse the strongest tsunamis (higher than the proposed *hazard level*) of the distribution. In our example, this analysis permits to conclude that, globally, the position and the orientation of the fault are influent parameters. In other words, for a given magnitude, a tsunami generated by a "*well located-oriented*" fault would be potentially more hazardous, at least for the sites located along the northern French Atlantic Coast. In this sense, the UQ study permitted to explore some rare combinations of seismic inputs parameters generating

higher tsunamis level (see Figure 9b). Moreover, the UQ permits to underline that globally the tsunamis generated by the *western shelf* are largely less impacting than the tsunamis generated by the *eastern shelf*, on average. Furthermore, GSA performed with Jansen's method permits to confirm these first results. Especially, the magnitude generating the tsunamis in the AGPB (depending on width, slip and length) is more relevant depending on the location of the gauge (the impacted

5 zone). For instance, only the earthquakes inducing large displacements of the sea-floor (large *slip*) are potentially capable to affect the middle/south of France. However, concerning the gauges located to the north of the French Atlantic Coast (*Brest* and *Saint-Malo*), other influencing parameters are, apart from the slip, the location (*longitude*) and the orientation (*strike* and *rake* parameters) of the fault sources. These results are also in agreement with results presented in (Allgeyer et al., 2013).

The data base was also tested through a comparison with the tsunamis generated by a Lisbon-like 1755 earthquake (paragraph 4.3). The latter representing a comparison between the global results from UQ (MCS\_h) and a more classical MCS\_p approach. Globally, the *hazard level* issued by the total database seems robust and representative of the tsunamis from *Gorringe* and *Horseshoe* banks once the uncertainty of numerical results is considered (see Figure 10), at least for French gauges off the Atlantic coast. Even if this work is mainly methodological, this results seems to confirm that the proposed MCS\_h is adapted to the French Atlantic Coast as the tsunamis height from these scenarios are the strongest of the

- 15 tsunamis heights resulting from the global data-base. However, to be conclusive, the same analysis should be conducted with other MCS\_p hypothesis for the AGPB. Moreover, it must also be noted that the choice of the hazard level is arbitrary and it depends on the chosen area, the available information about potential tsunami sources and the analysis of the obtained tsunami heights distribution.
- Tsunami hazard is analysed only off the French Atlantic Coast and by using, as input data for meta-model's construction, the deterministic simulations performed with a rough computation grid-resolution of 2" degrees (~3.6 km) which are not adapted to reproduce the influence of bathymetry on tsunami propagation in the vicinity of the coastal areas. For an operational DTHA along the French Atlantic Coast, it should be necessary, at first, to improve the actual numerical model in order to better represent the tsunamis run-up and the inundated areas, with a more accurate bathymetric grid. Then, the meta-models could be constructed with the methodology reported in section 2 and with a design data-base permitting to
- 25 explore as largely as possible the tsunamigenic potential from AGPB. Once validated, these meta-models could be used for DTHA as reported in section 3.3. However, in order to really assess the hazard level for the French Atlantic Coast, the constructed global data-base should be compared, at least, to the main MCS\_p hypothesis for this area, to ensure that the chosen hazard level is not underestimated with respects to these scenarios.
- In perspective, for an operational application of the proposed methodology to locations closer to the source zones 30 (i.e. Portugal or Spain, Morocco), the sensitivity analysis will probably show that slip, length and width are the dominant parameters. In such cases, a more in-depth source analysis is necessary and the research performed in the region should be carefully considered (i.e. (Vilanova and Fonseca, 2007;Woessner et al., 2015;Matias et al., 2013;Omira et al., 2016;Omira et al., 2015)). However, uncertainties will always remain. Thus even in such cases, where more refined databases need to be established, the proposed approach should be of interest. In fact, as shown by the Fukushima accident, it is today necessary

to go beyond the classical MCS\_p approach and to focus on all the possible tsunami heights at a given location (MCS\_h). In this sense, we consider that the modelled tsunamis heights of our data-base are probably already representatives of the tsunamis heights generated by these scenarios for the French Atlantic Coast.

- Finally, other analysis would be of interest for the improvement and the application of the methodology. At first, it should be possible to build more meta-models, in order to cover other area in locations where historical data from the 1755 Lisbone-tsunami are available (i.e. Santos and Koshimura, 2015;Baptista et al., 1998), thus permitting to further validate the constructed meta-models. Then, it would be of interest to test other hypothesis during the UQ step necessary for the assessment of the hazard (MCS\_h), as different probability distribution of inputs parameters, a more realistic non-uniform slip distribution, considering its impact on tsunami heights (Geist and Dmowska, 1999), or, to analyze other approximation
- 10 of the fault geometry.

As perspective, the example presented in this study for the AGPB suggests that it could be of interest to introduce the meta-models in the systems developed for tsunami early warning, considering the low computational time inherent to this statistical tool.

[revised manuscript text omitted]

5 **Table 4:** Maximum water height associated with the design database;  $\mu$ ,  $\sigma$  and Max correspond to the mean, standarddeviation and maximum modelled values.

| Congos      | Training set | Validation set |                    |                  |                   |         |  |  |
|-------------|--------------|----------------|--------------------|------------------|-------------------|---------|--|--|
| Gauges      | L_CV [-]     | R2 [%]         | μ r [m] | $+\sigma_{r}[m]$ | $+2\sigma_{r}[m]$ | Max [m] |  |  |
| Saint-Malo  | 0.004        | 87%            | 0.000              | 0.066            | 0.131             | 0.498   |  |  |
| Brest       | 0.012        | 94%            | 0.001              | 0.112            | 0.224             | 0.982   |  |  |
| La Rochelle | 0.029        | 76%            | 0.001              | 0.170            | 0.339             | 1.323   |  |  |
| Gastes      | 0.021        | 80%            | 0.004              | 0.148            | 0.293             | 1.392   |  |  |

**Table 5:** Summary of meta-model evaluations with the testing and the training data base;  $\mu_r$ ,  $\sigma_r$  and Maxr correspond to the mean, standard-deviation and maximum residual respectively. L\_CV is defined in Eq. 5.

|                              | Hypothesis reported in Allgeyer et al. (2013) |                |                   |             |               |             |                   |            | Results from meta-models
constructed in this study |                      |              |                       |               |
|------------------------------|-----------------------------------------------|----------------|-------------------|-------------|---------------|-------------|-------------------|------------|-------------------------------------------------------|----------------------|--------------|-----------------------|---------------|
| Source                       | Strike
[°]                                 | Length
[km] | Dip
[°] | Rake
[°] | Width
[km] | Slip
[m] | Lon
[°] | Lat
[°] | Depth
[km]                                         | Saint
Malo
[m] | Brest
[m] | La
Rochelle
[m] | Gastes
[m] |
| Johnston
(1996)           | 60                                            | 200            | 40                | 90          | 80            | 13.1        | 11.4
5         | 36.9
5  | 27                                                    | 0.37                 | 0.85         | 0.57                  | 0.41          |
| Baptista
et al.
(2003) | 250                                           | 155            | 45                | 90          | 55            | 20          | -8.7              | 36.1       | 20.5                                                  | 0.20                 | 0.45         | 0.38                  | 0.33          |
| Gutscher                     | 21.7                                          | 96             | 24                | 90          | 55            | 20          | -10               | 36.8       | 20.5                                                  | 0.24                 | 0.39         | 0.42                  | 0.28          |
| et al.                       | 346                                           | 180            | 5                 | 90          | 197           | 20          | -7.5              | 35.5       | 15.1                                                  | 0.35                 | 0.54         | 0.78                  | 0.40          |
| (2006)                       | 346                                           | 180            | 30                | 90          | 12            | 20          | -8.6              | 35.3       | 3.2                                                   | 0.10                 | 0.21         | 0.10                  | 0.10          |

**Table 6:** Maximum tsunamis height computed with meta-models using seismic sources simulated in (Allgeyer et al., 2013)for the 1755 Lisbon-tsunami.

15

|                                       | Saint Malo | Brest | La Rochelle | Gastes |
|---------------------------------------|------------|-------|-------------|--------|
| μ m [m]                    | 0.09       | 0.18  | 0.19        | 0.13   |
| $\mu_{m} + \sigma_{m} [m]$            | 0.17       | 0.34  | 0.36        | 0.27   |
| $\mu_{\rm m}$ + $2\sigma_{\rm m}$ [m] | 0.24       | 0.51  | 0.54        | 0.41   |
| Max m [m]                  | 0.77       | 2.01  | 1.81        | 1.92   |

**Table 7:** Maximum water height associated with the global database;  $\mu_m$ ,  $\sigma_m$  and  $Max_m$  correspond to the mean, standard-
deviation and maximum modelled values.

|             | HorseShoe
(µm)
[m] | Gorringe
(µm)
[m] | $\begin{array}{c} Global\\ Database\\ (\mu_m+2\sigma_m)\\ [m] \end{array}$ | Meta-model
uncertainty
(+/-2σ r ) [m] | Max m
[m] |
|-------------|--------------------------|-------------------------|----------------------------------------------------------------------------|--------------------------------------------------------|-------------------------|
| Saint Malo  | 0.18                     | 0.25                    | 0.24                                                                       | +/- 0.13                                               | 0.77                    |
| Brest       | 0.31                     | 0.53                    | 0.51                                                                       | +/- 0.22                                               | 2.01                    |
| La Rochelle | 0.26                     | 0.32                    | 0.54                                                                       | +/- 0.34                                               | 1.81                    |
| Gastes      | 0.18                     | 0.29                    | 0.41                                                                       | +/- 0.29                                               | 1.92                    |

**Table 8:** Numerical tsunami database and scenarios-based hazard levels. For each gauge, the uncertainty corresponds to the  $+/-2\sigma_r$  also reported in Table 5. The maximum modelled tsunamis height is also reported in the next column.

**STEP 1: PROBLEM SPECIFICATION**

Goal: enable the modelling of maximum tsunami heights generated by a chosen tsunamigenic area

**Method:** generation of tsunamis according to Okada (1985) surface deformation model and 2D propagation of tsunamis using Boussinesq equations as reported in Allgeyer et al. (2013)

**STEP 2: META-MODELS CONSTRUCTINO AND VALIDATION**

**Goal:** replace the numerical model, reduce the computational coast, extensive exploration of the seismic area

Method: kriging meta-model (see section 2.2 and APPENDIX 1)

| STEP 3: DTHA ASSESSMENT THROUGH UQ AND GSA                                                                                                                                               |                                                                                                                                           |  |  |  |  |  |  |
|------------------------------------------------------------------------------------------------------------------------------------------------------------------------------------------|-------------------------------------------------------------------------------------------------------------------------------------------|--|--|--|--|--|--|
| Goal: (i) estimation of all the possible tsunami heights generated by a seismic area and impacting the target zone; (ii) assessment of MCS_h (see sections 3.3, 4.1, 4.2 and 4.3) | Goal: critical analysis of the relevance of the seismic source parameters on tsunami heights at a given location (see section 4.4) |  |  |  |  |  |  |
| Method: Monte-Carlo simulations using kriging meta-model instead of the original model                                                                                            | Method: Computation of Sobol Indices using the methodology proposed by Jansen et al. (1994). See section 2.3.2                     |  |  |  |  |  |  |

Figure 1: Global methodology proposed in this study.